# Single-cell RNA sequencing identifies shared differentiation paths of mouse thymic innate T cells

Minji Lee[1,7], Eunmin Lee [2,7], Seong Kyu Han[3,7], Yoon Ha Choi[2], Dong-il Kwon[1], Hyobeen Choi [3], Kwanghwan Lee[3], Eun Seo Park[2], Min-Seok Rha[4], Dong Jin Joo [5,6], Eui-Cheol Shin[4], Sanguk Kim [3✉], Jong Kyoung Kim [2✉] & You Jeong Lee [1✉]

Invariant natural killer T (iNKT), mucosal-associated invariant T (MAIT), and γδ T cells are innate T cells that acquire memory phenotype in the thymus and share similar biological characteristics. However, how their effector differentiation is developmentally regulated is still unclear. Here, we identify analogous effector subsets of these three innate T cell types in the thymus that share transcriptional profiles. Using single-cell RNA sequencing, we show that iNKT, MAIT and γδ T cells mature via shared, branched differentiation rather than linear maturation or TCR-mediated instruction. Simultaneous TCR clonotyping analysis reveals that thymic maturation of all three types is accompanied by clonal selection and expansion. Analyses of mice deficient of TBET, GATA3 or RORγt and additional in vivo experiments corroborate the predicted differentiation paths, while human innate T cells from liver samples display similar features. Collectively, our data indicate that innate T cells share effector differentiation processes in the thymus.

[1] Division of Integrative Biosciences and Biotechnology, Pohang University of Science and Technology (POSTECH), Pohang 37673, Republic of Korea.
[2] Department of New Biology, DGIST, Daegu 42988, Republic of Korea. [3] Department of Life Sciences, Pohang University of Science and Technology (POSTECH), Pohang 37673, Republic of Korea. [4] Laboratory of Immunology and Infectious Diseases, Graduate School of Medical Science and Engineering, KAIST, Daejeon, Korea. [5] Department of Surgery, Yonsei University, College of Medicine, Seoul, Republic of Korea. [6] The Research Institute for Transplantation, Yonsei University, College of Medicine, Seoul, Republic of Korea. [7] These authors equally contributed: Minji Lee, Eunmin Lee, Seong Kyu Han.
✉email: sukim@postech.ac.kr; jkkim@dgist.ac.kr; youjeong77@postech.ac.kr

Memory T cells are a diverse population that develops via multiple pathways[1,2]. Conventional or adaptive memory T cells develop as a result of foreign antigenic interactions, whereas virtual or homeostatic memory T cells respond to IL-7/IL-15 cytokines and self-antigens in the lymphopenic environment[3]. Alternatively, innate or innate-like T cells developmentally acquire a memory phenotype in the thymus expressing PLZF (encoded by *Zbtb16*), which reciprocally regulate the development of Eomes-expressing bystander memory T cells by secreting IL-4[1,2]. This category of memory T cells includes invariant natural killer T (iNKT), mucosal-associated invariant T (MAIT), and γδ T cells, that together represent abundant T cell subsets that migrate throughout the body and play important roles in early response to infection by a broad range of pathogens. However, it remains unclear how their effector differentiation is developmentally regulated in the thymus.

iNKT and MAIT cells have canonical TCRVα/Jα and biased usage of TCRVβ chains that recognize glycolipid or vitamin metabolites via CD1d and MR1, respectively[4–6]. The development of iNKT cells occurs via the lineage differentiation model, in which NKT1, NKT2, and NKT17 cells all develop from common progenitors[7–9]. In contrast, the development of MAIT cells has been described as a linear maturation model, in which CD24$^{hi}$ (stage 1) cells serially mature into CD24$^{low}$ (stage 2) and CD44$^{hi}$ (stage 3) cells[10]. Stage 3 MAIT cells include cells expressing TBET or RORγt, and are designated as MAIT1 and MAIT17 cells, respectively. A third type of innate T cells, the γδ T cells, originate from double negative (DN) 2 or 3 thymocytes, and their transcriptional profiles based on their T-cell receptor (TCR) usage defined three distinct effector subsets secreting IFN-γ, IL-4, and IL-17[11]. The presence of analogous effector subsets of innate T cells suggests that they share common developmental programs, but previous studies were based on different developmental models and it was not possible to directly compare their thymic ontogenies.

In this report, we simultaneously analyze all three types of innate T cells to understand their developmental landscape at the clonal level using single-cell RNA sequencing (scRNA-seq) paired with TCR clonotyping analysis in the same cell. As a result, we define their analogous effector subsets and validate the presence of predicted developmental intermediates of these cells. In addition, we find that the development of innate T cells commonly occurs via lineage differentiation, with a highly diverse TCR repertoire of immature cells undergoing clonal selection and expansion. These features are also seen in human liver innate T cells, indicating that innate T cells have shared developmental program in mice and humans.

## Results

**Innate T cells have analogous effector subsets.** We previously showed that iNKT cells differentiate into NKT1, NKT2, and NKT17 effector cells in the thymus, which express TBET, GATA3, and RORγt with low, high, and intermediate levels of PLZF, respectively[8]. In this study, using the same combination of transcription factors, we defined the analogous effector subsets in MAIT cells, designated as MAIT1, MAIT2, and MAIT17, and in γδ T cells, designated as Tγδ1, Tγδ2, and Tγδ17 cells (Fig. 1a). MAIT1 and MAIT17 cells were previously reported[12] and we further discovered PLZF$^{hi}$ MAIT2 cells. Previously, it was reported that MAIT2 cells were absent from CD44$^{hi}$ stage 3 MAIT cells[12]. However, we found that PLZF$^{hi}$ type 2 subsets of MAIT and γδ T cells were relatively enriched at stage 2 compared to type 1 or type 17 cells (Supplementary Fig. 1A). This finding is consistent with a recent report that showed the presence of MAIT cells

expressing PLZF[13]. Tγδ1 and Tγδ17 cells have also been previously described[14] and PLZF$^{hi}$ γδ T cells had been known as γδNKT cells as they express iNKT cell markers[15]. Here, we use the term Tγδ2 cells for consistency. Cytokine profiles of these subsets closely reflected their transcription factor expressions, and type 1, type 2, and type 17 cells secreted IFN-γ, IL-4, and IL-17, respectively, upon activation (Supplementary Fig. 1B, C). As previously showed CD44$^{low}$ and CD44$^{hi}$ NKT2 cells produced IL-4[16], and this feature was also found in MAIT and γδ T cells (Supplementary Fig. 1D). We further analyzed the expression patterns of nine lineage specific markers and performed hierarchical clustering, which grouped together analogous effector subsets of NKT, MAIT, and γδ T cells (Fig. 1b).

In MAIT cells, it was shown that MAIT1 and MAIT17 cells are similar with NKT1 and NKT17 cells respectively[12]. In γδ T cells, TCR Vγ chain usage is known to be highly correlated with their developmental window, pattern of tissue localization, and cytokine secretion[17]. However, each Vγ chain generated multiple Tγδ subsets with age-related variations (Supplementary Fig. 1E, F) and we tested if Tγδ subsets have transcriptional similarities with those of iNKT cells (Fig. 1c–e). For this, we sorted γδ$^{25+}$, Tγδ1, Tγδ2, and Tγδ17 cells using surface markers as in Fig. 1b and performed bulk RNA-seq analysis (Supplementary Fig. 2A–C). For the analysis, we first defined 200–435 differentially expressed genes (DEGs) in each subset, and listed cytokines, receptors, and transcription factors in heat maps (Fig. 1c). When we performed principal component analysis (PCA) on 18,064 genes, the four subsets were distinctly separated by two principal components accounting for 85% of the total variance (Fig. 1d), indicating they are distinct entities. Then, we used a dataset that we previously obtained from iNKT subsets[18] to assess if the analogous effector subsets of iNKT and Tγδ cells (e.g., NKT1 and Tγδ1) had similar transcriptomic features. Specifically, we simulated random events of the overlapping number of DEGs between the iNKT and Tγδ subsets as depicted in Supplementary Fig. 3A and found that the number of genes commonly upregulated or downregulated in side by side comparisons was significantly higher than random expectation (Supplementary Fig. 3B). Using the same algorithm, we compared transcriptional similarities between iNKT cells, γδ T cells, T helper CD4 T cells and ILCs using a published dataset[19,20] and found iNKT and γδ T cells have higher similarity compared to others, which is consistent with our previous report[18] (Supplementary Fig. 3C). Additionally, the overlap of DEGs and the correlation of differential-expression patterns between subsets were significantly stronger for cytokines and receptors as compared to the other genes (Supplementary Fig. 3D, E). Volcano plots and heat maps also showed that genes reportedly associated with lineage differentiation[18] were commonly shared between iNKT and Tγδ subsets (Supplementary Fig. 3F, G). We also defined differentially regulated pathways between iNKT and Tγδ subsets (Supplementary Fig. 4). Finally, PCA on the 120 functional genes using six different subsets from iNKT and γδ T cells showed that their transcriptional profiles were grouped according to their cytokines, receptors, and transcription factors rather than their antigen receptors (Fig. 1e). Collectively, these results indicate that the analogous effector subsets of iNKT and γδ T cells share significant number of genes, particularly for cytokines, receptors, and transcription factors. In addition, we analyzed CD24$^{hi}$ PLZF$^-$ RORγt$^+$ γδ T cells as immature Tγδ17 (Tγδ17i) cells, which were equivalent to CD24$^{hi}$ Vγ4$^+$ or Vγ6$^+$ γδ T cells (nomenclature of Heilig & Tonegawa[21])[11]. These cells were absent in iNKT or MAIT cells (Supplementary Fig. 5A), and using fate mapping reporter mice of *Rorc* and *Il17a*, we showed that RORγt expression is irreversible and only CD24$^{low}$ mature cells produce IL-17 (Supplementary Fig. 5B, C). We also analyzed the signature genes of Tγδ17i cells and found 349 DEGs to be specifically

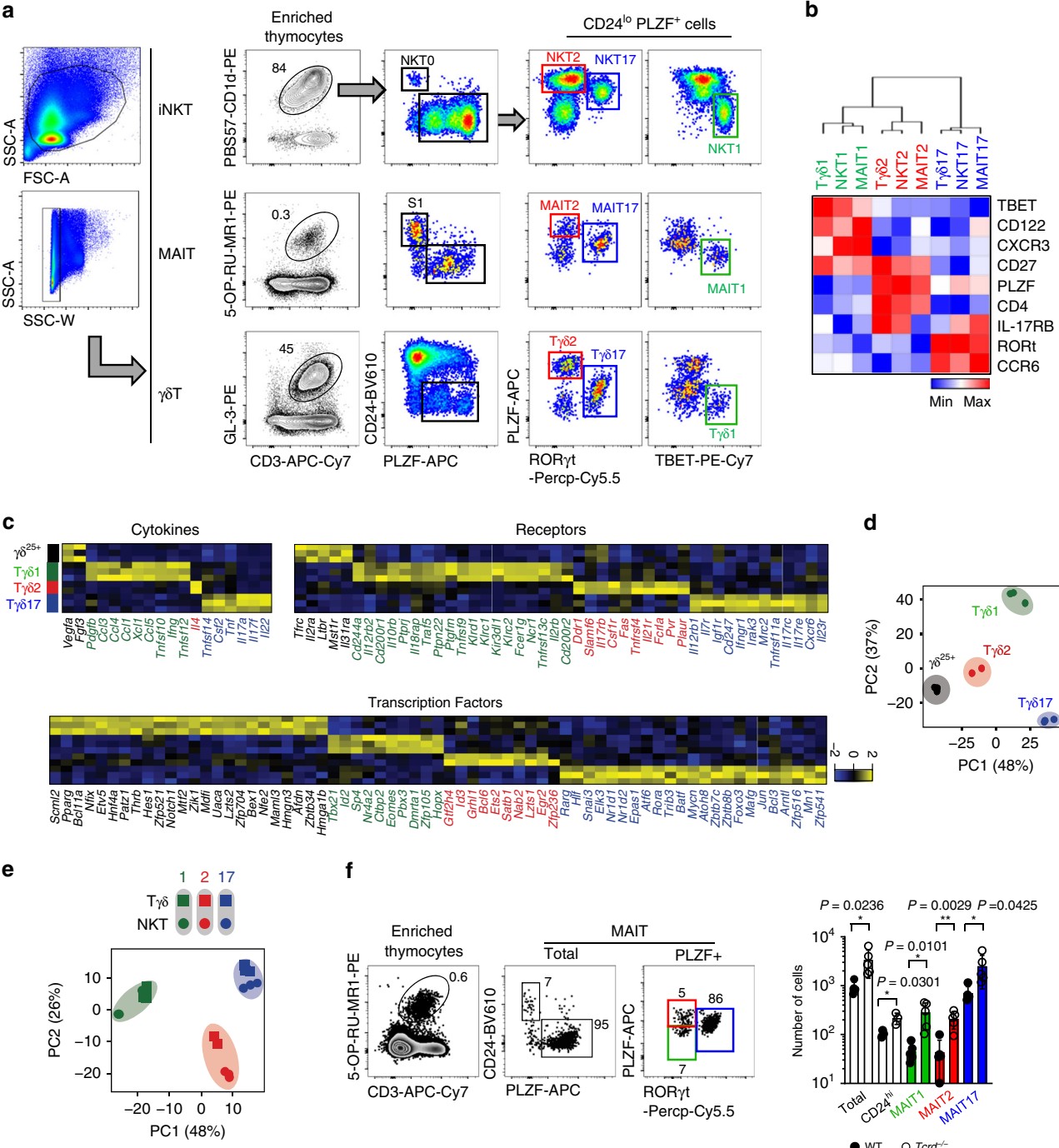

**Fig. 1 Innate T cells have analogous effector subsets. a** Single-cell suspensions of 7-week-old BALB/c thymocytes were stained with PBS57 loaded CD1d tetramer, 5-OP-RU loaded MR1 tetramer, and anti-TCRγδ (GL3), and enriched for iNKT, MAIT, and γδ T cells respectively using MACS beads. Representative dot plots are shown and numbers indicate frequencies of cells in adjacent gates. Representative results of at least 10 independent experiments are shown. **b** Heat map shows $\log_2$ values of mean fluorescence intensities of cells expressing indicated markers analyzed by flow cytometry. Hierarchical clustering was made by Pearson correlation. Representative results of three independent experiments are shown. **c** Heat maps show expression patterns of cytokines, receptors, and transcription factors mapped to overexpressed genes in each Tγδ subset. Expression patterns were quantified by column $Z$-scores of regularized $\log_2$-value of read counts. **d** Principal component analysis (PCA) plot shows subset distribution of Tγδ subsets. Each dot represents a biological replicate. **e** PCA plot using 120 functional genes including cytokines, receptors, and transcription factors shows subset distribution of iNKT and γδ T cells. Each dot represents a biological replicate. **f** MAIT cells were enriched from *Tcrd* KO mice and analyzed for their subset profiles. Graph shows statistical analysis of number of MAIT subsets in indicated mice ($n = 5$ except analysis for CD24hi stage 1 MAIT cells ($n = 3$), right). Results are pooled from three independent experiment and numbers indicate frequencies of cells in adjacent gates (left). Each dot represents an individual mouse. Data are presented as mean ± SD. Unpaired two-tailed *t*-test was used. *$P < 0.05$, **$P < 0.01$. Source data are provided as a Source Data file.

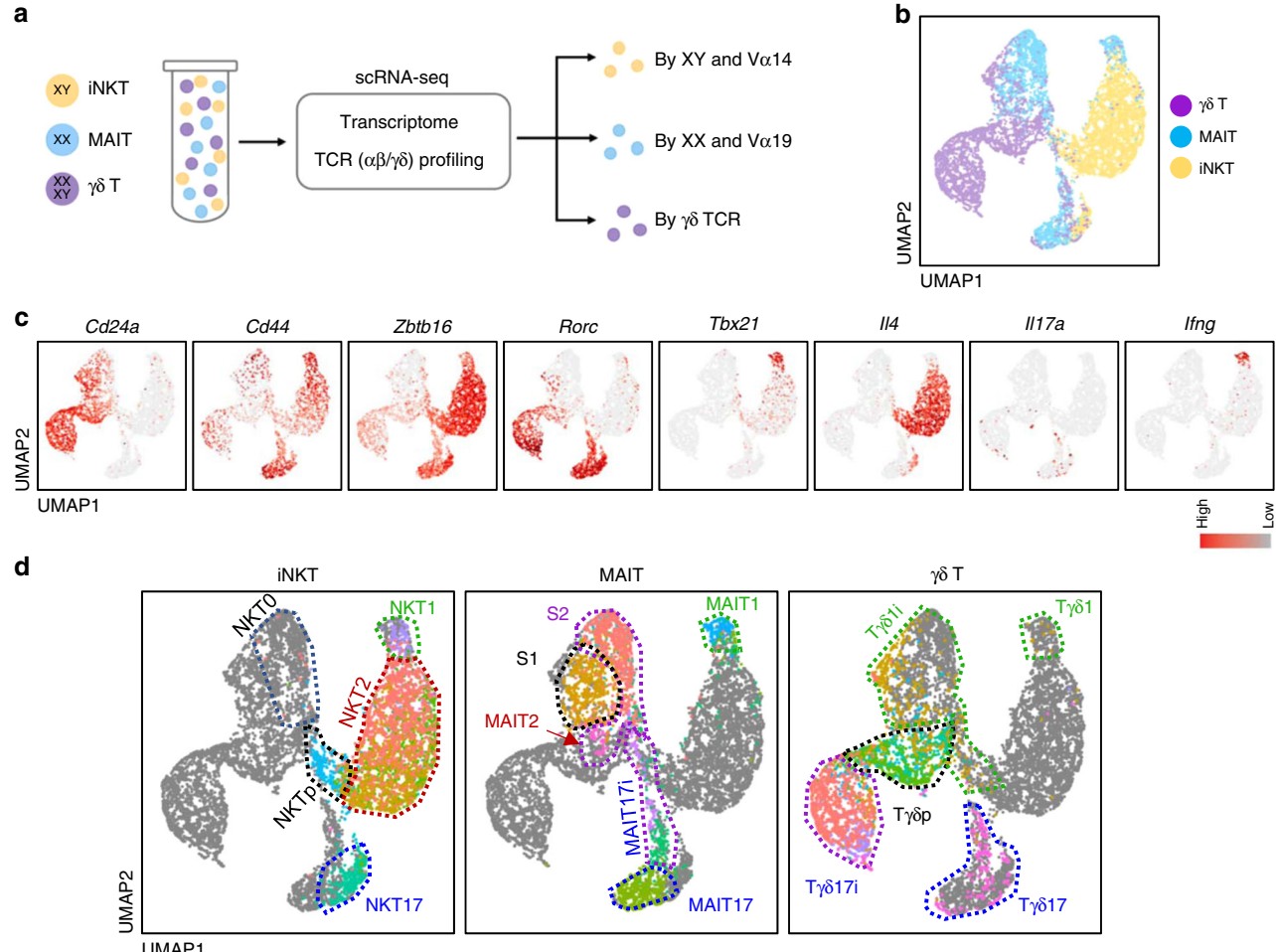

**Fig. 2 scRNA-seq defines developmental intermediates of innate T cells. a** Experimental scheme of scRNA-seq analysis is shown. **b** Uniform manifold approximation and projection (UMAP) plot shows all innate T cells (8239 cells) derived from two pooled replicates. Each cell type was labeled with indicated colors. **c** UMAP plots show the expression levels of indicated marker genes. Colors represent the log$_2$-transformed normalized counts of genes. **d** Combined UMAPs with each type of innate T cell divided into cell clusters (colored): iNKT (3285 cells, left), MAIT (2287 cells, middle), and γδ T cells (2667 cells, right). Cell clusters were annotated as analyzed in Supplementary Figs. 8–10.

upregulated in Tγδ17i cells compared to all the other Tγδ subsets (Supplementary Fig. 5D, E), indicating they are unique developmental intermediates of Tγδ17 cells.

The above similarities suggested that innate T cells might compete for thymic niches with one another; indeed, a previous report showed that MAIT cells expanded in BALB/c $Cd1d^{-/-}$ mice[10]. We additionally found that thymi of $Tcrd^{-/-}$ mice contained three times more MAIT cells (Fig. 1f), indicating that not only NKT cells, but also γδ T cells suppress the development of MAIT cells.

Collectively, these results indicate analogous effector subsets of innate T cells share lineage specific markers, and MAIT cells compete with both iNKT and γδ T cells for their thymic niche.

**scRNA-seq defines developmental steps of innate T cells**. To further analyze developmental pathways of innate T cells at the clonal level in an unbiased manner, we combined scRNA-seq and paired V(D)J sequencing. For this, we sorted total iNKT, MAIT, and γδ T cells from the pooled thymi of BALB/c mice (Supplementary Fig. 2D) and processed two independent pools of cells by mixing equal numbers of the three types of innate T cells in parallel to minimize batch effects. We demultiplexed cell types by combining the sex of mice (male iNKT and female MAIT) with their TCR information (Fig. 2a). From the two

pooled replicates, a total of 8239 cells consisting of 3285 iNKT, 2287 MAIT, and 2667 γδ T cells passed our quality control criteria (Supplementary Table 1 and Supplementary Fig. 6), with an average of 3251 genes and 12,661 unique molecular identifiers (UMIs) per cell.

We used uniform manifold approximation and projection (UMAP) for dimensionality reduction to relate each innate T cell type in a shared low-dimensional representation (Fig. 2b)[22]. Reassuringly, cells from the two pooled replicates were evenly distributed, showing minimal batch effects (Supplementary Fig. 6D). The expression of $Cd24a$, $Cd44$, $Zbtb16$ (encoding PLZF), $Rorc$ (encoding RORγt), $Tbx21$ (encoding TBET), and other markers defined immature populations and effector subsets (Fig. 2c). The annotated cell subtypes were confirmed by examining the signature scores of a subset unique genes of iNKT and γδ T cells that we obtained from our bulk RNA-seq and previous studies[18,23] (Supplementary Fig. 7). To systematically characterize the subpopulation structures, we next applied unsupervised clustering to each type of innate T cell by excluding TCR genes. This TCR-independent transcriptome analysis yielded 22 clusters (Supplementary Figs. 8–10). We manually annotated each cluster type based on the signature scores of subsets and expression of lineage specific markers, and listed cluster-specific upregulated genes (Supplementary data 1).

In iNKT cells, we defined seven clusters, and annotated N1 as NKT progenitor (NKTp) cells, N2 as NKT1 cells, N3–N6 as NKT2 cells, and N7 as NKT17 cells (Supplementary Fig. 8). The signature gene set of CD24$^{hi}$ NKT0 cells had been analyzed before[23], and we detected 11 cells highly expressing them in *Cd24a*$^{hi}$ iNKT cells (Fig. 2d, left, and Supplementary Fig. 7B, top, far left). NKT0 signatures were also highly expressed in *Cd24a*$^{hi}$ MAIT and γδ T cells, indicating that CD24$^{hi}$ immature innate T cells share common transcriptomes (Supplementary Fig. 7B, far left). Likewise, we identified eight clusters within the MAIT cells, and annotated M1 as CD24$^{hi}$ CD44$^{low}$ stage 1, M2–M4 as CD24$^{low}$ CD44$^{low}$ stage 2, and M5 and M6–M8 as CD24$^{low}$ CD44$^{hi}$ stage 3 MAIT1 and MAIT17 cells, respectively, consistent with the three-stage model of MAIT cell development and previous scRNA-seq analysis[10,24] (Supplementary Fig. 9A, B). In γδ T cells, we obtained seven clusters (G1–G7), as annotated in Supplementary Fig. 10A, B. We subdivided G6 into G6-1 and G6-2, according to their usage of TRGV4 and TRGV6, as the latter is known to originate from fetal thymus[25] (Supplementary Fig. 10D, E). Genes including *Pdcd1*, *Cxcr6*, *Zbtb16*, and *Cd44*, distinguished Vγ6$^{+}$ (G6-2) from Vγ4$^{+}$ (G6-1) γδ T cells, and we validated these results by flow cytometry (Supplementary Fig. 10E). G7 cells included a small fraction of *Tbx21*$^{+}$ Tγδ1 cells, and we separated them as G7-2 for mature Tγδ1 and other cells as G7-1 for immature Tγδ1 (Tγδ1i) cells (Supplementary Fig. 10C, F). *Cd122* was highly expressed in the Tγδ1i population before the expression of *Tbx21* or *Cxcr3*. Unfortunately, we were unable to find separate cluster corresponding to Tγδ2 cells. Because we have used only female MAIT cells, we further validated that male and female MAIT cells have no transcriptional difference using previous dataset that used both male and female mice (personal communication with Dr. Lantz, Supplementary Fig. 11)

Overall, the type 1 and 17 effector subsets of iNKT, MAIT, and γδ T cells were clustered together in the UMAP analysis (Fig. 2d), further supporting their developmental similarities and we defined their precursors in scRNA-seq analysis.

**Trajectory analyses predict precursors of MAIT and γδ T cells.** We further analyzed the potential precursor–progeny relationships between subpopulations of MAIT and γδ T cells by deriving a pseudo-temporal ordering of cells along differentiation trajectories using Palantir (Fig. 3) and Monocle 3 (Supplementary Fig. 12). In the MAIT cells, trajectory analysis showed three linear differentiation pathways; MAIT1 cells (M1–M3–M4–M5), MAIT2 cells (M1–M2), and MAIT17 cells (M1–M3–M6–M7–M8; Fig. 3a). Phenotypically, M5 was MAIT1 expressing *Tbx21*, and M8 was MAIT17 expressing *Rorc* and *Ccr6* (Supplementary Fig. 9A, B). M4 was derived from M3 and upregulated type 1 signature genes, such as *Nkg7* and *Ccl5* (Supplementary data 1), indicating they are immature MAIT1 (MAIT1i). M6 and M7 were localized close to NKTp in combined UMAP (Fig. 2d), and they shared their signature genes with NKTp (N1) (Fig. 3c and Supplementary Fig. 13), indicating they are immature MAIT17 (MAIT17i) cells. As M3 is a developmental intermediate of both MAIT1 (M5) and MAIT17 (M8), we designated them as common precursors of MAIT1 and MAIT17 (immature MAIT1/17 or MAIT1/17i). M2 MAIT cells were an immediate progeny of M1 cells that expressed GATA3 and PLZF (Supplementary Fig. 9A) and their phenotype is similar with that of MAIT cells expressing PLZF that identified previously[13]. Although M2 MAIT cells did not co-localize with NKT2 cells in combined UMAP analysis (Fig. 2d, middle panels), they shared their signature genes mainly with NKT2 cells (Fig. 3c and Supplementary Fig. 13), suggesting that M2 corresponds to MAIT2 cells that we identified in flow cytometry (Fig. 1a). However, it requires further

investigations to determine whether MAIT2 cells are terminally differentiated and their developmental relationships with NKTp cells. Overall, these trajectories defined all cells in a three-stage intra-thymic development model of MAIT cells[10], and we newly defined MAIT2 cells and developmental intermediates of MAIT1 and MAIT17 cells.

In the trajectory analysis of γδ T cells, two differentiation pathways were identified: G1–G2/3–G4–G5–G6 for Tγδ17 cells, and G1–G2/G3–G7-1– G7-2 for Tγδ1 cells (Fig. 3b). Based on this trajectory, we annotated G1 as the most immature precursors of γδ T cells (Tγδp), G2 and G3 as common precursors of Tγδ1 and Tγδ17 cells (immature Tγδ1/17 or Tγδ1/17i), G4, and G5 as Tγδ17i cells (Fig. 3b and Supplementary Fig. 10A–C). The signature gene set of γδ$^{25+}$ cells and *Cd25* was rather highly expressed in G2 (Supplementary Fig. 10A–C, H), suggesting Tγδp (G1) cells are earlier precursors than γδ$^{25+}$ cells. Consistent with this, G1 had more diverse TCR genotypes than G2–G7 (Fig. 3d). To validate this finding, we performed fetal thymic organ culture experiment (FTOC) and found the generation of γδ$^{24+25+}$ cells from γδ$^{24+25-}$ cells, indicating γδ$^{25+}$ cells are not the earliest precursors among TCRγδ$^{+}$ cells (Supplementary Fig. 14). Vγ4$^{+}$ cells were a major genotype of G4–G6, whereas Vγ6$^{+}$ cells were only found in G6 as they are fetal-derived remnants (Fig. 3d). Vγ7$^{+}$ cells are abundant in the intraepithelial layer of the small intestine with type 1 phenotype[18] and were enriched in G1, G2/G3, and G7, consistent with the maturation pathways predicted in the trajectory analysis.

Overall, the scRNA-seq analysis predicted that MAIT and γδ T cells have common precursors into type 1/17 lineages, and immature type 1 and type 17 cells at stage 2 or CD24$^{hi}$ cells, respectively.

**Innate T cells are clonally selected during development.** To gain deeper insight into clonal expansion and selection of innate T cells during intra-thymic lineage differentiation, we evaluated the dynamic changes of TCR diversity along trajectories by defining TCR clonotypes as measured by V/J composition (CDR1 and CDR2) and CDR3 sequences (Fig. 4 and Supplementary data 2). For the analysis, we used the terms canonical and oligoclonal to indicate common TCRVα/Jα and TCRVβ usages of iNKT and MAIT cells as in Supplementary Figs. 8E and 9E and first analyzed CDR3 length distribution and sequence variations. CDR3α lengths of canonical TCRα chains of iNKT and MAIT cells were highly uniform and had little sequence variations (Fig. 4a, left and middle), emphasizing the critical role of CDR3α for antigen recognition of iNKT and MAIT cells[26–28]. However, CDR3α lengths of non-canonical TCRs, and CDR3β lengths of both oligoclonal and non-oligoclonal TCRs were relatively diverse with substantial sequence variations. Interestingly, each Vγ chain had a narrow range of CDR3γ length distributions in the order of Vγ6, Vγ4, and Vγ7 with little sequence variations. However, CDR3δ lengths were broadly distributed as previously reported[17], with significant sequence variations (Fig. 4a, right). Therefore, γ and δ TCRs are similar to canonical α and oligoclonal β TCRs of iNKT and MAIT cells respectively in that TCR α/γ had less variation in CDR3 length and sequence compared to TCR β/δ.

Next, we analyzed the clonal repeats of innate T cells and surprisingly found they have highly oligoclonal repeats (Fig. 4b–d and Supplementary Fig. 15 and Supplementary data 2). In iNKT cells, we detected 1898 distinct clonotypes from 2775 cells and found that 86% of clonotypes (or 59% of cells) had a single repeat, and 112 clonotypes (5.9%) were repeated more than three times (Fig. 4b, left). In MAIT cells, we detected 1760 clonotypes from 1892 cells: 95% of clonotypes (or 88% of cells) were single

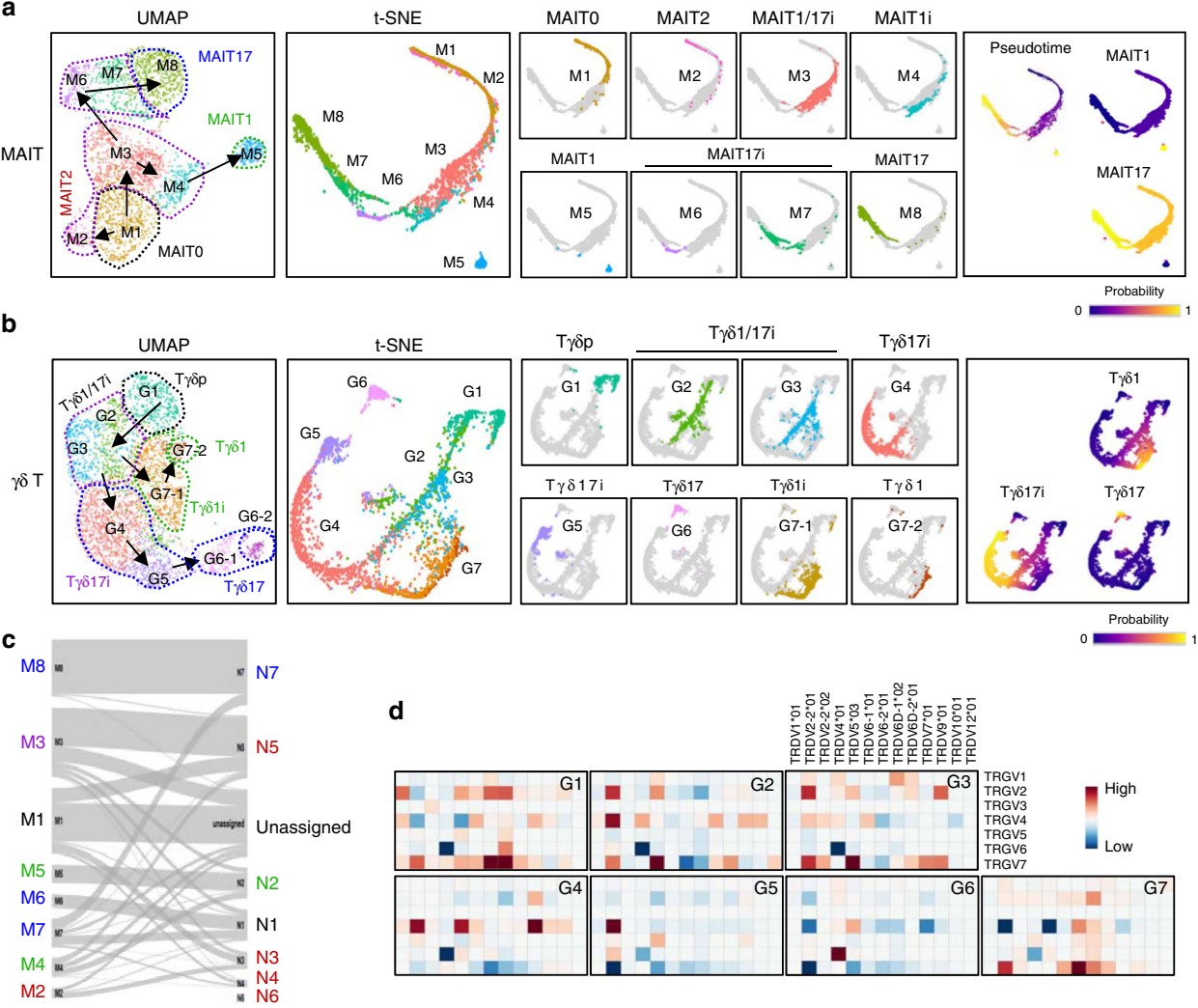

**Fig. 3 Trajectory analysis predicts precursors of MAIT and γδ T cells. a, b** Far left: UMAP plots of MAIT (**a**) and γδ T cells (**b**) show schematic representation of trajectories. Left to far right: t-SNE plots of MAIT (**a**) and γδ T cells (**b**) colored by cell clusters (left), Palantir pseudotime (right), and Palantir branching probabilities (far right). **c** Projections of the MAIT clusters to iNKT clusters by scamp-cluster. **d** Heat maps illustrate log₂-transformed fold change of frequency of each TRGV/TRDV gene pair in a given cell cluster with respect to all γδ T cells.

repeats, and only 0.96% (17 clonotypes) was repeated more than three times (Fig. 4b, middle). In γδ T cells, we detected 1142 clonotypes from 1665 cells and 91% of clonotypes (or 63% of cells) were single repeats (Fig. 4b, right). Although four γδ TCR clonotypes repeated more than 50 times, the pattern of cumulative clonal repeat of γδ T cells was similar to that of iNKT cells, which were less than that of MAIT cells (Fig. 4c and Supplementary Fig. 15). Interestingly, MAIT cells had less clonotypic repeat (Fig. 4b and Supplementary Fig. 15) or overlap between clusters (Supplementary Figs. 8D, 9C, and 10G), compared to NKT and γδ T cells. These features indicate that MAIT cells are less efficient in their clonal proliferation compared to iNKT cells and γδ T cells, consistent with their paucity in the thymus. One possible explanation for such low clonal expansion of MAIT cells is that positively selected MAIT cells go through maturation processes without proliferation. However, this is unlikely as we and others observed a substantial level of Ki-67 expressions at least in stage 2 and 3 MAIT cells[29] (Supplementary Fig. 16A), and cell cycle-regulated genes as well as lineage specific signatures were upregulated during maturation (Supplementary Fig. 17 and Supplementary Table 2). Consequently, the Shannon indexes for TCR diversity of MAIT cells were uniformly high,

unlike NKT and γδ T cells (Fig. 4d). Considering that we collected MAIT cells from a total of 16 mice (two replicates using 8 mice each), this result indicates that individual mice had almost no overlap in their clonality. It is also unlikely that each mouse had unique clonal repeat, because when we separated two biological replicates of MAIT cells (eight mice each), there was more reduction in their clonal repeat (Supplementary Fig. 16B). In γδ T cells, the Shannon indexes for TCR clonotypes were the highest in Tγδp (G1, 0.98) cells, and lower in Tγδ17i (G4 and G5, 0.92) cells, and further decreased in Vγ4⁺ Tγδ17 (G6-1, 0.73) cells (Fig. 4d, right), indicating that limited Tγδ17 clonotypes are selected after positive selection. As Vγ6⁺ Tγδ17 cells are mostly fetal derived, which do not have a junctional diversity, their Shannon index was lower than that of Vγ4⁺ Tγδ17 cells (0.34 vs. 0.73). We detected 21 clonotypes present both in Tγδ1i and Tγδ17i cells, which repeated total 359 times and occupied 21% of total cells and 58% of repetitive clones (Supplementary data 2). Figure 4e shows the distribution of most repetitive clonotype, which repeated 141 times in the tSNE plot. This feature indicates that identical TCR clonotype can generate diverse functional lineages although there are strong bias for their TCR usage in each Tγδ subset.

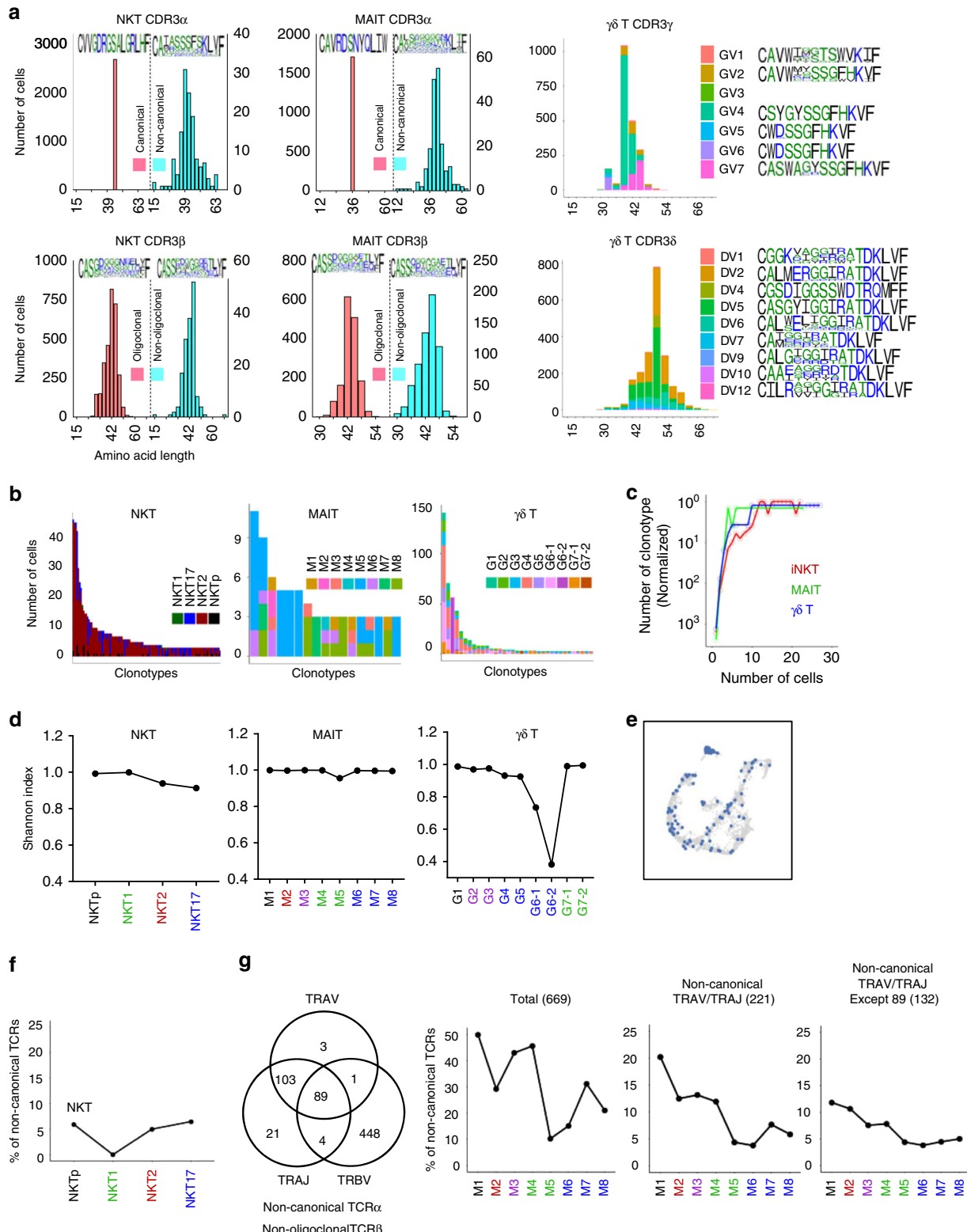

Next, we analyzed the non-canonical TCRα and/or non-oligoclonal TCRβ usage of iNKT and MAIT cells. In iNKT cells, 204 out of 1898 clonotypes (10.7%) or 253 out of 2775 cells (9.1%) had at least one of the non-canonical TCR Vα/Jα and/or non-oligoclonal TCR Vβ chains (Fig. 4f, left and Supplementary Figs. 8E−G). The most frequent non-canonical TCR of iNKT cells was Vα10/Jα50, which was previously reported in *Jα18* KO BALB/c mice[30] (Supplementary Fig. 8F). We detected three different

clonotypes of them, which were all NKT2 cells and one of them repeated 15 times, accounting for a total of 18 cells out of 1898 analyzed iNKT clonotypes (Supplementary Table 1 and Supplementary data 2). In MAIT cells, 1107 out of 1760 (69%) analyzed clonotypes had canonical TCRs (Supplementary Fig. 9D–E). Interestingly, MAIT cells having one or more non-canonical TCRα and/or non-oligoclonal TCR Vβ chains were ~50% in M1, but decreased dramatically in MAIT1 (M5, 10%), MAIT2 (M2,

**Fig. 4 Innate T cells are clonally selected during development. a** Graphs show the distribution of CDR3α/γ (top) and CDR3β/δ (bottom) lengths of iNKT, MAIT, and γδ T cells, presented as the number of cells for each length. Canonical (top left, orange) and non-canonical (top right, turquoise) TCRα are shown together in iNKT and MAIT cells. The relative amino acid composition is shown for the most common length by using the WebLogo application (hydrophilic, blue; neutral, green; and hydrophobic, black). **b** Bar plots show the ordered number of cells for each clonotype repeated 3 or more in iNKT (left), MAIT (middle), and γδ T cells (right), colored by each subset (iNKT) or their UMAP regions (MAIT and γδ T). Each bar represents an individual clonotype from Supplementary data 2. **c** Normalized number of clonotypes (*y*-axis) within each type of innate T cells (colored lines) plotted over the number of cells having the identical clonotype. **d** Line plots show the Shannon equitability indexes of clonotypes for the indicated subset of cell cluster in iNKT (left), MAIT (middle), and γδ T cells (right). **e** t-SNE plot of γδ T cells colored by cells having the most abundant single clonotype from Supplementary data 2. **f** Graph shows percentage of cells having non-canonical TCRα subchain for each subset of iNKT. **g** Venn diagram shows the non-canonical TCRα and/or non-oligoclonal TCRβ usage of MAIT cells. Numbers indicate number of cells with each combination. Graphs show distribution of total cells or MAITs with non-canonical TCRα or non-canonical TCRα except 89 cells in Venn diagram in each MAIT cluster. Numbers in parentheses indicate total number of cells analyzed in each graph. Source data are provided as a Source Data file.

28%), and MAIT17 (M8, 21%) cells (Fig. 4g). It is extremely unlikely that these cells are contaminated cells as we have seen only 89 cells out of 669 cells that had non-canonical TCRα paired with non-oligoclonal TCRβ of MAIT cells, and we obtained same trends when we excluded these 89 cells (Fig. 4g). We also observed similar trends when we analyzed all different combinations of non-canonical TCRα and/or non-oligoclonal TCRβ usages (Supplementary Fig. 18A). To further rule out the possibility of cell contamination, we analyzed signature gene expression patterns between MAIT cells with canonical and non-canonical TCRα in each cluster using defined gene sets produced from previous research[29] and found that there were few DEG between them except their TCRs (Supplementary Fig. 18B). In contrast, we found 55–430 DEGs between canonical MAITs and CD4+CD8+ DP thymocytes that we excluded from analysis as in Supplementary Fig. 6A (Supplementary Fig. 18D). Although our results showed the overall frequencies of non-canonical TCRs were two to three-times higher than those of previous reports[10,31], it might be because we used BALB/c mice instead of B6 mice. Therefore, it is conceivable that the semi-invariant nature of MAIT cells is not a result of positive selection of DP thymocytes but a consequence of clonal selection after stage 2. It is unlikely that this reflects a different rate of clonal expansion as we observed almost non-overlapping clonotypes in MAIT cells (Fig. 4b–d). These features were not analyzable in the iNKT cells, however, as we could detect rare CD24hi NKT0 cells (11 out of 2775 cells) and TCR sequences were detected in four cells of them, three of which had canonical TCRs (Supplementary data 2).

Collectively, our results indicate that positive selection repertoires of MAIT and γδ T cells are more diverse than their progenies and that the canonical MAIT TCRs are selected during their maturation process.

**Lineage differentiation into Tγδ17 cells is flexible.** Our analysis of γδ T cells using bulk and scRNA-seq indicates that their development could be explained by the lineage differentiation process rather than instruction by TCRs, and we further validated these results by in vivo experiments. scRNA-seq predicted the presence of Tγδ1i cells expressing *Cd122* but not *Tbx21*, and we found CD24hi CD122+ cells that expressed TBET upon downregulation of CD24 (Fig. 5a, gating 1). Similarly, RORγt+ γδ T cells upregulated CCR6 upon CD24 downregulation (Fig. 5a, gating 2). We further validated differentiation of γδ T subsets by FTOC experiments (Fig. 5b), in which we isolated four subsets (a–d) according to expression patterns of CD24, and tdTomato in *Rorc*Cre Rosa26 LSL tdTomato mice. After 5–7 days of FTOC, we found CD24hi tdTomato− cells (a) generated all the other populations, including tdTomato+ cells. Isolated Tγδ17i (c) cells uniformly generated Tγδ17 cells and once generated Tγδ17 cells (d) did not change their phenotype. In addition, CD24low

RORγt− cells (b) did not generate Tγδ17 cells, indicating all Tγδ17 are terminally differentiated and exclusively derived from Tγδ17i population. These results indicate lineage differentiation between Tγδ1 and Tγδ17 cells occur at CD24hi stage, consistent with the finding that RORγt expression is irreversible (Supplementary Fig. 5B).

Lineage differentiation of γδ T cells is highly linked to their TCR usage. In particular, Tγδ17 cells were mainly composed of Vγ4+ or Vγ6+ cells. Therefore, we further experimentally addressed whether TCR alone directs lineage fate determination of Tγδ17 cells by analyzing Vγ4/6 and *Rorc* KO mice. Surprisingly, the number of Tγδ17 cells was not decreased in Vγ4/6 KO mice, and Vγ1+ (Vδ 6.3+ or Vδ 6.3−) cells replaced Tγδ17 lineages (Fig. 5c). In contrast, *Rorc* KO mice, in which Tγδ17 differentiation is blocked, had a reciprocal expansion of Tγδ1 cells (Fig. 5d). In this mouse, the number of immature and mature Vγ4+ cells were not decreased, and Vγ4+ Tγδ1 cells increased 2.5-fold (Fig. 5e). Because small number of Vγ1+ Tγδ17 cells already present in WT mice (Fig. 5c, upper panels), it raises the issue whether the expansion of Vγ1+ Tγδ17 cells represents a simple niche filling or re-direction of their fate at the progenitor stage. We found some evidence supporting the latter case. First, there was no proportional expansion of other minor Tγδ17 cells (Vγ5 and Vγ7, Fig. 5c, pie chart) and, second, Ki-67 levels of Vγ1+Vδ6.3+ cells were not increased in Vγ4/6 KO mice (Supplementary Fig. 19A). Third, the frequency of Vγ1+Vδ6.3+ cells among Tγδ2 cells was decreased in Vγ4/6 KO mice (Supplementary Fig. 19B), suggesting some of them redirected their fate from Tγδ2 cells to Tγδ17 cells at the progenitor stage. Although these are indirect evidence, above findings collectively indicate that, rather than TCR Vγ instructing the lineage fate of γδ T cells, there is a plasticity in their fate decision.

**γδ T and MAIT cells develop via lineage differentiation.** We showed that RORγt deficiency redirected the fate of Vγ4+ cells from Tγδ17 to Tγδ1 cells (Fig. 5). To further address the issue of lineage plasticity in the other subsets of γδ T cells, we depleted Tγδ2 and Tγδ1 cells using *Cd4*cre *Gata3*f/f (*Gata3* cKO) and *Tbx21* KO mice, respectively (Fig. 6). γδ T cells are mostly double negative, and CD4 expression was limited to Tγδ2 cells (Fig. 1b) and they were efficiently depleted in *Gata3* cKO mice (Fig. 6a). In these mice, there were reciprocally increased frequencies of Tγδ1 and Tγδ17 cells, indicating that abortive differentiation into Tγδ2 cells induced the expansion of Tγδ1 or Tγδ17 cells. In *Gata3* cKO mice, we compared the frequencies, instead of numbers, of each Tγδ subset because they had much smaller thymi due to the reduced number of DP and SP thymocytes as previously known[32]. As DP thymocytes are required for trans-conditioning of γδ T cells[33] and the frequency of γδ T cells were not different between *Gata3* cKO and WT mice, we think it is reasonable to

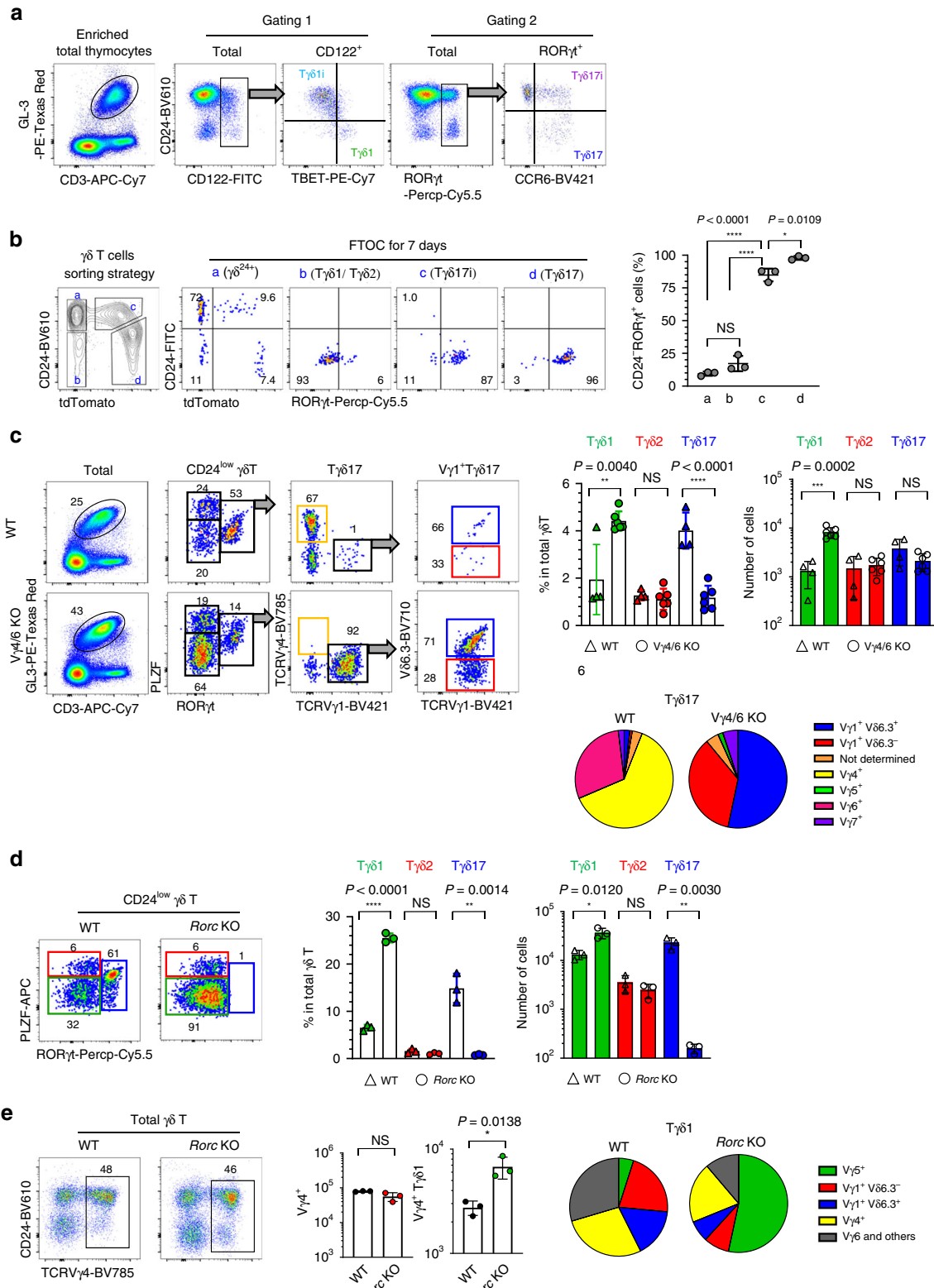

compare their relative frequencies, rather than absolute numbers. This result is consistent with the phenotype of iNKT cells in *Gata3* cKO mice, in which NKT1 cells expanded in the absence of NKT2 and NKT17 cells[34] as GATA3 was highly expressed in both NKT2 and NKT17 cells[8]. However, the development of MAIT cells was arrested at stage 1 in *Gata3* cKO mice, indicating GATA3 is essential for their early development (Fig. 6b).

In TBET-deficient mice, we previously showed that the development of NKT2 and NKT17 cells were reciprocally expanded in the absence of NKT1 cells[8]. Consistent with this result, we observed the expansion of both Tγδ2/17 and MAIT2/17 cells in the absence of TBET (Fig. 6c, d). In RORγt deficient mice, we found there are expansion of Tγδ1 cells (Fig. 5d) as Tγδ17i cells expressed RORγt (Supplementary Fig. 5A) and their

**Fig. 5 Lineage differentiation into Tγδ17 cells is flexible. a** Single-cell suspensions of BALB/c thymocytes were enriched for γδ T cells using MACS beads and analyzed for the expression of indicated markers. Representative data of three independent experiments are shown. **b** Four different subsets of γδ T cells were sorted as indicated (left) and performed fetal thymic organ culture (FTOC) for 7 days (middle). Graph shows the frequencies of CD24⁻RORγt⁺ cells after culture for 7 days (right, $n = 3$). Results from three independent sets of experiments are shown. **c–e** Thymi from adult B6 WT ($n = 4$) and Vγ4/6⁻ᐟ⁻ ($n = 6$) (**c**) or 7-day-old BALB/c WT ($n = 3$) and BALB/c $Rorc$⁻ᐟ⁻ ($Rorc$^EGFP/EGFP, $n = 3$) (**d, e**) mice were analyzed for the development of indicated γδ T subsets using flow cytometry. Representative dot plots are shown and graph shows statistical comparisons. Results are from two independent sets of experiments. Pie charts show mean frequencies of each subset among Tγδ17 cells (**c**) from adult B6 WT ($n = 4$) and Vγ4/6⁻ᐟ⁻ ($n = 5$) and Tγδ1 (**d**) cells from BALB/c WT ($n = 3$) and 7 day-old BALB/c $Rorc$⁻ᐟ⁻ ($n = 3$). Numbers indicate frequencies of cells in adjacent gates. Data are presented as mean ± SD. Unpaired two-tailed $t$-test was used. *$P < 0.05$, **$P < 0.01$, ***$P < 0.001$, ****$P < 0.0001$, NS not significant ($P > 0.05$). Source data are provided as a Source Data file.

absence would directly block the differentiation of type 17 lineages from the progenitors (Fig. 3b, G1–3). These features, however, could not be analyzed in iNKT and MAIT cells as they are dependent on RORγt for their TCR rearrangement[35].

Collectively, the above results provide in vivo evidence supporting the lineage differentiation pathways of γδ T cells predicted from the scRNA-seq analysis.

**Human and mouse innate T cells have analogous subsets**. Finally, we tested human innate T cells to determine if they also have common subset composition. For the experiments, we used liver perfusion fluids, which contain more abundant innate T cells than the peripheral blood (Fig. 7). Among all the CD3⁺ T cells, we could detect iNKT, MAIT, and γδ T cells using CD1d and MR1 tetramers and anti-TCRγδ antibody, as shown in Fig. 7a. We distinguished two distinct subpopulations of iNKT, MAIT, and γδ T cells, which were RORγt⁺ (TBET^int) and TBET^hi (RORγt⁻) cells. The former was a major population of iNKT and MAIT cells, whereas, the latter was more dominant in γδ T cells. We further analyzed γδ TCR usage and found that Vδ1⁺Vγ9⁻ cells were enriched in the TBET^hi population, whereas Vδ2⁺Vγ9⁺ cells were enriched in RORγt⁺ γδ T cells (Fig. 7b). In cytokine analysis, RORγt⁺ NKT, MAIT, and γδ T cells produced both IFN-γ and IL-17A, whereas TBET^hi cells produced only IFN-γ but not IL-17A (Fig. 7c). IL-4 production was detected in TBET^hi iNKT cells, but not in MAIT or γδ T cells, which is similar to mouse NKT1 cells that produced IL-4 upon activation but not at the steady state[8]. Overall, these features show that human innate T cells share analogous effector subsets with each other, despite being different from those in mice, in that human RORγt⁺ cells simultaneously express an intermediate level of TBET with RORγt.

**Discussion**
In this study, we showed that iNKT, MAIT, and γδ T cells have analogous effector subsets and they not only compete for thymic niches, but also exhibit great similarity in their transcriptional nature at the single-cell level. Previously, we had shown that the development of iNKT cells can be explained by the lineage differentiation model and NKT2 cells are terminally differentiated[8]. We additionally tested NKT1 and NKT17 cells and found that they are also terminally differentiated cells (Supplementary Fig. 20). Previous literature showed that MAIT1 and MAIT17 cells are similar with NKT1 and NKT17 cells respectively[12,36] and we further extended our research scope to find a general rule to explain the development of innate T cells. In γδ T cells, the absence of TBET (*Tbx21* KO), GATA3 (CD4^Cre Gata3^f/f) and RORγt (*Rorc* KO) specifically blocked the differentiation of Tγδ1, Tγδ2, and Tγδ17 cells respectively, and there were reciprocal expansions of the others (Figs. 5 and 6). On the other hand, TCR Vγ4/6 deficiency did not abrogate the development of Tγδ17 cells (Fig. 5c) and identical clonotypes can differentiate into both

Tγδ17 and Tγδ1 lineages (Fig. 4e). In MAIT cells, we also observed the expansion of MAIT2 and MAIT17 cells in the absence of TBET, but GATA3 and RORγt deficiency could not be tested due to the complete absence of MAIT cells. Overall, these results indicate that effector subsets of innate T cells develop via lineage differentiation process rather than TCR-mediated instruction or linear maturation.

We and others showed the expansion of MAIT cells in the absence of NKT or γδ T cells[10,37] and interpreted this result that innate T cells compete for their developmental niches. Interestingly, previous report suggested that neomycin cassette affects *Tcra* gene rearrangements as seen in *Traj18* KO mice[38]. Because *Tcrd* KO mice were also generated by using this, there is a possibility that the expansion of MAIT cells is an artificial effect. However, *Tcrd* recombination generally facilitates MAIT cell development by enhancing diverse *Tcra* gene rearrangements[39] and we observed no effects of NKT cell frequencies in *Tcrd* KO mice. Further investigations are required to rule of this possibility.

In this study, we newly defined MAIT2 cells that correspond to NKT2 or Tγδ2 cells (Fig. 1a). Unlike NKT2 or Tγδ2 cells, however, MAIT2 cells did not express IL17RB (Fig. 1b), were less efficient for IL-4 production (Supplementary Fig. 1B, C) and were not co-localized with NKT2 cells in UMAP (Fig. 2d). Previously, we showed that PLZF^hi NKT cells are subdivided as IL17RB-positive IL-4-producing NKT2 cells and IL17RB -negative IL-4-non-producing NKT progenitors (NKTp)[18], which defined as N1 cluster in an unbiased clustering (Supplementary Fig. 8A). Based on this, it is possible that PLZF^hi MAIT cells could correspond to NKTp cells rather than NKT2 cells. However, M2 (MAIT2) had similarity with N3 (NKT2) and N1 (NKTp) was more likely M6 (MAIT17i) in their transcriptional nature (Fig. 3c and Supplementary Fig. 13). It is possible that MAIT2 cells are not fully differentiated IL-4 producing subset and their developmental nature is in between NKTp and NKT2 cells, which requires further investigation for the analysis of their exact ontogeny.

There are strong correlations between the types of γδ TCRs and their lineage fates. In this study, however, we showed that the absence of certain TCRs (e.g., Vγ4/6) or lineage specific transcription factors (TBET, RORγt, and GATA3) can re-direct lineage fates of γδ T cells. Therefore, γδ TCRs seem to be one of the factors, rather than a single determinant, that direct the lineage differentiation of γδ T cells, which recognize thymic self-antigens and provide certain signaling threshold. Consistent with this idea, a previous report showed lineage conversion of Vγ5⁺ dendritic epidermal T cells (DETCs) from Tγδ17 cells into IFN-γ secreting Tγδ1 cells in the absence of Skint1[40]. Previously, Kang and colleagues[15,41] elegantly showed that cell intrinsic program can pre-determine the lineage fate of Tγδ17 cells at the DN1d stage, which express *Rorc* and *Sox13* before they express γδ TCRs. Consistent with this, we also found *Rorc*⁺*Ccr6*⁺*Sox13*⁺ cells present in most immature G1 cluster (Supplementary Fig. 10C). However, this report supports the idea that lineage fate of γδ

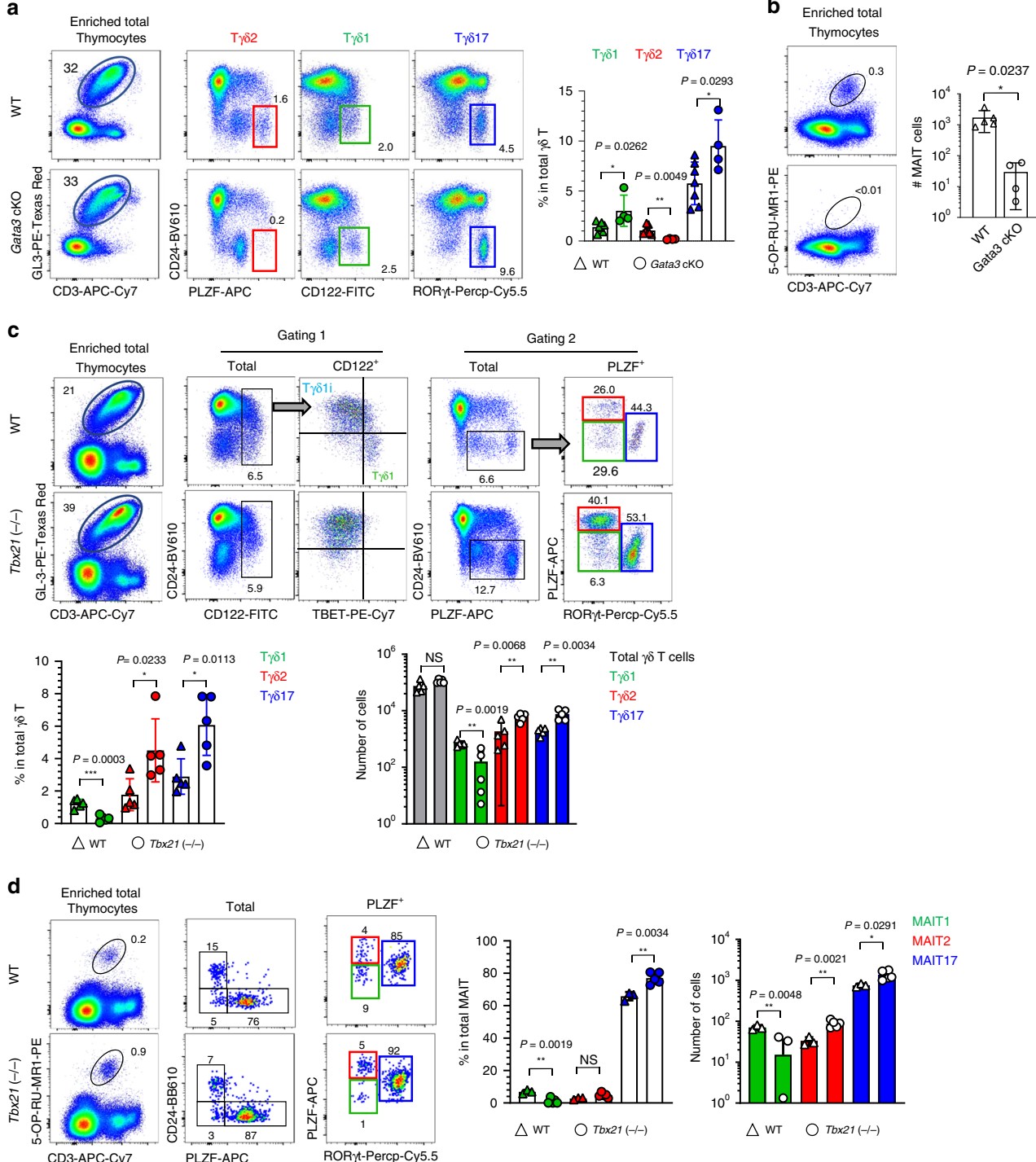

**Fig. 6 γδ T and MAIT cells develop via lineage differentiation process. a** Thymocytes from *Cd4*^Cre (WT, n = 7) or *Cd4*^Cre *Gata3*^f/f (conditional KO (cKO), n = 4) mice were enriched for γδ T cells using MACS beads and analyzed for their subset frequencies amongst total γδ T cells (left). Graph shows statistical analysis (right). Representative data of two independent sets of experiment are shown. **b** MAIT cells were enriched from total thymocytes of WT (n = 5) or *Gata3* cKO (n = 4) mice and compared for their absolute numbers. Representative dot plots (left) and statistical comparison is shown (right). Representative results from two independent sets of experiments are shown. **c, d** Single-cell suspensions of thymocytes from WT (n = 5) or TBET-deficient mice (n = 5) were enriched for γδ T (**c**) and MAIT (**d**) cells using MACS beads and analyzed for their subset development. Representative dot plots are from three independent experiments (left) and graph shows statistical analysis of three independent sets of experiment (right). Numbers indicate frequencies of cells in adjacent gates. Data are presented as mean ± SD. Unpaired two-tailed *t*-test was used. *P < 0.05, **P < 0.01, ***P < 0.001. NS not significant (P > 0.05). Source data are provided as a Source Data file.

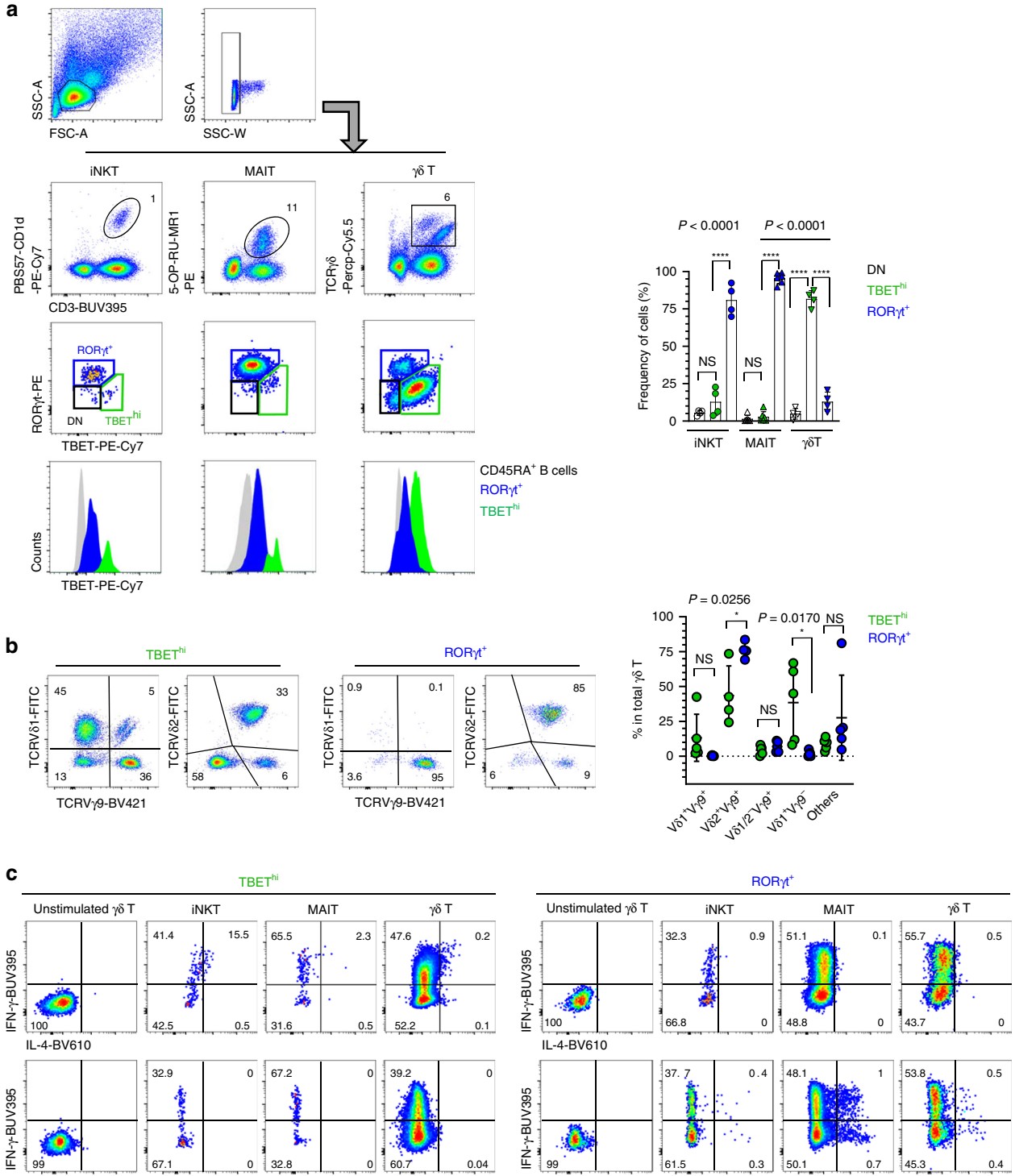

**Fig. 7 Human and mouse innate T cells have analogous subsets. a** Human mononuclear cells obtained from liver perfusion fluid were stained with the indicated markers. Representative dot plots (left) show frequencies of iNKT (left), MAIT (middle), and γδ T cells (right) amongst total mononuclear cells. Representative FACS plots are from seven independent experiments and graph shows statistical analysis of their frequencies using pooled data ($n = 4$ for NKT cell analysis, $n = 6$ for MAIT cell analysis, $n = 4$ for γδ T cell analysis). **b** Total γδ T cells from (**a**) were stained with the indicated anti-TCR antibodies and representative FACS plots are from three independent experiments (left) and graph shows pooled results of their frequencies with statistical analysis (right, $n = 5$ except Vδ2+Vγ9+ cell analysis ($n = 4$)). **c** Indicated cells were stimulated with PMA and ionomycin, and intracellularly stained with the anti-cytokine antibodies. Representative dot plots are shown from three independent experiments. Numbers indicated frequencies of cells in adjacent gates or each quadrant. Data are presented as mean ± SD. Unpaired two-tailed $t$-test was used. *$P < 0.05$, **$P < 0.01$, ***$P < 0.001$, ****$P < 0.0001$. NS not significant ($P > 0.05$). Source data are provided as a Source Data file.

T cells are determined independent of their TCRs, and the presence of Soxpro cells is not compatible with our results that showed lineage fate of Tγδ17 cells are determined at CD24[hi] immature stage. In our FTOC experiment, we showed CD24[+] *Rorc*[−] cells generated *Rorc*[+] population (Fig. 5b), indicating RORγt is inducible after γδ TCR expression. One possible explanation for this discrepancy is that there are multiple pathways for the Tγδ17 generations; one is predetermined at DN1d stage and the other one is programed during their development by lineage differentiation, which requires further investigation.

γδ T cells mostly develop independently of MHCs, and the structure of their TCRs shares more resemblance to that of immunoglobulin than the αβ TCRs[17]. Instead of MHCs, thymic epithelial expression of immunoglobulin superfamily molecules, such as Skint1 and Btnl, directed the development and local recruitment of Vγ5[+] DETCs and intestinal Vγ7[+] γδ T cells, respectively[40,42,43]. Interestingly, a recent report showed that Btnl recognition of Vγ7[+] γδ T cells was mediated by a germline-encoded motif of Vγ chains, whereas antigen recognition was mediated by CDR3s generated during the TCR rearrangement, indicating that γδ T cells use a different part of their TCRs for positive selection and activation[44]. This is consistent with structural analysis of T10/22-restricted γδ TCRs, which showed that TCRδ mainly recognized the antigens[45]. Based on these findings, we speculate that positive selection of γδ T cells is mediated by TCRγ chains recognizing endogenous self-antigens and, for their final maturation, they need to recognize additional antigens by their TCRδ chains. Consistent with this, we showed that the diversity of Tγδp (G1) cells is very high (Shannon index, 0.98), which decreases as they mature into the Tγδ17 lineage (Fig. 4d, right). This trend was not observed in Tγδ1 cells, probably because Vγ7 cells leave the thymus at the CD24[hi] stage, and we did not include many Tγδ1 cells for analysis. Currently, we are preparing to analyze the peripheral repertoire of γδ T cells, especially in the gut, to compare it with the thymic one.

For the thymic development of iNKT cells, homotypic interactions between immature thymocytes with SLAM-SAP signaling are critical[46]. For their effector differentiation into NKT1, NKT2, and NKT17 cells, various combination of cytokine and transcription factors are also required. These features are also similarly conserved in MAIT cells, but the specific requirement of each factors for the development of γδ T cells have not been clearly defined. In the thymus, not all CD24[low] γδ T cells express PLZF, and there are naïve-like γδ T cells especially in the periphery, which do not express activation markers or lineage specific transcription factors. As iNKT cells are long resident population in the thymus, it is possible that thymic γδ T cells are enriched with innate-like population, whereas naïve-like populations leave the thymus early. Further investigations are required to define factors conditioning innate versus naïve-like γδ T cells.

The TCR clonotypic analysis of innate T cells is reminiscent of limited mice in which fixed TCR Vα3.2 and Vβ5 transgene is paired with two Jα mini-locus and generate diverse CDR3s[47]. In this analysis, they showed that the TCR repertoire of conventional T cells is highly diverse at the pre-selection DP stage, but post-selection thymocytes and mature peripheral T cells have an overlapping but distinct bumpy TCR repertoire. In MAIT cells, we observed that up to 50% of MAIT0 (M1) cells have non-canonical TCRα and/or non-oligoclonal TCRβ, whereas cells with canonical TCRs are predominantly mature subsets (Fig. 4g). It is possible that the MAIT cells should recognize additional ligands for their final maturation that are more specific to the canonical TCRs. Germ-free mice are more deficient for mature stage 3 MAIT cells than stage 1 or 2 immature ones[10], and it is possible that exogenous ligands provided by intestinal bacteria would favor canonical TCRs. This result suggests that positive selection

of MAIT cells is dependent on endogenous self-ligands, which are more permissive to non-canonical TCRs, but their final maturation requires exogenous antigens. In this perspective, the microbial difference between those of humans and SPF mice would explain the abundance of MAIT cells in humans.

Overall, our results show that the effector differentiation of innate T cells is closely shared and regulated by clonal selection, proliferation, and competition. Recent reports showed the critical role of innate T cells in the pathogenesis of human disease and there have been attempts to use them for immunotherapeutic purpose[48,49]. In that regard, our results would be important for understanding the various functional aspects of innate T cells and their potential for use in immunotherapeutic settings.

# Methods

**Mice.** B6 (C57BL/6J, Stock# 000664), BALB/cJ (Stock# 000651), BALB/cByJ (Stock# 001026), *Tcrd*[−/−] B6 (B6.129P2-*Tcrd*[tm1Mom]/J, Stock# 002120), *Tbx21*[−/−] B6 (B6.129S6-*Tbx21*[tm1Glm]/J, Stock# 004648), B6 *Rorc*[cre] (B6.FVB-Tg(Rorc-cre) 1Litt/J, Stock# 022791), B6 *Il17a*[cre], B6.Cg-*Gt(ROSA)26Sor*[tm14(CAG-tdTomato)Hze]/J (Stock# 007914), and B6 Rorc(γt)[EGFP] (B6.129P2 (Cg)-Rorc [tm2Litt] /J, Stock# 007572) mice were from the Jackson laboratory. BALB/cAnNCrl mice were purchased from Charles River. KN2 and *Tbx21*[gfp] reporter mice were previously described[8] and B6.Cg-*Gt(ROSA)26Sor*[tm6(CAG-ZsGreen1)Hze]/J mice were received from Dr. Charles D. Surh (POSTECH, Korea). Vγ4/6 KO mice were kindly provided by Dr. Rebecca O'Brien (National Jewish Health, USA) under the permission from Koichi Ikuta (Kyoto University, Japan). BALB/c Rorc(γt)[EGFP] mice were generated by backcrossing B6 Rorc(γt)[EGFP] mice at least five generations into BALB/cJ mice. *Cd4*[Cre] *Gata3*[f/f] were obtained from Dr. Sin-Hyeog Im (POSTECH, Korea). All mice were used at the age of 6–12 weeks unless indicated and age- and sex-matched animals were used as controls. In experiments analyzing *Tbx21*[−/−] B6 or *Cd4*[Cre] *Gata3*[f/f] mice, littermate controls were bred in same cages. In experiments analyzing *Tcrd*[−/−] B6, Vγ4/6 KO, or BALB/c Rorc(γt)[EGFP] mice, WT control mice were bred separately. Both female and male mice were used in experiments. Euthanasia was performed by carbon dioxide inhalation. Mouse care and experimental procedures were performed in accordance with all institutional guidelines for the ethical use of non-human animals in research protocols approved by the Institutional Animal Care and Use Committees (IACUC) of the Pohang University of Science and Technology (POSTECH). All animals were bred in a specific pathogen-free (SPF) conditions, ambient temperature 23 ± 1 °C, humidity 50 ± 10%, and a dark/light cycle of 12 h.

**Human samples.** Human liver perfusates was obtained from healthy living liver transplant donors who were hepatitis B virus (HBV) DNA, hepatitis C virus (HCV), and anti-human immunodeficiency virus (HIV) antibody negative. Graft livers were perfused with Custodiol HTK (Essential Pharmaceuticals) solution during the bench procedure in the setting of living donor liver transplantation. Of the 1000 ml of total perfusate, the first 500 ml was discarded and the second 500 ml collected and filtered. Peripheral blood mononuclear cells (PBMCs) and liver sinusoidal mononuclear cells (LSMCs) were isolated by density gradient centrifugation using Ficoll-Paque (GE Healthcare Life Science). This study was reviewed and approved by the institutional review board of Severance Hospital (Seoul, Republic of Korea; 2013-1071-001) and conducted according to the principles of the Declaration of Helsinki. Informed consent was obtained from all study participants.

**Bulk cell isolation and RNA preparation.** For bulk cell RNAseq, single-cell suspensions of day 5 thymi of BALB/cByJ mice were stained with anti-TCRγδ (GL3) and other surface markers and sorted using a Moflo-XDP (Beckman Coulter). TRIzol (Life Technologies)-chloroform (Sigma) extraction protocol was used to isolate RNA obtained from each sample.

**Bulk RNA sequencing and data analysis.** Bulk RNA sequencing of γδ T was done at Macrogen (www.macrogen.co.kr) as previously described[18]. Briefly, the average of 20 million reads per triplicates of Tγδ1, Tγδ17, Tγδ17i, and γδ25[+] and duplicate of Tγδ2 was obtained (total 14 samples). Raw data are available at the National Center for Biotechnology Information under accession number PRJNA549112. RNA-seq reads were aligned to the mouse reference genome (mm10) and most recent transcript annotations (GRCm38_ensGene_94) using STAR (v2.6.1-d)[50]. Expression levels of all transcripts were quantified by RSEM (v1.3.1)[51]. Differentially expressed genes were determined by DESeq2[52]. Expression levels for heat map and principal component analysis were based on regularized log₂-count of reads using DESeq2. Volcano plots were generated using Enhanced Volcano R package[52]. Gene set enrichment analysis was performed to calculate enrichment *P*-value with Benjamini-Hochberg correction procedure using Ingenuity Pathway Analysis[53]. To analyze iNKT, Th and ILC subsets, raw data of RNA-seq reads was downloaded from (SRA Project accession number: PRJNA318017 for iNKT,

ArrayExpress accession number: E-MTAB-2582 for Th, and GEO accession number: GSE85154 for ILC) and reanalyzed with the corresponded pipeline.

**Cell isolation for single-cell analysis.** Pooled thymi of 6-week-old BALB/c mice were used to isolate iNKT and MAIT cells and post-natal day 5 thymi were used isolate T cells. Single-cell suspensions of thymocytes were stained with PE conjugated CD1d or MR1 tetramers or anti-TCRγδ (GL3) antibody and enriched with anti-PE microbeads (Miltenyi) according to the manufacturer's instructions. After sorting, equal numbers of iNKT, MAIT, and γδ T cells were mixed and processed altogether.

**Single-cell RNA sequencing.** Libraries for scRNA-seq were prepared using the Chromium Single Cell 5′ Library & Gel Bead Kit (PN-1000014, 10X Genomics), Chromium Single Cell A Chip kit (PN-1000009, 10X Genomics), and Chromium i7 Multiplex Kit (PN-120262, 10X Genomics). Samples were loaded onto the Chromium Controller (10X Genomics) to generate gel bead-in-emulsions (GEMs) of 5000–7000 cells. Reverse transcription was performed using C1000 Touch Thermal Cycler with a deep-well block (Bio-Rad). Subsequent DNA purification and library generation was performed according to the manufacturer's instruction provided. Libraries were sequenced on an Illumina HiSeq4000 (paired-end 100 bp reads) aiming at an average of 50,000 read pairs per cell.

**Paired single-cell TCRαβ/ γδ sequencing.** Single-cell TCRαβ sequencing library was generated using the Chromium Single Cell 5′ Library Construction kit (PN1000020, 10X Genomics) and Chromium Single Cell V(D)J Enrichment Kit for mouse T cell (PN-1000071, 10X Genomics). In all, 1/22.5 of total cDNA was used to generate single-cell TCR sequencing libraries. Single-cell TCRγδ sequencing libraries were generated using the Chromium Single Cell 5′ Library construction kit (PN1000020, 10X Genomics) and custom primer sets as below. 1st PCR: 2 μM of forward primer (5′-AATGATACGGCGACCACCGAGATCTACACTCTTTCCCT ACACGACGCTC-3′) and 0.5 μM of reverse primers for each (5′-TCGAATCTC CATACTGACCAAGCTTGAC-3′, 5′-GTCTTCAGCGTATCCCCTTCCTGG-3′, 5′-CTTTCAGGCACAGTAAGCCAGC-3′ and 5′-TCTTCAGTCACCGTCAGCCA ACTAA-3′). 2nd PCR: 1 μM of forward primer (5′-AATGATACGGCGACCACC GAGATCT-3′) and 1 μM of reverse primers for each (5′- CCACAATCTTCTTG GATGATCTGAGACT-3′ and 5′- GTCCCAGTCTTATGGAGATTTGTTTCAG C-3′). Pooled libraries were sequenced on an Illumina HiSeq2500 (paired-end 150 bp reads) aiming at an average depth of 5000 read pairs per cell.

**Single-cell RNA-seq data preprocessing.** Raw reads from scRNA-seq were processed using the Cell Ranger software suite (v2.2.0). Briefly, reformatted reads were mapped to the mouse reference genome (GRCm38) with the Ensembl GRCm38.91 GTF file. For each replicate, a gene-by-cell count matrix was generated with default arguments except for expect-cells=5500 or 7000 and then aggregated into a single count matrix. Cells associated with empty droplets were identified and removed using the emptyDrops function of the DropletUtils (v1.2.2) R package[54] with FDR < 0.05. To filter out low-quality cells, cells with <1000 unique molecular identifiers (UMIs) and with >10% of UMIs assigned to mitochondrial genes were excluded, where the thresholds were determined by visually inspecting outliers in the PCA plot on the quality control metrics using the calculateQCMetrics function of the scater (v1.10.1) R package[55]. To remove cell-specific biases, cells were clustered using tse quickCluster function of the scran (v1.10.2) R package[56] with default arguments and cell-specific size factors were calculated using the compu-teSumFactors function of the same package with the minimum and maximum pool sizes of 100 and 200, respectively. Raw counts of each cell were divided by their cell-specific size factor and then log2-transformed with a pseudocount of 1.

A mixture of iNKT, MAIT, and γδ T cells was demultiplexed based on the sex of mice and TCR genotypes. First, cells were assigned to iNKT (containing the canonical iNKT Vα14-Jα18 (TRAV11/TRAV11D-TRAJ18) TCRα rearrangement), MAIT (containing the canonical MAIT Vα19-Jα33 (TRAV1-TRAJ33) TCRα rearrangement and not expressing Y-chromosomal genes), and γδ T cells (containing productive TCRγ and TCRδ rearrangements). Putative doublets, which contain both productive TCRα and TCRδ rearrangements or both canonical iNKT and MAIT TCRα rearrangements, were removed. Second, unassigned cells were reclassified based on the major cell type of clusters to which they belong. We identified highly variable genes (HVGs) using the decomposeVar function of the scran package with FDR ≤ 0.05 and biological variability >0.1, grouped all cells into 19 clusters using the FindClusters function of the Seurat (v2.3.4) R package[57] on the first 20 PCs of HVGs with resolution = 1.5 and visualized cells in the two-dimensional UMAP plot using the RunUMAP function of the Seurat package on 20 PCs. Unassigned cells in two clusters (cluster 11 and 14) annotated as CD4$^+$ CD8$^+$ double-positive cells were removed and remaining cells were re-clustered using the same method as above except for 25 PCs. For each cluster with >80% of the most abundant cell type, unassigned cells were classified into the major cell type and cells assigned to other minor cell types were removed as putative doublets. In other clusters (cluster 10, 13, and 17), using TCR expression, unassigned cells were annotated as MAIT cells (expressing TRAV1) or γδ T cells (expressing V gene segments for both TCRγ and TCRδ chains). For each cell type, all assigned cells

underwent a third round of clustering to filter out misclassified outlier cells. In γδ T cells, one cluster of 14 cells (cluster 7) was removed from further analysis.

**Single-cell RNA-seq data analysis.** For each cell type, we identified HVGs excluding TCR genes, clustered cells with 25 PCs and resolution = 0.8 and visualized cells in the two-dimensional UMAP plot, using the same methods as above. To visualize all assigned cells of iNKT, MAIT, and γδ T cells in the shared UMAP plot, HVGs were identified from all cells and 5 PCs were used. To identify subpopulations within G7 (Tγδ1), cells in G7 were grouped into four clusters using the SC3 (v1.10.1) R package[58] on HVGs. One cluster was assigned to G7-2 (Tγδ1) and other clusters were assigned to G7-1 (Tγδ1i) based on the expression levels of *Tbx21* and *Ifng*. Similarly, cells in G6 (Tγδ17) were grouped into G6-1 and G6-2 using TCR genes belonging to HVGs. For each cluster, marker genes were identified using the FindAllMarkers function of the Seurat package with default parameters. Differentially expressed genes (DEGs) between G7-1 and G7-2 were detected using the same method except logfc.threshold = 0.2 and min.pct = 0.05. The signature score of each functional subset of iNKT and γδ T cells was calculated by the average Z-score of log2 normalized counts of signature genes.

The pseudotime analysis was performed for each cell type using the Palantir (v0.2) python package[59]. Briefly, a nearest-neighbor graph ($k = 30$) was constructed using the first 10 diffusion components (DCs) of the 100 PCs of HVGs excluding TCR genes and visualized in the t-SNE plot based on the first four (for iNKT and MAIT cells) or five (for γδ T cells) DCs. An initial cell was defined by choosing a cell in N1 with the highest signature score of NKTp for iNKT, a cell in M1 randomly for MAIT cells and a cell in G1–3 with the highest signature score of Tγδp for γδ T cells. Gene expression trends of a union of marker genes of each cluster and subtype along differentiation trajectories toward type 1 and 17 were computed using the generalized additive models after imputing data with MAGIC[60]. The Z-scores of hierarchically clustered genes by the hclust function in R were visualized in a heat map. The enrichment analysis for gene ontology biological process terms in gene clusters were performed using the topGO (v.2.34.0) R package with the org.Mm.eg.db (v3.7.0) annotation data package. The clusters of MAIT cells were projected to the clusters of iNKT cells using the scmapCluster function of the scmap (v.1.4.1) R package with HVGs of all assigned cells of iNKT, MAIT, and γδ T cells[61]. The pseudotime analysis was validated using the Monocle 3 (v0.2.0) R package[62]. Cells were visualized in the UMAP plot from 50 PCs (MAIT cells) and 30 PCs (γδ T cells) with minimum distance of 0.3 and 15 (MAIT cells)/25 (γδ T cells) nearest neighbors. The trajectory was built using the learn_graph function of Monocle 3 package with our cell clusters and minimal_branch_len of 20.

**Single-cell TCR repertoire analysis.** Raw reads from paired V(D)J sequencing were processed using the cellranger vdj of the Cell Ranger (v2.2.0) with --chain = all for γδ T cells. The V(D)J segment based reference was constructed from IMGT using the fetch-imgt and cellranger mkvdjref of the Cell Ranger. Clonotypes (the same V/J composition and the same rearranged CDR3 sequences) called as high-confidence and productive were used for further analysis. For iNKT cells, TRAV11 and TRAV11D were considered to be the same.

**Flow cytometry.** Biotinylated PBS57 loaded or unloaded CD1d monomers and 6-FP or 5-OP-RU loaded MR1 monomers were obtained from the tetramer facility of the US National Institutes of Health. For intracellular staining, single-cell suspensions were surface stained, fixed, and permeabilized with eBioscience Foxp3 staining buffer set. 17D1 hybridoma[63,64] was provided from Robert E. Tigelaar (Yale University, USA) and used to detect TCR Vγ6. Biotinylated anti-Vγ7 antibody[65] was provided from Pablo Pereira (Institut Pasteur, France). Cells were analyzed on an LSR II (Becton Dickinson) and data were processed with FlowJo software (TreeStar). Antibodies used in the experiments are listed in Table 1.

**Intra-thymic injection.** Mice were anesthetized with an intraperitoneal injection of ketamine (90 mg/kg) and xylazine (9 mg/kg) and $0.1$–$1.0 × 10^5$ cells were directly injected into thymus after 1 mm incision of upper sternum. Buprenorphine (Buprenex, 900 μg) were injected four times for analgesics.

**Fetal thymic organ culture.** On embryonic day 15.5 (E15.5), fetal thymuses from C57BL/6 mice were removed and cultured on hydrophilic isopore membrane filter (0.8-μm pore size, Millipore, ATTP01300) placed on gelfoam sponge (Millipore, Medford, MA) in RPMI 1640 medium supplemented with 10% fetal bovine calf serum (Atlas Biologicals), 1% penicillin and streptomycin (HyClone), and 50 nM 2-ME (Sigma). Thymi were cultured with 2′-deoxyguanosine (Bio Basic, Amherst, NY) for 7 days and donor cells were colonized using hanging drop culture and analyzed after 7 days.

**In vitro cytokine production.** For in vitro stimulation with PMA and ionomycin, total thymocytes were enriched with iNKT, MAIT, or γδ T cells using MACS, were plated at a density of $1 × 10^6$ cells per ml in RPMI medium plus 10% (vol/vol) FCS.

**Table 1 List of antibodies and reagents.**

| | | |
|---|---|---|
| Anti-CD3ε APC-Cy7 (clone 145-2C11) (1/200) | TONBO | 25-0031-U100 |
| Anti-CD4 BUV395 (clone GK1.5) (1/300) | BD Biosciences | 563,790 |
| Anti-CD4 BV510 (clone RM4-5) (1/200) | BD Biosciences | 563,106 |
| Anti-CD8α BV650 (clone 53-6.7) (1/300) | BD Biosciences | 563,234 |
| Anti-CD19 PE-Cy7 (clone 1D3) (1/400) | BD Biosciences | 552,854 |
| Anti-CD24 BV605 (clone M1/69) (1/1000) | Biolegend | 101,827 |
| Anti-CD24 FITC (clone M1/69) (1/700) | BD Biosciences | 553,261 |
| Anti-CD24 PE/Cy7 (clone M1/69) (1/800) | Biolegend | 101,821 |
| Anti-CD25 APC (clone PC61) (1/800) | Biolegend | 102,012 |
| Anti-CD27 PerCP-eFluor710 (clone LG.GF9) (1/200) | Thermo Fisher Scientific | 46-0271-82 |
| Anti-CD44 PE (clone IM7) (1/300) | Biolegend | 103,008 |
| Anti-CD44 redFluor710 (clone IM7) (1/300) | TONBO | 80-0441-U100 |
| Anti-CD45R/B220 BV711 (clone RA3-6B2) (1/300) | BD Bioscience | 563,892 |
| Anti-CD122 FITC (clone TM-BETA 1) (1/100) | BD Bioscience | 553,361 |
| Anti-CD122 PE (clone TM-b1 [TM-beta1]) (1/100) | Thermo Fisher Scientific | 12-1222-81 |
| Anti-CD183 (CXCR3) PE-Cy7 (clone CXCR3-173) (1/100) | Biolegend | 126,516 |
| Anti-CD186 (CXCR6) BV421 (clone SA051D1) (1/150) | Biolegend | 151,109 |
| Anti-CD196 (CCR6) BV421 (clone 29-2L17) (1/100) | Biolegend | 129,828 |
| Anti-CD279 (PD-1) APC (clone J43) (1/400) | BD Bioscience | 562,671 |
| Anti-IL-25R (IL17RB) Alexa Fluor647 (clone 9B10) (1/200) | Biolegend | 146,304 |
| Anti-IL-25R (IL17RB) PE (clone MUNC33) (1/100) | Thermo Fisher Scientific | 12-7361-80 |
| Anti-γδ T-Cell Receptor BV421 (clone GL3) (1/200) | BD Bioscience | 562,892 |
| Anti-γδ T-Cell Receptor PE (clone GL3) (1/100) | BD Bioscience | 553,178 |
| Anti-γδ T-Cell Receptor PE-CF594 (clone GL3) (1/300) | BD Bioscience | 563,532 |
| Anti-Vγ1.1 (Heilig and Tonegawa's system: Vγ1) TCR BV421 (clone 2.11)(1/300) | BD Bioscience | 566,308 |
| Anti-Vγ1.1 + Vγ1.2 (Heilig and Tonegawa's system: Vγ1 + Vγ2) TCR PE (clone 4B2.9) | Biolegend | 142,704 |
| Anti-Vγ2 (Heilig and Tonegawa's system: Vγ4) TCR BV786 (clone UC3-10A6) (1/300) | BD Bioscience | 742,313 |
| Anti-Vγ3 (Heilig and Tonegawa's system: Vγ5) TCR BV510 (clone 536) (1/200) | BD Bioscience | 743,239 |
| Anti-Vγ5/Vδ1+ and Vγ6/Vδ1+ (Heilig and Tonegawa's system: Vγ5Vδ1+ and Vγ6Vδ1+) TCR rat IgM antibody (clone 17D1) | kindly provided by Dr. Robert Tigelaar | |
| Anti- Vγ7 TCR Biotinylated (clone F2.67) (1/600) | kindly provided by Dr. Pablo Pereira | |
| Anti-TCRβ chain (clone H57-597) (1/200) | BD Bioscience | 560,656 |
| Anti-Vδ 6.3/2 TCR BV711 (clone 8F4H7B7) (1/100) | BD Bioscience | 744,476 |
| Purified CD16/32 (clone 93) (1/200) | Biolegend | 101,302 |
| Anti-EOMES eFluor 450 (clone Dan11mag) (1/100) | Thermo Fisher Scientific | 48-4875-82 |
| Anti-IFN-γ PE-CF594 (clone XMG1.2) (1/200) | BD Bioscience | 562,303 |
| Anti-IL-4-Alexa647 (clone 11B11) (1/100) | Biolegend | 504,110 |
| Anti-IL-4 BV421 (clone 11B11) (1/50) | Biolegend | 504,119 |
| Anti-Ki-67 FITC (clone SolA15) (1/600) | Thermo Fisher Scientific | 11-5698-82 |
| Anti-IL-17A BV650 (clone TC11-18H10) (1/200) | BD Bioscience | 564,170 |
| Anti-PLZF Alexa Fluor647 (clone R17-809) (1/200) | BD Bioscience | 563,490 |
| Anti-PLZF PE-CF594 (clone R17-809) (1/400) | BD Bioscience | 565,738 |
| Anti-RORγt PerCP-Cy5.5 (clone Q31-378) (1/200) | BD Bioscience | 562,683 |
| Anti-RORγt PE-CF594 (clone Q31-378) (1/300) | BD Bioscience | 562,684 |
| Anti-T-bet PE-Cy7 (clone eBio4B10) (1/200) | Thermo Fisher Scientific | 25-5825-82 |
| Anti-rat IgM FITC (clone MRM-47) (1/300) | Biolegend | 408,905 |
| Streptavidin APC-Cy7 (1/400) | BD Bioscience | 554,063 |
| Streptavidin PE | BD Bioscience | 554,061 |
| Streptavidin-R-Phycoerythrin | ProZyme | PJRS25 |
| Anti-PE MicroBeads Ultrapure | Miltenyi Biotec | 130-105-639 |
| Anti-Human CD2 FITC (clone RPA-2.10) (1/50) | BD Bioscience | 555,326 |
| Anti-CD3 BUV395 (clone UCHT1) (1/300) | BD Bioscience | 563,546 |
| Anti-CD3 Alexa Fluor594 (clone UCHT1) (1/300) | Biolegend | 300,446 |
| Anti-CD4 APC/Cy7 (clone OKT4) (1/300) | Biolegend | 317,418 |
| Anti-CD8α Alexa Fluor700 (clone RPA-T8) (1/300) | Biolegend | 301,028 |
| Anti-CD19 V500 (clone HIB19) (1/300) | BD Bioscience | 561,121 |
| Anti-CD45RA BV650 (clone HI100) (1/300) | BD Bioscience | 563,963 |
| Anti-CD45RO BV711 (clone UCHL1) (1/300) | BD Bioscience | 563,722 |
| Anti-CD161 Alexa Fluor488 (clone HP-3G10) (1/25) | Biolegend | 339,924 |
| Anti-TCR Vα7.2 BV785 (clone 3C10) (1/25) | Biolegend | 351,722 |
| Anti-TCR γ/δ PerCP/Cy5.5 (clone B1) (1/25) | Biolegend | 331,224 |
| Anti-TCR Vγ9 BV421 (clone B3) (1/50) | BD Bioscience | 744,034 |
| Anti-TCR δ (TCR Vδ1-Jδ2) FITC (clone TS-1) (1/50) | Thermo Fisher Scientific | TCR2055 |
| Anti-TCR Vδ2 FITC (clone B6) (1/50) | Biolegend | 331,406 |
| Anti-IFN-γ BUV395 (clone B27) (1/25) | BD Bioscience | 563,563 |
| Anti-IL-4 BV605 (clone MP4-25D2) (1/25) | Biolegend | 500,827 |
| Anti-IL-17A APC (clone eBio64DEC17) (1/25) | Thermo Fisher Scientific | 17-7179-42 |
| Anti-T-bet PE-Cy7 (clone eBio4B10) (1:100) | Thermo Fisher Scientific | 25-5825-82 |
| Anti-PLZF Alexa Fluor647 (clone R17-809) (1/100) | BD Bioscience | 563,490 |
| Anti-RORγt PE (clone Q21-559) (1/20) | BD Bioscience | 563,081 |

Cells were incubated for 4 h with cell stimulation cocktail (ThermoFisher) with Monensin (eBioscience) for last 2 h and analyzed for intracellular cytokines by flow cytometry.

**Statistical analysis.** Prism software (Graphpad) was used for statistical analysis and all data were represented as mean ± SD. Unpaired two-tailed *t*-tests were used for data analysis and the generation of *P* values. *P* < 0.05 was defined as significant.

**Reporting summary.** Further information on research design is available in the Nature Research Reporting Summary linked to this article.

## Data availability

The bulk and single-cell RNA sequencing data have been deposited in the SRA database under the accession code PRJNA549112 and are available. All the other data are included within the article, source data or supplemental information or available from the authors upon reasonable requests. Source data are provided with this paper.

## Code availability

https://github.com/scg-dgist/Nature-Communications-innate-tcells

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

## Acknowledgements

We wish to thank Rebecca O'Brien (National Jewish Health, USA) and Koichi Ikuta (Kyoto University, Japan) for providing Vγ4/6 KO mice, Robert E. Tigelaar (Yale University, USA) for providing 17D1 hybridoma, and Pablo Pereira (Institut Pasteur, France) for providing anti-Vγ7 antibodies. We also thank Sung-Wook Hong for helping IPA analysis (University of Minnesota, USA) and Joonsoo Kang (University of Massachusetts, USA) and Kristin Hogquist (University of Minnesota, USA) for critical reading of the manuscript. This work was supported by projects NRF-2019R1F1A1059237 (to Y. J.L.), 2018R1A2B6002657 (to S.K.), and 2018R1A5A1025511 and 2019M3A9D5A01102794 (to J.K.K.) funded by the Korean Ministry of Science, Information/Communication Technology and Future Planning and BK21 Plus (10Z20130012243) funded by the Ministry of Education (to M.L.), Korea.

## Author contributions

M.L. designed and performed experiments and analyzed data; E.L. and E.S.P. analyzed scRNA-seq data; S.K.H. and K.L. analyzed bulk RNAseq data and provided research interpretation; Y.H.C., D.K., H.C., and M.-S.R. performed experiments; D.J.J. and E.C.S. provided human samples and research interpretation; and S.K., J.K.K., and Y.J.L. directed the study, analyzed data, and wrote the manuscript. Y.J.L. conceptualized the research.

## Competing interests

The authors declare no competing interests.
