## [Peer Review File · Nature Communications]

Editorial Note: Parts of this peer review file have been redacted as indicated to remove third-party material where no permission to publish could be obtained.

Reviewers' comments:

Reviewer #1 (TF regulation, B/T development, hematopoiesis) (Remarks to the Author):

Transcriptional landscapes of innate T cells at the single cell level reveal a shared lineage differentiation process regulated by clonal selection and proliferation

In this manuscript the authors define new subtypes for MAIT and $\gamma\delta$ T cells: MAIT1, MAIT2, MAIT17, $\gamma\delta$ T1, $\gamma\delta$ T2 and $\gamma\delta$ T17. The transcriptional differences among these subtypes were similar to the differences observed for NKT1, NKT2 and NKT17. The authors conducted single cell RNA-seq and TCR-seq, and confirmed the clusters of MAIT1, MAIT2, MAIT17, $\gamma\delta$ T1 and $\gamma\delta$ T17 at the single cell transcriptome level. The authors employed a Palantir package to identify the potential differentiation pathway of these subtypes. The TCR clonotyping analysis revealed that these subtypes underwent clonal selection prior to terminal differentiation. Finally, the authors confirmed that similar subtypes of MAIT, $\gamma\delta$ T and iNKT can be found in humans.

This is a nice paper with significant new data that is of significant interest. The paper is well written and overall the data are convincing. The major issue of this paper is that the results do not rule out other possible developmental pathways. Rather than claiming the MAIT, $\gamma\delta$ T and iNKT subtypes mature via lineage differentiation rather than linear maturation it might be more prudent to conclude that $\gamma\delta$ T1/2/17 I types mature via lineage differentiation and the corresponding MAIT and iNKT subtypes are most likely to mature through lineage differentiation.

Additional comments:

The authors have used only a single cell RNA-seq lineage identification package (Palantir). I recommend that the authors apply other packages, e.g. Monocle, Velocyto, GBfate, etc. as well to complement their analysis.

The authors showed that NKT1 and NKT2/17 are in two different lineages. However, there is no further evidence supporting the claim that NKT2 and NKT17 are two different differentiation end points. (Gata3 cKO mouse has depleted NKT2/17 levels and increased NKT1 level; TBET-deficient mouse has depleted NKT1 level and increased NKT2/17 levels)

There is not sufficient evidence supporting the branching of MAIT1/2/17. In TBET-deficient mouse, MAIT17 cells are depleted while MAIT2 cells are increased in abundance. As revealed in Figure 3A it is possible that MAIT17 represents a later stage of maturation and MAIT2 cell reflects an earlier stage (MAIT17 is the M8 cluster in the plot and MAIT2 is the M2 cluster).

Minor comment:

The authors generated two replicates for the scRNA-seq. It seems to me that the two replicates are not evenly mixed in Fig.S5D. It would be better to present a table of "clusters by replicates" with the values indicating how many cells from each cluster were derived from which replicate. This would be a more straight-forward way of showing that the batch effects were minimal.

Reviewer #2 (gd T, unconventional T, NKT) (Remarks to the Author):

This is an extensive and ambitious study that attempts to examine, in parallel, MAIT cells, NKT cells and gd T cells and their different developmental stages in mice and to some extent in humans. There are some very interesting findings in this study and it potentially represents an important contribution to the field. However because there is so much going on and so many different factors under investigation, I found some of the results to be very confusing and in some sections I could not

understand how a particular interpretation or conclusion was derived. I have provided a list of concerns, both major and minor, that I hope will help the authors to better analyse and interpret their findings and present this study more clearly.

1. A major concern relates to the statement on page 9 "Surprisingly, MAIT cells having one or more non-canonical TCR subchains were approximately 50% in M1, but decreased dramatically in MAIT1 (M5, 10%), MAIT2 (M2, 28%) and MAIT17 (M8, 21%) cells (Figure 4F, right). Therefore, it is conceivable that the semi-invariant nature of MAIT cells is not a result of positive selection but a consequence of clonal selection after stage 2." The problem here is that non-canonical TCRs may represent contaminating cells that were not really MR1-tetramer reactive, because earlier reports have shown that 98% of MAIT cells in thymus used the canonical Va19Ja33 TCR (Koay et al 2019 Nature Comms 10, 2243) including stage 1 MAIT cells (M1) where all cells tested were Va19Ja33 (ref 10). If these non-canonical cells are contaminants that are not really MR1-restricted T cells, this could have a major impact on interpretation of the transcriptomic data from these populations, especially M1 where 50% were non-canonical. This would also impact on the clonal selection hypothesis proposed to explain this phenomenon. This needs to be more carefully investigated. To be sure that real MAIT cells follow the transcriptional patterns as described for the entire population, if possible the transcriptomic data should separately examine canonical and non-canonical MAIT cells. The authors should also test some of the non-canonical TCR sequences for MR1 reactivity by generating TCR transfected cell lines.

2. An unexpected finding in this paper was the identification of a MAIT2 subset that has an IL-4 phenotype. Others have failed to find these cells in the thymus (ref 8). The results state that the MAIT2 cells are encompassed within stage 2 which could explain this discrepancy. However stage 2 of MAIT cell development is defined as CD44-, and it is clear from the CD44 histogram of MAIT2 cells in Fig S1A that many or most of the MAIT2 cells are CD44+, overlapping with MAIT1 cells that are also CD44+. Therefore, this does not adequately explain the discrepancy between this paper and the data described in ref 8. The problem stated above that some (28%) of these may be contaminants is also relevant here. Because there is controversy about whether MAIT2 cells exist, the use of huCD2 as an IL-4 reporter is not really adequate here. Direct detection of IL-4 by ICS would help to make this case convincingly. It is also possible that the difference relates to the different mouse strain backgrounds examined in this study versus ref 8.

3. The comparison of DEGs between Tgd1:Tgd2:Tgd17 and NKT1:NKT2:NKT17 was confusing. If I understood correctly, it shows a significant overlap between lineages such that roughly one fifth of DEGs were shared between lineages. While this suggests some overlap, it also highlights that most (roughly 4/5ths) of the DEGs are actually distinct for these lineages and it could be argued that they are mostly distinct in their DEGs. With this in mind, is it reasonable to argue that they are 'highly similar'? Some overlap is expected, given common markers are used to isolate them such as IL-17 and its associated family members. How would these patterns compare to conventional Th1:Th2:Th17 subsets? Would they be expected to overlap to a similar extent? In some ways, what distinguishes these lineages versus what do they have in common is an equally interesting question.

4. I was unable to understand the data and explanation for Fig S4B-C leading to the point that "Tγδ 17i cells exclusively generated IL-17 producing mature Tγδ 17 cells". Also "We also analyzed the signature genes of Tγδ 17i cells and found 349 DEGs to be specifically upregulated (Figure S4D-E), indicating they are unique developmental intermediates of Tγδ 17 cells." I also do not understand this conclusion has been reached. The 349 DEGs seem to be common to gd1, gd2 and gd17 cells. Some more careful explanation here will help.

5. Because NKT0 and NKTp populations are very rare and difficult to isolate, these need to be confirmed as NKT based on TCRA chain expression. It is important to show their TCR sequence

because other analyses of these cells have been incorporated in the paper. Were they all Va14Ja18?

6. p.6 "and we validated these results by flow cytometry" – no data is shown or referenced.

7. For the multiplexed scRNAseq, using males for NKT and females for MAIT cells – doesn't this create a confounding variable being the impact of sex difference?

8. The data in Fig S6 is inadequately explained and confusing. A code for what red and grey colours represent is also required.

9. This statement "indicating that M2 contains both immature and mature MAIT2 cells that we identified in flow cytometry (Figure 1A)." seems inconsistent with the data in Fig 1A

10. The statement "we observed the generation of $\gamma\delta 24+25+$ cells from $\gamma\delta 24+25-$ cells in fetal thymic organ culture experiment (FTOC) (data not shown)" seems like an important and relevant result that should be shown. Also, this approach would be very useful for determining precursor/progeny relationships proposed for MAIT cell subsets and validating the scRNAseq based data for other findings in this paper.

11. The statement "we observed a uniformly high level of Ki-67 expression in MAIT cells (data not shown), and cell cycle-regulated genes as well as lineage specific signatures were upregulated during maturation (Figure S10 and Table S4)." is inconsistent with a recent report (Winter et al 2018 Immunology Cell Biology, 97, 190).

12. There is a problem with the staining in Fig 7b. Of the Vg9+ T cells, a major population of Vg9 intermediate cells are Vd2. However, when stained with Vd1, this population of Vg9 intermediate cells disappears and all Vg9 cells are Vg9hi. This is reflected by the angled quadrant gates for Vd2 staining suggesting a compensation problem.

13. The statement on p11 that "ROR γ t+ NKT, MAIT, and $\gamma\delta$ T cells produced both IFN- γ and IL-17A" is inaccurate. For gd and MAIT cells, it looks like just as many produce IL-17 but not IFN γ . It is also unclear how the quadrant markers were placed, as they move around between populations. This has an impact on the % reported.

14. The final statement in results "human ROR γ t+ cells simultaneously express an intermediate level of TBET with ROR γ t." is problematic because there are no negative controls for this staining, so it is impossible to tell if the Tbet staining is negative, positive or intermediate. How were the regions positioned to determine what was negative or positive?

15. Discussion p12. "The absence of TBET rather suppressed the development of type 17 cells both in MAIT and $\gamma\delta$ T cells and this could be explained by the presence of immature type 1 and type 17 precursors, predicted by the scRNAseq analysis (Figure 6)". This is confusing and I don't understand how this point comes from the data.

16. A minor point - the word 'quasi-identical' (p4) to compare MAIT1 and MAIT17 to NKT1 and NKT17 cells is confusing and seems to be an oxymoron. Identical means exactly the same, quasi means partial but not complete. Why not just 'similar'.

Reviewer #3 (NKT, NK, MAIT) (Remarks to the Author):

In this manuscript Lee et al. examine the development of the most common "innate" T lymphocytes together, namely the iNKT cells, the MAIT cells and the $\gamma\delta$ T cells. Using multicolor flow cytometry as well as single cell RNA-Seq coupled with TCR sequencing, they show that for each of these lymphocyte populations, similar effector subsets exist. This mostly confirms what was already known in the field, with the exception of MAIT2-like cells that had not been previously described. Furthermore, through the use of V γ -specific knock-out mice, the authors argue that the specific nature of the $\gamma\delta$ TCR plays no role in the subset specification.

There are a lot of data in the manuscript and it is sometimes difficult to keep track as to what the actual message is. Although the experiments are properly performed and controlled, there are a few points that need clarification, as they do not conform to what would have been expected from previous literature. In addition, I am unsure that the TCR sequencing data have enough power to make any argument as to whether the exact nature of a given TCR might play a role, or not, in the subset specification of any of these lymphocyte populations. In conventional BALB/c mice, most of $\gamma\delta^+$ Ror γ t $^+$ cells are also V γ 4 $^+$. The authors show that in the V γ 4/6 KO mice, the proportion of Ror γ t $^+$ $\gamma\delta$ T cells do not change compared to control mice but that the Ror γ t $^+$ cells have now been replaced by V γ 1/V δ 6.3 $^+$ cells. This finding is used to conclude that $\gamma\delta$ TCR specificity is irrelevant to subset specification. However, this conclusion does not seem entirely supported by the data showed in Fig 5c, where the proportion of V γ 1/V δ 6.3 $^+$ amongst T $\gamma\delta$ 17 cells in WT vs V γ 4/6 KO actually does not change (66% vs 71%). From these data, it would appear that some V γ 1/V δ 6.3 $^+$ T cells are in fact T $\gamma\delta$ 17 in wildtype animals and that these cells could be expanding and filling the niche when V γ 4 $^+$ cells are absent in the KO mice and not that there is a diversion of the development of V γ 1/V δ 6.3 $^+$ T cells from a T $\gamma\delta$ 2 to a T $\gamma\delta$ 17 phenotype.

Finally, the way the data are presented and structured suggest that all developing $\gamma\delta$ T cells commit to one of three functionally different subsets exemplified by the expression of either Ror γ t, PLZF or Tbet, similarly to MAIT and iNKT cells. While this certainly happens to some $\gamma\delta$ T cells, a significant proportion of mature (as defined the lack of expression of CD24) $\gamma\delta$ T cells in the mouse thymus do not express any of these transcription factors. These cells are not apparent in any of the transcriptional analyses performed. How are these finding reconciled with previous literature where non-innate programmed $\gamma\delta$ T cells have been defined (see Bonneville et al. 2010 for review).

Comments:

It is interesting that the authors identify a previously unknown subset of MAIT cells. Do these MAIT2 cells also actively secrete IL-4 at steady-state much like iNKT2 and T $\gamma\delta$ 2 (Figure 2C would argue that they do not)? If not, what is unique to their transcriptional program that they do not produce IL-4? This is perhaps evident even in their expression of IL17RB as they do not express high levels of this cytokine receptor and consequently, may not have received the proper cues to differentiate further as iNKT2 and T $\gamma\delta$ 2 cells may have?

It would be important to conduct PCA on all the 18000 genes for all subsets of each of the three T cell populations to see how all of them separate, instead of just the 120 DEGs between iNKT and $\gamma\delta$ subsets as conducted in Figure 1E. By selecting the genes that are common to the cells types, it is perhaps not surprising that they would then cluster together in PCA analysis. Because the expression of several of these genes are likely consequential to the expression of the "master" transcription factors that define the various subsets, it is not surprising that the cells share a similar transcriptional profile. What is perhaps more relevant is what transcriptional profile defines the unique identity of each of these lymphocyte populations?

Although the authors claim in Figure S4B-C that they are conducted fate mapping studies to conclude that the T $\gamma\delta$ 17i cells give rise to the T $\gamma\delta$ 17 cells, this is an inappropriate statement. The FTOC

experiments conducted later in the submission certainly suggest a precursor-product relationship but simply marking these cells using reporters is insufficient to conclude that the immature cells generate the mature population. This statement here should be removed and added to the section with the FTOC figure.

Why do the authors recover such few iNKT1 cells in their scRNA-seq experiment while they sorted total iNKT cells from the thymus of 7 weeks old BALB/c mice? Also, the IFN γ signature in iNKT1 and T $\gamma\delta$ 1 cells does not appear to be shared by the majority of the cells belonging to these subsets unlike that observed in MAIT1 cells. Can the authors identify any transcriptional signature leading to this difference?

As mentioned earlier, it is unclear whether the experiments depicted in Figure 5C actually support the authors' claim. There is already a similar proportion of V γ 1V δ 6.3+ cells that are Ror γ t+ in WT mice. It is possible that these cells simply expand in the V γ 4/6 KO mice due to lack of competition and not that other V γ 1 cells are diverted to fill the niche.

How do the authors account for the expression of CD122 in the immature populations before the expression of T-bet? T-box proteins have been previously shown to drive the expression of CD122 but that is not the case here.

The development of iNKT and MAIT cells is thought to occur through positive selection by CD1d and MR1 expressed by DP thymocytes in a SLAM/SAP/Fyn-dependent manner (Wang and Hogquist, 2018). Such process is thought to uniquely impart the innate features that are acquired by the iNKT and the MAIT cells. Since the authors propose that $\gamma\delta$ T cells also end up with a very similar transcriptional profile to iNKT and MAIT cells, it would perhaps be informative to discuss whether similar developmental pathways might also be involved in the development of $\gamma\delta$ subsets.

Reviewer #1 (TF regulation, B/T development, hematopoiesis) (Remarks to the Author):

Transcriptional landscapes of innate T cells at the single cell level reveal a shared lineage differentiation process regulated by clonal selection and proliferation

In this manuscript the authors define new subtypes for MAIT and $\gamma\delta$ T cells: MAIT1, MAIT2, MAIT17, $\gamma\delta$ T1, $\gamma\delta$ T2 and $\gamma\delta$ T17. The transcriptional differences among these subtypes were similar to the differences observed for NKT1, NKT2 and NKT17. The authors conducted single cell RNA-seq and TCR-seq, and confirmed the clusters of MAIT1, MAIT2, MAIT17, $\gamma\delta$ T1 and $\gamma\delta$ T17 at the single cell transcriptome level. The authors employed a Palantir package to identify the potential differentiation pathway of these subtypes. The TCR clonotyping analysis revealed that these subtypes underwent clonal selection prior to terminal differentiation. Finally, the authors confirmed that similar subtypes of MAIT, $\gamma\delta$ T and iNKT can be found in humans.

This is a nice paper with significant new data that is of significant interest. The paper is well written and overall the data are convincing. The major issue of this paper is that the results do not rule out other possible developmental pathways. Rather than claiming the MAIT, $\gamma\delta$ T and iNKT subtypes mature via lineage differentiation rather than linear maturation, it might be more prudent to conclude that $\gamma\delta$ T1/2/17 I types mature via lineage differentiation and the corresponding MAIT and iNKT subtypes are most likely to mature through lineage differentiation.

We thank for the positive comments of the reviewer. In the above sentence, we understood underlined phrase as 'linear maturation'. In this manuscript, we showed that innate T cells share common developmental pathway and **lineage differentiation model** explains our experimental results. In **linear maturation model**, NKT cells serially mature from IL-4 or IL-17 producing 'immature cells' into IFN γ secreting 'mature cells'. However, recent reports supported lineage differentiation model, in which progenitor cells give rise to terminally differentiated functional subsets, which were designated as NKT1, NKT2 and NKT17 cells. To confirm this, we did intra-thymic transfer experiments and showed NKT1, NKT2 and NKT17 cells are terminally differentiated (ref #8 and added Figure S14). In $\gamma\delta$ T cells, their functional classification has been based on TCR γ chain usage (**instruction by TCR**), in which each TCR V γ chain determines functional fate of cells. However, we found identical clonotypes both in T $\gamma\delta$ 1 and T $\gamma\delta$ 17 cells (Figure 4E), and TCR sequencing data supported the idea that intrathymic innate differentiation of $\gamma\delta$ T cells accompanies clonal selection and proliferation, indicating lineage fate of each TCR V γ chain is flexible. These results indicate that lineage fate of $\gamma\delta$ T cells are not instructed by their TCRs, but determined stochastically during lineage differentiation. If we say $\gamma\delta$ T cells develop via *linear maturation process*, we should show that there are precursor-progeny relationships between T $\gamma\delta$ 1, T $\gamma\delta$ 2 and T $\gamma\delta$ 17 cells, which are very unlikely.

Additional comments:

The authors have used only a single cell RNA-seq lineage identification package (Palantir). I recommend that the authors apply other packages, e.g. Monocle, Velocity, GBfate, etc. as well to complement their analysis.

We thank for the reviewer's suggestion. Additional analysis using Monocle3 was added in Figure S10, which we obtained similar results.

The authors showed that NKT1 and NKT2/17 are in two different lineages. However, there is no further evidence supporting the claim that NKT2 and NKT17 are two different differentiation end points. (Gata3 cKO mouse has depleted NKT2/17 levels and increased NKT1 level; TBET-deficient mouse has depleted NKT1 level and increased NKT2/17 levels)

In our previous experiments, we showed NKT2 cells do not give rise to NKT1 or NKT17 cells in intrathymic transfer experiments (Ref #8, Lee et al, 2013 Nat Immunol). We additionally tested the fate of NKT1 and NKT17 cells in both B6 and BALB/c mice after intrathymic injection and found they are also terminally differentiated. We added this result in Figure S14.

There is not sufficient evidence supporting the branching of MAIT1/2/17. In TBET-deficient mouse, MAIT17 cells are depleted while MAIT2 cells are increased in abundance. As revealed in Figure 3A it is possible that MAIT17 represents a later stage of maturation and MAIT2 cell reflects an earlier stage (MAIT17 is the M8 cluster in the plot and MAIT2 is the M2 cluster).

In Figure 3A, t-SNE plot shows the developmental trajectory of MAIT2 is in the middle of MAIT1 or 17 differentiation route. However, UMAP plots of Figure 3A and Figure S10 show M2 is not in the middle of pathway from M1 to M8. These analysis, however, would provide indirect messages and we need direct in vivo data to prove precursor-progeny relationship between MAIT2 and MAIT17 cells.

More convincing evidence of terminal differentiation of MAIT subsets would be intrathymic transfer experiments as we have done using NKT cells. However, sorting each MAIT subset was highly inefficient due to their extreme rarity, especially MAIT2 cells, which were only 50 cells per mice. In TBET deficient mice, MAIT1 cells are absent, and it is not clear whether increased MAIT2 cells are due to decreased MAIT1 or MAIT17 cells. As for $\gamma\delta$ T and iNKT cells, $T\gamma\delta17$ cells are absent in *Rorc* KO mice, but there is no increase in $T\gamma\delta2$ cells (Figure 5D) and NKT2 cells were shown to be terminally differentiated (ref#8). Based on the developmental similarities between NKT, MAIT and $\gamma\delta$ T cells, we speculate that MAIT2 cells are also terminally differentiated. However, as we cannot prove or disprove whether MAIT2 cells are developmental intermediate or terminally differentiated, we edited the manuscript in page 7 line 11-13 and highlighted in yellow.

Minor comment:

The authors generated two replicates for the scRNA-seq. It seems to me that the two replicates are not evenly mixed in Fig.S5D. It would be better to present a table of "clusters by replicates" with the values indicating how many cells from each cluster were derived from which replicate. This would be

a more straight-forward way of showing that the batch effects were minimal.

We thank for the reviewer's suggestion. We added cluster of replicates in Figure S5D.

Reviewer #2 (gd T, unconventional T, NKT) (Remarks to the Author):

This is an extensive and ambitious study that attempts to examine, in parallel, MAIT cells, NKT cells and gd T cells and their different developmental stages in mice and to some extent in humans. There are some very interesting findings in this study and it potentially represents an important contribution to the field. However because there is so much going on and so many different factors under investigation, I found some of the results to be very confusing and in some sections I could not understand how a particular interpretation or conclusion was derived. I have provided a list of concerns, both major and minor, that I hope will help the authors to better analyse and interpret their findings and present this study more clearly.

We thank for the reviewer's comments and tried our best to address all concerns.

1. A major concern relates to the statement on page 9 "Surprisingly, MAIT cells having one or more non-canonical TCR subchains were approximately 50% in M1, but decreased dramatically in MAIT1 (M5, 10%), MAIT2 (M2, 28%) and MAIT17 (M8, 21%) cells (Figure 4F, right). Therefore, it is conceivable that the semi-invariant nature of MAIT cells is not a result of positive selection but a consequence of clonal selection after stage 2." The problem here is that non-canonical TCRs may represent contaminating cells that were not really MR1-tetramer reactive, because earlier reports have shown that 98% of MAIT cells in thymus used the canonical Va19Ja33 TCR (Koay et al 2019 Nature Comms 10, 2243) including stage 1 MAIT cells (M1) where all cells tested were Va19Ja33 (ref 10). If these non-canonical cells are contaminants that are not really MR1-restricted T cells, this could have a major impact on interpretation of the transcriptomic data from these populations, especially M1 where 50% were non-canonical. This would also impact on the clonal selection hypothesis proposed to explain this phenomenon. This needs to be more carefully investigated. To be sure that real MAIT cells follow the transcriptional patterns as described for the entire population, if possible the transcriptomic data should separately examine canonical and non-canonical MAIT cells. The authors should also test some of the non-canonical TCR sequences for MR1 reactivity by generating TCR transfected cell lines.

We added Figure S13 to address this issue and added explanation in page 9 line 12-23 in the manuscript and highlighted in yellow. A detailed description follows.

- 1) We used MR1 tetramer positive cells that had more than 98% purity after sorting (Figure S2D). We detected about 2% of contaminated TCR negative DP thymocytes, which were excluded from analysis (Figure S5A).
- 2) In our analysis, we sequenced total 1,892 cells and defined non-canonical MAITs as cells that have at least one of three TCR sub-chains (TRAV, TRAJ and TRBV) are non-canonical. With this criteria we detected 669 MAIT cells have non-canonical TCRs (35%). Majority of them (447 cells out of 669 cells, 66.7%) had non-canonical TCRs only in TRBV (Figure S13A) with canonical usage of TRAV1 and TRAJ33. It is extremely unlikely that these cells are the result of random contamination

considering the rare usage of TRAV1/TRAJ33 in conventional T cells. We detected 89 cells that had all non-canonical TRAV, TRAJ and TRBV and these cells might be worthy of testing MR1 reactivity in TCR transfected cell lines. However, their frequencies in each MAIT clusters were very similar with other non-canonical MAITs and even we exclude these 89 cells from analysis, we still have same conclusion. We have not established TCR transfection system, and it would take at least a few months for the analysis. Even in that case, we cannot test all the non-canonical TCRs we identified.

- 3) As the reviewer suggested, we compared transcriptional profile of MAITs with canonical and non-canonical TCRs. We found that DEGs were only TCR genes themselves (added Figure S13C). In addition, we compared the expression pattern of stage specific MAIT signature genes that had been identified by others (Legoux and Lantz, Nat Immunol, 2019, 1244-1255). In this analysis, we also found similar transcriptional profiles in MAITs with canonical and non-canonical TCRs for signature gene sets of MAIT0, MAIT1 and MAIT17b cells (Figure S13B). As a control, we found that there are 102-429 DEGs between MAITs and contaminated DP populations (Figure S13D).
- 4) In previous literature (ref #28, Koay et al 2019 Nature Comms 10, 2243), they depleted CD24⁺ cells before sorting, and therefore, 98% usage of canonical TRAV1 gene is within CD24⁻ cells. In this analysis, they detected 1 cell with non-canonical TRAV out of 53 cells. In our analysis, we found 1460 cells are CD24⁻ MAIT cells and among them, 113 cells had non-canonical TRAV sequence ($113/1460 \times 100 = 7.7\%$, from Table S3). Koay et al also detected two non-canonical TRAJ ($2/53 \times 100 = 3.7\%$) and nine non-canonical TRBV ($9/53 \times 100 = 16.9\%$). In our analysis, these were 8.8% and 25%, respectively. In other previous literature (ref#10 Koay et al, 2016, Nat Immunol), they showed TCR sequences of stage1 CD24⁺CD44⁻ (13 cells) and stage 3 CD24⁻CD44⁺ (4 cells) MAITs, and found 2 out of 13 stage 1 CD24⁺ cells are non-canonical (Table S1, cell #6 and 7, $2/13 \times 100 = 13\%$). Considering that we have analyzed much higher cell number than previous reports and used different mice strains (BALB/c vs B6), we would consider that these differences are within reasonable range of variation.

2. An unexpected finding in this paper was the identification of a MAIT2 subset that has an IL-4 phenotype. Others have failed to find these cells in the thymus (ref 8). The results state that the MAIT2 cells are encompassed within stage 2 which could explain this discrepancy. However stage 2 of MAIT cell development is defined as CD44⁻, and it is clear from the CD44 histogram of MAIT2 cells in Fig S1A that many or most of the MAIT2 cells are CD44⁺, overlapping with MAIT1 cells that are also CD44⁺. Therefore, this does not adequately explain the discrepancy between this paper and the data described in ref 8. The problem stated above that some (28%) of these may be contaminants is also relevant here. Because there is controversy about whether MAIT2 cells exist, the use of huCD2 as an IL-4 reporter is not really adequate here. Direct detection of IL-4 by ICS would help to make this case convincingly. It is also possible that the difference relates to the different mouse strain backgrounds examined in this study versus ref 8.

Figure S1A shows type2 cells have low to intermediate levels of CD44, whereas type17 cells are all CD44^{high}. We re-analyzed Figure S1A in attached Figure 1A for reviewers, which showed S2 cells

contain PLZF^{hi} MAIT2 cells, whereas they are almost absent in S3 gating. This is consistent with previous report that showed the absence of PLZF^{hi} cells in CD44^{hi} S3 MAITs (ref #12, Salou et al, JEM 2019). Also recent publication analyzed single cell transcriptomics of MAITs and showed there are PLZF^{hi} population at stage2 (Figure 2a, cluster 7a and 7b, Legoux and Lantz, Nat Immunol, 2019, 1244-1255). As the reviewer suggested, we performed direct IL-4 ICS and showed the results in Figure 1B for reviewers, in which we analyzed three different mice and concatenated figure for IL-4 ICS.

3. The comparison of DEGs between Tgd1:Tgd2:Tgd17 and NKT1:NKT2:NKT17 was confusing. If I understood correctly, it shows a significant overlap between lineages such that roughly one fifth of DEGs were shared between lineages. While this suggests some overlap, it also highlights that most (roughly 4/5ths) of the DEGs are actually distinct for these lineages and it could be argued that they are mostly distinct in their DEGs. With this in mind, is it reasonable to argue that they are 'highly similar'? Some overlap is expected, given common markers are used to isolate them such as IL-17 and its associated family members. How would these patterns compare to conventional Th1:Th2:Th17 subsets? Would they be expected to overlap to a similar extent? In some ways, what distinguishes these lineages versus what do they have in common is an equally interesting question.

In Figure S3B, the frequencies of overlapping DEGs are 26% to 46%. For example, number of DEGs between T γ δ 1 and T γ δ 2 cells are 1812 (1330+482) and the frequency of overlapping DEGs with NKT1 vs NKT2 cells is 26% (482/1812x100). When we extracted random genes from γ δ T and NKT cells, the number of overlapping DEGs are about 100 (Figure S3B, below left graph). Overall, the numbers of overlapped DEGs between γ δ T and NKT are average 4-fold higher than those from bootstrapping results (N = 10,000 for each subset pair). We explained this in the manuscript as 'significantly higher than random expectation' (page 4 line 32). We used the term 'highly similar' when we explained about cytokines, receptors and transcription factors (page 5 line 9-10). However, as the reviewer pointed, 'highly' is a subjective term and we rephrased the manuscript.

In addition, we compared the degree of transcriptomic similarity between γ δ T, NKT, Th and ILC subsets using the public RNA-seq data (Stubington et al. Biology Direct 2014 and Gury-BenAri et al. Cell 2016) and the same analysis pipeline. We found that a fraction of overlapped DEGs between γ δ T and NKT were average two-fold higher than those with Th or ILCs (added in Figure S3C), concordant with our previous experiments (Lee et al. Journal of Immunology 2016).

Suggested by the reviewer, we extended transcriptomic comparison not only to validate their similarity but also analyze molecular functions that distinguish or combine these lineages. Analysis of functional enrichment studies is added in Figure 2 for reviewers. However, we could not extract particular meaning of this list, and we did not include them in the main and supplementary figures.

4. I was unable to understand the data and explanation for Fig S4B-C leading to the point that "T γ δ 17i cells exclusively generated IL-17 producing mature T γ δ 17 cells". Also "We also analyzed the signature genes of T γ δ 17i cells and found 349 DEGs to be specifically upregulated (Figure S4D-E),

indicating they *are* unique developmental intermediates of T $\gamma\delta$ 17 cells.” I also do not understand this conclusion has been reached. The 349 DEGs seem to be common to gd1, gd2 and gd17 cells. Some more careful explanation here will help.

Our interpretation of Figure S4B-C was based on the assumption that CD24^{hi} cells are immature and CD24^{low} cells are mature. In Figure S4B, Rorc-GFP positive cells represent current expression of ROR γ t and tdTomato expression indicates current and previous ROR γ t expression. If ROR γ t expression is reversible, there should be tdTomato⁺GFP⁻ cells, which were rarely observed. In Figure S4C, we showed only CD24^{lo} cells express *Il17a*, indicating CD24^{hi} cells are not IL-17 producing cells. To avoid confusion, we edited the manuscript (page 5 line 13) and highlighted in yellow. In Figure S4D, numbers indicated number of genes upregulated in T $\gamma\delta$ 17i cells compared to each population. We found the figure legend is confusing and clarified it.

5. Because NKT0 and NKTp populations are very rare and difficult to isolate, these need to be confirmed as NKT based on TCR α chain expression. It is important to show their TCR sequence because other analyses of these cells have been incorporated in the paper. Were they all Va14Ja18? As Figure 4F showed, NKTp (N1 cluster) had similar frequency of canonical V α 14J α 18 TCRs with that of NKT2 or NKT17 cells. In Figure S6B, we could define 11 NKT0 cells, and TCR sequence was detected in 4 cells of them (Clonotypes 961, 963, 1066 and 1874 in Table S3). Of these 4 cells, three cells had canonical V α 14 J α 18 TCRs and clonotype 1874 had non-canonical TCR. However, we did not include this data in Figure 4F as we had only 4 cells analyzed. These cells were marked as NKT0 cells in Table S3 and explained in page 9 line 26-28.

6. p.6 “and we validated these results by flow cytometry” – no data is shown or referenced.
We are sorry for the confusion. Figure S9E has this data and we highlighted yellow in the manuscript.

7. For the multiplexed scRNAseq, using males for NKT and females for MAIT cells – doesn't this create a confounding variable being the impact of sex difference?
We personally communicated with Dr. Lantz who recently published a paper of single cell RNAseq of MAITs (Legoux and Lantz, Nat Immunol, 2019, 1244-1255). They used mixture of cells from both male and female and when we compared signature gene expression patterns of our data using their signature gene sets, or vice versa, we found no difference (Figure 3 for reviewers).

8. The data in Fig S6 is inadequately explained and confusing. A code for what red and grey colours represent is also required.
We are sorry for the confusion. We rewrote the legend and added color code.

9. This statement “indicating that M2 contains both immature and mature MAIT2 cells that we identified in flow cytometry (Figure 1A).” seems inconsistent with the data in Fig 1A
We found the term ‘immature MAIT2’ is misleading. We deleted this term and rewrote the sentence.

10. The statement “we observed the generation of $\gamma\delta 24+25+$ cells from $\gamma\delta 24+25-$ cells in fetal thymic organ culture experiment (FTOC) (data not shown)” seems like an important and relevant result that should be shown. Also, this approach would be very useful for determining precursor/progeny relationships proposed for MAIT cell subsets and validating the scRNAseq based data for other findings in this paper.

Thank you for the suggestion. We added this data in Figure S11.

11. The statement “we observed a uniformly high level of Ki-67 expression in MAIT cells (data not shown), and cell cycle-regulated genes as well as lineage specific signatures were upregulated during maturation (Figure S10 and Table S4).” is inconsistent with a recent report (Winter et al 2018 Immunology Cell Biology, 97, 190).

We had very similar results with those of above reference in WT, *Cd1d* KO and *Tcrd* KO mice (Figure 4 for reviewers). However, compared to FMO control, stage 1 or 2 cells were still positive for Ki-67 although stage 3 cells expressed higher level of it. We deleted ‘uniformly’ to avoid confusion (page 8 line 21-22).

12. There is a problem with the staining in Fig 7b. Of the Vg9+ T cells, a major population of Vg9 intermediate cells are Vd2. However, when stained with Vd1, this population of Vg9 intermediate cells disappears and all Vg9 cells are Vg9hi. This is reflected by the angled quadrant gates for Vd2 staining suggesting a compensation problem.

We used both Vd1 and Vd2 antibodies as FITC conjugated ones, and dot plots are from different tubes. We carefully looked at the result again and found that all the other samples had same pattern without compensation problem. We concluded that Vd2 staining lowered the staining intensity of Vg9 and, therefore, Vg9 intermediate cells were not present without Vd2 staining. However, Vd2 staining did not change the frequency of Vg9+ cells and we kept the current data.

13. The statement on p11 that “ROR γ t+ NKT, MAIT, and $\gamma\delta$ T cells produced both IFN- γ and IL-17A” is inaccurate. For gd and MAIT cells, it looks like just as many produce IL-17 but not IFN γ . It is also unclear how the quadrant markers were placed, as they move around between populations. This has an impact on the % reported.

Because ROR γ t+ NKT, MAIT and $\gamma\delta$ T cells also express intermediate level of TBET, it is understandable that they simultaneously produce IFN γ and IL17A. IL-17A production seems to be minimal (around 2%), but we observed consistent results compared to TBET^{hi} population, which had zero frequency of IL-17A staining. To clarify the expression of TBET in ROR γ t+ in NKT, MAIT and $\gamma\delta$ T cells, we added the expression of CD45RA+ B cells as negative control for TBET staining. We also had same results when we used FMO control (Figure 5 for reviewer). In cytokine staining, we used same quadrant gating in all cells. IFN γ staining is clearly separated in NKT and MAITs and same gating was applied in $\gamma\delta$ T cells.

14. The final statement in results “human ROR γ t⁺ cells simultaneously express an intermediate level of TBET with ROR γ t.” is problematic because there are no negative controls for this staining, so it is impossible to tell if the Tbet staining is negative, positive or intermediate. How were the regions positioned to determine what was negative or positive?

Please see the answer for Q13.

15. Discussion p12. “The absence of TBET rather suppressed the development of type 17 cells both in MAIT and $\gamma\delta$ T cells and this could be explained by the presence of immature type 1 and type 17 precursors, predicted by the scRNAseq analysis (Figure 6)”. This is confusing and I don’t understand how this point comes from the data.

We tried to explain that the phenotype of \$\gamma\delta\$ T and MAIT cells in TBET deficient mice is different from that of NKT cells due to the presence of developmental intermediates. We rewrote result section in page 11 to clarify our points.

16. A minor point - the word ‘quasi-identical’ (p4) to compare MAIT1 and MAIT17 to NKT1 and NKT17 cells is confusing and seems to be an oxymoron. Identical means exactly the same, quasi means partial but not complete. Why not just ‘similar’.

Thank you for the suggestion. We changed the term.

Reviewer #3 (NKT, NK, MAIT) (Remarks to the Author):

In this manuscript Lee et al. examine the development of the most common “innate” T lymphocytes together, namely the iNKT cells, the MAIT cells and the $\gamma\delta$ T cells. Using multicolor flow cytometry as well as single cell RNA-Seq coupled with TCR sequencing, they show that for each of these lymphocyte populations, similar effector subsets exist. This mostly confirms what was already known in the field, with the exception of MAIT2-like cells that had not been previously described. Furthermore, through the use of V γ -specific knock-out mice, the authors argues that the specific nature of the $\gamma\delta$ TCR plays no role in the subset specification.

There are a lot of data in the manuscript and it is sometimes difficult to keep track as to what the actual message is. Although the experiments are properly performed and controlled, there are a few points that need clarification, as they do not conform to what would have been expected from previous literature. In addition, I am unsure that the TCR sequencing data have enough power to make any argument as to whether the exact nature of a given TCR might play a role, or not, in the subset specification of any of these lymphocyte populations. In conventional BALB/c mice, most of $\gamma\delta^+$ Ror γ t $^+$ cells are also V γ 4 $^+$. The authors show that in the V γ 4/6 KO mice, the proportion of Ror γ t $^+$ $\gamma\delta$ T cells do not change compared to control mice but that the Ror γ t $^+$ cells have now been replaced by V γ 1/V δ 6.3 $^+$ cells. This finding is used to conclude that $\gamma\delta$ TCR specificity is irrelevant to subset specification. However, this conclusion does not seem entirely supported by the data showed in Fig 5c, where the proportion of V γ 1/V δ 6.3 $^+$ amongst T $\gamma\delta$ 17 cells in WT vs V γ 4/6 KO actually does not change (66% vs 71%). From these data, it would appear that some V γ 1/V δ 6.3 $^+$ T cells are in fact T $\gamma\delta$ 17 in wildtype animals and that these cells could be expanding and filling the niche when V γ 4 $^+$ cells are absent in the KO mice and not that there is a diversion of the development of V γ 1/V δ 6.3 $^+$ T cells from a T $\gamma\delta$ 2 to a T $\gamma\delta$ 17 phenotype.

Finally, the way the data are presented and structured suggest that all developing $\gamma\delta$ T cells commit to one of three functionally different subsets exemplified by the expression of either Ror γ t, PLZF or Tbet, similarly to MAIT and iNKT cells. While this certainly happens to some $\gamma\delta$ T cells, a significant proportion of mature (as defined the lack of expression of CD24) $\gamma\delta$ T cells in the mouse thymus do not express any of these transcription factors. These cells are not apparent in any of the transcriptional analyses performed. How are these finding reconciled with previous literature where non-innate programmed $\gamma\delta$ T cells have been defined (see Bonneville et al. 2010 for review).

If the functional differentiation of $\gamma\delta$ T cells is determined by their TCRs, identical clonotypes should designate the same function. As also reviewed in Bonneville et al. (*Nat Rev Immunol*, 2010) and in many other papers, the classification of $\gamma\delta$ T cells was based on their TCR γ chain usage and V γ 1/V δ 6.3 $^+$ $\gamma\delta$ T cells were considered as PLZF expressing IL-4 producing subset. However, we showed they expanded in the absence of V γ 4/6 and became T $\gamma\delta$ 17 cells. This is not simple niche filling, however, in that there is a simultaneous expansion of V γ 1 $^+$ T $\gamma\delta$ 1 cells in V γ 4/6 deficient mice (Figure 5C graph). Also we found that proliferative capacity in terms of Ki-67 expression was not

different between WT and $V\gamma 4/6$ deficient $V\gamma 1^+ V\delta 6^+ ROR\gamma t^+ T\gamma\delta 17$ cells (Figure 6 for reviewer). Also it was not proportional expansion of minor population of pre-existing $T\gamma\delta 17$ cells as seen in updated Figure 5C pie chart, which showed only $V\gamma 1^+$ cells expanded while $V\gamma 3^+$ or $V\gamma 7^+$ cells did not. These findings argue that this is a simple niche filling.

We also acknowledged that there are non-innate programmed $CD24^{low} \gamma\delta$ T cells, which do not express any transcription factors in the thymus and periphery. As our main research topic is about *innate differentiation* of NKT, MAIT and $\gamma\delta$ T cells in the thymus, we added the term 'in the thymus' in the title and abstract and discussed about this (page 14 line 16-26).

Comments:

1) It is interesting that the authors identify a previously unknown subset of MAIT cells. Do these MAIT2 cells also actively secrete IL-4 at steady-state much like iNKT2 and $T\gamma\delta 2$ (Figure 2C would argue that they do not)? If not, what is unique to their transcriptional program that they do not produce IL-4? This is perhaps evident even in their expression of IL17RB as they do not express high levels of this cytokine receptor and consequently, may not have received the proper cues to differentiate further as iNKT2 and $T\gamma\delta 2$ cells may have?

This is very interesting point. MAIT2 cells produced IL-4 when they were stimulated as seen in human CD2 expression (Figure S1B) or intracellular IL-4 staining (Q2 of reviewer 2 and Figure 1 for reviewer). However, as the reviewer pointed out, there were no IL-4 transcripts in MAIT2 cells at the steady state (Figure 2C) and human CD2 was not expressed in KN2 mice without stimulation. In our previous research (Lee et al, 2013 Nat Immunol, ref #8) we showed that homeostatic IL-4 production from NKT2 cells require CD1d expression in the medulla and recent publication (Wang and Hogquist, PNAS 2019) showed myeloid cells provide it. We speculate that there are lack of MR1 expression in the thymic medulla that can stimulate MAIT2 cells.

2) It would be important to conduct PCA on all the 18000 genes for all subsets of each of the three T cell populations to see how all of them separate, instead of just the 120 DEGs between iNKT and $\gamma\delta$ subsets as conducted in Figure 1E. By selecting the genes that are common to the cells types, it is perhaps not surprising that they would then cluster together in PCA analysis. Because the expression of several of these genes are likely consequential to the expression of the "master" transcription factors that define the various subsets, it is not surprising that the cells share a similar transcriptional profile. What is perhaps more relevant is what transcriptional profile defines the unique identity of each of these lymphocyte populations?

When we generated PCA plot using six different subsets from iNKT and $\gamma\delta$ T cells, PC2 responsible for 27% of variation grouped analogous subset of NKT and $\gamma\delta$ T cells, but PC1 responsible for 46% of variation separated these cell types (Figure 7 for reviewers). Although we used same methodology for library creation and analysis platform, we could not determine if PC1 is real biological difference or batch effect. This is why we conducted single cell analysis using all three types of innate T cells

simultaneously. Also according to the suggestion of reviewer #2 question 3, we defined commonly or uniquely enriched pathways of NKT or $\gamma\delta$ T cell (Figure 2 for reviewers)

3) Although the authors claim in Figure S4B-C that they are conducted fate mapping studies to conclude that the T $\gamma\delta$ 17i cells give rise to the T $\gamma\delta$ 17 cells, this is an inappropriate statement. The FTOC experiments conducted later in the submission certainly suggest a precursor-product relationship but simply marking these cells using reporters is insufficient to conclude that the immature cells generate the mature population. This statement here should be removed and added to the section with the FTOC figure.

Please see our comment on question #4 of reviewer #2. As the reviewers suggested, we edited the manuscript and moved this part in FTOC section.

4) Why do the authors recover such few iNKT1 cells in their scRNA-seq experiment while they sorted total iNKT cells from the thymus of 7 weeks old BALB/c mice? Also, the IFN γ signature in iNKT1 and T $\gamma\delta$ 1 cells does not appear to be shared by the majority of the cells belonging to these subsets unlike that observed in MAIT1 cells. Can the authors identify any transcriptional signature leading to this difference?

In our single cell analysis, frequency of NKT1 cells was 6.3% (210 / 3285 cells, Table S1). When we analyzed the frequency of NKT1 cells by FACS, we also observed similar results (Figure 8 for reviewer). Therefore, NKT1 frequency in single cell analysis is compatible with actual composition of NKT1 cells. As for $\gamma\delta$ T cells, we found that T $\gamma\delta$ 1 cells were under-represented in our scRNAseq analysis for uncertain reason. We could define only 76 cells in G7-2 and among them, 11 cells expressing high level of Tbx21 and CXCR3 were co-localized with NKT1 cells.

5) As mentioned earlier, it is unclear whether the experiments depicted in Figure 5C actually support the authors' claim. There is already a similar proportion of V γ 1V δ 6.3+ cells that are Ror γ t+ in WT mice. It is possible that these cells simply expand in the V γ 4/6 KO mice due to lack of competition and not that other V γ 1 cells are diverted to fill the niche.

Please see the answer for main question.

6) How do the authors account for the expression of CD122 in the immature populations before the expression of T-bet? T-box proteins have been previously shown to drive the expression of CD122 but that is not the case here.

Using *Tbx21*-GFP reporter mice, we confirmed again that CD122⁺ CD24⁺ $\gamma\delta$ T cells do not express TBET (Figure 9 for reviewer). Part of these cells were Eomes⁺, suggesting it may play some role. We found small number of T $\gamma\delta$ 1 cells present in *Tbx21* KO mice, which expressed CD122 and Eomes (Figure 6C). We speculate that CD122 might serve as IL-15 receptor, which previously found to induce TBET in NKT cells (Gordy and Joyce, J Immunol, 2011). However, further investigations are required to find out upstream regulator of CD122 other than T-box proteins.

7) The development of iNKT and MAIT cells is thought to occur through positive selection by CD1d and MR1 expressed by DP thymocytes in a SLAM/SAP/Fyn-dependent manner (Wang and Hogquist, 2018). Such process is thought to uniquely impart the innate features that are acquired by the iNKT and the MAIT cells. Since the authors propose that $\gamma\delta$ T cells also end up with a very similar transcriptional profile to iNKT and MAIT cells, it would perhaps be informative to discuss whether similar developmental pathways might also be involved in the development of $\gamma\delta$ subsets.

Thank you for the suggestion. We added this in discussion (page 14 line 16-26).

Reviewers' comments:

Reviewer #1 (Remarks to the Author):

The authors have addressed the majority of my comments. The manuscript is now suitable for publication.

Reviewer #2 (Remarks to the Author):

The authors have attempted to address some concerns which is appreciated but unfortunately some concerns remain. The study is potentially important because it contains a number of new findings such as an abundant population of non-canonical MAIT cells and a population of IL4 producing MAIT cells exists in thymus and develops as a separate lineage but I am not yet convinced about the data supporting these claims. I also found that many of the responses to my original comments are unclear.

1. Some of the confusion from point 1 of my first review comes from the word 'non-canonical' to describe MAIT and NKT cells with different TCRb. Canonical normally relates to TCRa for these cells which is normally invariant for V and J genes. NKT and MAIT cells are well known to use diverse Vb and Jb TCR genes with some bias towards particular Vb genes, so it is confusing to refer to MAIT and NKT cells with different TCRb genes as non-canonical. Nonetheless, there were 222 (out of 1760) non-canonical TCRa chains detected in MAIT cells. At 13%, this is still unexpected and not a trivial result because it raises the question of whether these really are MAIT cells. If not then it is a problem to include them in a study of MAIT cell development, eg it may explain some of the DRGs observed and some other phenotypes in this paper. In the DRG comparison (Supp Fig 13b and c) it is these non-canonical TCRa cells that should be compared at each stage of MAIT cell development, otherwise they are outnumbered by MAIT cells with different TCRb genes which also are described as non-canonical. The authors state in the rebuttal that 'even if we exclude these 89 cells, we still have the same conclusion'. This data is not included and, in my view, this is important to show because then we can be sure that only MAIT cells are being compared at each stage. For example, are these contributing to the IL4 production in the M2 population?

2. Regarding the non-canonical TCRa MAIT cells – are they diverse in CDR3 length? TRAV or TRAJ usage? Are they PLZF+? Are they 5OPRU specific?

3. My previous comment that MAIT2 cells were not encompassed within stage 2 remains a concern that is exacerbated by the extra data provided for reviewers. As seen in Supp Fig 1, it looks like many (no % provided) of the MAIT2 cells have the same amount of CD44 as IFNg+ MAIT cells and these cells are CD44+ stage 3. A problem with the CD44 gating can be seen in Fig 1A for reviewers. The blue stage 2 gate captures many cells from the left edge of the CD44+ stage 3 population. Therefore, the stage 2 gate includes CD44-/lo (stage 2 cells) and CD44+ (stage 3 cells). This gate problem could be impacting on many aspects of this paper and needs to be carefully reassessed throughout all figures in the study.

4. Unfortunately, the IL4 staining provided in Figure 1B for reviewers is not reassuring. IL4+ MAIT2 cells are represented by 1 or 2 dots per mouse. In Figure 1B for reviewers, individual mouse 2, there are 3 IL4+ dots for the blue population of MAIT17 cells compared to only 2 less bright dots for the red MAIT2 population. The concatenated file from 3 mice shows only 5 or 6 IL4+ dots in the MAIT2 population which also illustrates the problem. As mentioned previously, others have been unable to find MAIT2 cells so this is not a trivial point and requires a more convincing demonstration. If the authors wish to make this point using the data shown with 5 or 6 IL4+ dots, this would need to include FMO isotype controls but even then it would be very difficult to believe with so few IL4 producing cells.

5. The new data provided in Supp fig 9 showing CD24+CD25+ cells arising from CD24+CD25- cells is

still confusing. Frequency of CD25+ cells is the same 22% regardless of which population was added to the culture. What was the initial sort purity like? The legend says the data is from 3 experiments. What does that mean?

6. The response to my question 13 from original review is inadequate as it does not address my comment. Also the quadrant gates are not in the same position between populations in figure 7C. They are clearly in different positions!

7. I still do not understand the point being made in my original point 12 about TBET KO mice. Also, related to this point I did not notice last time but the figure legends do not describe how many experiments were performed for each figure. This is important because in some cases when the number of mice is very low and differences not very clear such as Figures 5 and 6 where n=3 mice are shown, this might be only one experiment, which is not sufficient. All experiments should be performed at least twice.

8. I don't understand the explanation about Ki67 results in response to my original review as point 11. In the Winter et al paper, stage 1 MAIT cells are described as barely proliferating. This is inconsistent with the statement "we observed a uniformly high level of Ki-67 expression in MAIT cells". Data is provided in Figure 5 for reviewers but it is very difficult to assess the data in Reviewer Figure 5 because no percentages or legend is provided and many stage 1 cells look like Ki67+ in contrast with the Winter et al paper.

9. All data provided for reviewers such as IL4 cytokine staining and Ki67 staining, should really be included in the actual paper as it is likely to also be relevant to the general readership.

Reviewer #3 (Remarks to the Author):

This resubmission of the manuscript entitled "transcriptional landscapes of innate T cells at the single cell level reveals a shared lineage differentiation process regulated by clonal selection and proliferation in the thymus" by Lee et al. examined by flow cytometry and sc-RNA-seq the developmental steps undertaken by NKT, MAIT and gd T cells in the mouse thymus. The message of the paper is still complicated and no effort has been made within the manuscript to try to clarify some of the statements. What does the sentence "their lineage fate is determined by stochastically by clonal selection" actually mean?

Furthermore, the new results presented do not adequately address the previous comments. For example, whether MAIT2 cells secrete IL-4 or not remains dubious. The new results presented are hardly believable and at best would suggest that a tiny fraction of MAIT2 is capable of IL-4 secretion. Because capacity of IL-4 secretion is essentially used to discriminate a so-called NKTp from NKT2 cells, this is a rather important point. Along these lines, it would be very informative to display on the UMAP the actual expression of the genes that characterize the signature of the NKTp cells, as described in Lee et al. 2016, as well as their cycling status. As stated before these cells should be "less transcriptionally active" and express Hes1, Bex1, Taf9 etc... I realize that a signature score is currently displayed but this is not particularly robust (Fig S7) and is a new signature of NKTp (based on the N1 cluster) is required then it should be re-defined. Finally, the UMAP defining NKT cell clusters is not particularly supportive of the author's message and previous conclusions. While NKT17 might appear to be developing from NKTp, the so-called NKT2 cells (what defines these 4 clusters?) appear to be on the path to NKT1 cells, not a final stage of development. Because NKT cells are used as a roadmap to explain MAIT cell and gd T cell development, this is concerning. It is also unclear why NKT and MAIT cells were sorted from the thymus of 6 week old mice but gd T cells from the thymus of 5 day old mice?

The idea proposed here that these three populations compete for a "developmental" niche needs further explanation. What are these cells competing for exactly? The proportion of MAIT cells within the mouse thymus was recently reported to be dependent upon the microbiota. Could it be that in

absence of gd T cells, the microbiota is perturbed and leads to an increase of MAIT cells? What is the "developmental niche" then?

The idea that the TCR specificity of any of these populations has no role in subset differentiation is based on the share of a few gd clonotypes amongst different subsets. This occurs albeit strong biases for each subset. Therefore, does it represent the rule or an exception? MAIT cells did not display such an overlap, while NKT cells did show some clonotype overlaps between subsets. Does this mean that the repertoire of NKT cells is actually oligoclonal with expansions of particular clonotypes? This is rather surprising in light of previous results with deeper sequencing showing that the NKT repertoire is extremely polyclonal.

The identical proportion of Ki-67+ in Vg1Vd6+ rorgt+ cells in wt and Vg4/6 Ko mice does not exclude the possibility of niche filling by a pre-existing population of such cells, since its precursor might be proliferating, not the end-product.

Further comments:

The naïve MAIT cells identified in Legoux et al. are PLZFneg and therefore cannot be similar to the M2 cluster identified here as stated in the text.

We thank all three reviewers for their constructive comments and the Editor for his helpful guidance through the process of revising this manuscript, which helped us a lot to improve our manuscript. We have added experiments and revised our manuscript according to the reviewers' suggestions and **highlighted them in yellow**.

Response to Reviewer #1

The authors have addressed the majority of my comments. The manuscript is now suitable for publication.

Response: We are very happy that our revision could finally convince this reviewer.

Response to Reviewer #2

Comment 1: The authors have attempted to address some concerns which is appreciated but unfortunately some concerns remain. The study is potentially important because it contains a number of new findings such as an abundant population of non-canonical MAIT cells and a population of IL4 producing MAIT cells exists in thymus and develops as a separate lineage but I am not yet convinced about the data supporting these claims. I also found that many of the responses to my original comments are unclear.

Response: We thank the reviewer for her/his comments and the endorsement of our work. The reviewer criticized that our manuscript made a number of claims, which were not well supported by data. We hope that we have succeeded in rectifying this issue now. Most of all, we would like to emphasize that **the presence of PLZF^{hi} MAIT cells had also been shown in two previous independent papers**. First, Koey et al showed there are PLZF^{hi} cells in stage 2 cells (small green bump, **Figure 1 for reviewer**). Second, Legoux et al performed scRNA-seq on MAIT cells and showed there are naïve-like MAIT populations (cluster 7), some of which expressed PLZF (Figure 1 for reviewer). They designated them as 'naïve-like' because they were neither MAIT1 nor MAIT17 cells and many of them belonged to stage 2. CD44 was variably expressed on these cell as shown in Figure 1 for reviewer. This is related to **Comment 4**, and we also found MAIT2 cells had a wide range of CD44 expression, which were 'relatively' lower than that of MAIT1 and MAIT17 cells. **We fully agree with the reviewer that CD44 levels of MAIT2 cells overlap with those of MAIT1 and MAIT17 cells**. As the reviewer suggested, we re-analyzed the distribution of MAIT2 cells at S2 and S3 (**Figure 2 for reviewer**), which showed PLZF^{hi} MAIT2 cells are enriched at stage2 (22.9%)

and also some of stage 3 cells (6.9%) are MAIT2. In previous studies, Salou et al (Figure 2 for reviewer, bottom) mentioned that the PLZF^{hi} population was not seen among CD44^{hi} MAITs but they showed a few PLZF^{hi} dots among CD44^{hi} MAITs (6.02% in BLAB/c or 1.6% in B6.Cast mice) in their representative FACS plots. Although it is also hard to convince whether these 6.02 or 1.6% represent real events, Legoux and Salou's papers are from the same laboratory (Dr Lantz), and we think all four papers (Koey et al, Legoux et al, Salou et al and ours) have consistent results that showed PLZF^{hi} MAIT cells are present and they are 'enriched' at stage 2 and some of them belong to stage 3 (although most of stage 3 cells are MAIT1 or MAIT17 cells). Overall, we found **it is misleading to say MAIT2 cells are mostly at stage 2**, and clarified the manuscript and highlighted in yellow as "PLZF^{hi} type2 subsets of MAIT and $\gamma\delta$ T cells were relatively enriched at stage 2 compared to type 1 or type 17 cells (Figure S1A). This finding is consistent with a recent report that showed the presence of naïve-like MAIT cells expressing PLZF¹³." (page 4, line 10-12).

Comment 2: Some of the confusion from point 1 of my first review comes from the word 'non-canonical' to describe MAIT and NKT cells with different TCR β . Canonical normally relates to TCR α for these cells which is normally invariant for V and J genes. NKT and MAIT cells are well known to use diverse V β and J β TCR genes with some bias towards particular V β genes, so it is confusing to refer to MAIT and NKT cells with different TCR β genes as non-canonical. Nonetheless, there were 222 (out of 1760) non-canonical TCR α chains detected in MAIT cells. At 13%, this is still unexpected and not a trivial result because it raises the question of whether these really are MAIT cells. If not then it is a problem to include them in a study of MAIT cell development, eg it may explain some of the DRGs observed and some other phenotypes in this paper. In the DRG comparison (Supp Fig 13b and c) it is these non-canonical TCR α cells that should be compared at each stage of MAIT cell development, otherwise they are outnumbered by MAIT cells with different TCR β genes which also are described as non-canonical. The authors state in the rebuttal that 'even if we exclude these 89 cells, we still have the same conclusion'. This data isn't included and, in my view, this is important to show because then we can be sure that only MAIT cells are being compared at each stage. For example, are these contributing to the IL4 production in the M2 population?

Response: We agree with the reviewer and apologize for this confusion. We have clarified this issue by **defining the term 'canonical/non-canonical' to indicate V α and J α chains, and 'oligoclonal/non-oligoclonal' to indicate V β chains (page 8, line 5-6)**. When we

compared transcriptional signatures between canonical and non-canonical MAITs using a new definition of “canonical/non-canonical MAITs”, we again found no differences between them in all stages, except MAIT1 signature score at (Figure S18B). However, the only DEG was their TCR itself (Trav1, Figure S18C) and they are very different from DP contaminated cells (Figure S18D). In Figure 4G, we re-analyzed our scRNA-seq data for MAITs by including all non-canonical or non-oligoclonal (left), non-canonical (middle), and non-canonical MAITs except 89 cells (right), which showed a similar trend of decreasing proportion of non-canonical/non-oligoclonal MAITs over the course of developmental stages. As we have mentioned during the last round of revision, **non-canonical MAITs except 89 cells were paired with oligoclonal repertoire of TRBV chains specific to MAIT cells**, which makes it extremely unlikely that these non-canonical MAIT cells are the result of random contamination during sorting. In Figure 4G, **we showed non-canonical MAITs occupy 10-15% of M2 cells**. Because not all M2 cells produced IL4 and it is very difficult to evaluate if MAITs with non-canonical TCRs contribute to actual IL4 production after stimulation. However, based on similar transcriptional signatures between canonical and non-canonical MAITs at M2 (Figure S18B-C), we think canonical and non-canonical MAIT cells have little functional differences.

Comment 3: Regarding the non-canonical TCR α MAIT cells – are they diverse in CDR3 length? TRAV or TRAJ usage? Are they PLZF+? Are they 5OPRU specific?

Response: In Figure 4A, we displayed CDR3 length distributions of TCR α and TCR β chains, which showed **non-canonical MAITs have wider ranges of CDR3 α length distribution, indicating they are more diverse in CDR3 length**. In Figure S9E, we showed lists of non-canonical TRAV and TRAJ genes with distribution of most frequent ones in the UMAP. Also, as shown in Figure S18, **there are minimal differences in their transcriptomes (including PLZF) between canonical and non-canonical MAITs**. Regarding the 5OPRU specificity, we showed it is extremely unlikely that these non-canonical or non-oligoclonal cells are the result of random contamination as they are paired with oligoclonal TCR β or canonical TCR α respectively except 89 cells (See **detailed explanation in our response to comment 1 during the last round of revision**). We also showed that without these 89 cells, our conclusion that non-canonical usage is higher in S1 and S2 than in S3 MAITs does not change (Figure 4G). The best way to prove the 5-OPRU specificity is to analyze tetramer reactivity after we transfect these TCRs in the cell line. This analysis, however, is not

technically possible to perform within three months of revision period.

Comment 4: My previous comment that MAIT2 cells were not encompassed within stage 2 remains a concern that is exacerbated by the extra data provided for reviewers. As seen in Supp Fig 1, it looks like many (no % provided) of the MAIT2 cells have the same amount of CD44 as IFN γ + MAIT cells and these cells are CD44+ stage 3. A problem with the CD44 gating can be seen in Fig 1A for reviewers. The blue stage 2 gate captures many cells from the left edge of the CD44+ stage 3 population. Therefore, the stage 2 gate includes CD44-/lo (stage 2 cells) and CD44+ (stage 3 cells). This gate problem could be impacting on many aspects of this paper and needs to be carefully reassessed throughout all figures in the study.

Response: We fully agree that MAIT2 cells are not mostly at stage 2. We have clarified this issue in our response to Comment 1 of Reviewer#2 and Figure 2 for reviewers.

Comment 5: Unfortunately, the IL4 staining provided in Figure 1B for reviewers is not reassuring. IL4+ MAIT2 cells are represented by 1 or 2 dots per mouse. In Figure 1B for reviewers, individual mouse 2, there are 3 IL4+ dots for the blue population of MAIT17 cells compared to only 2 less bright dots for the red MAIT2 population. The concatenated file from 3 mice shows only 5 or 6 IL4+ dots in the MAIT2 population which also illustrates the problem. As mentioned previously, others have been unable to find MAIT2 cells so this is not a trivial point and requires a more convincing demonstration. If the authors wish to make this point using the data shown with 5 or 6 IL4+ dots, this would need to include FMO isotype controls but even then it would be very difficult to believe with so few IL4 producing cells.

Response: We agree that 1 or 2 dots per mouse are not convincing to make a conclusion. As we addressed in our response to Comment 1, we think the **presence of PLZF^{hi} MAIT cells has been shown by two independent research groups** and we additionally showed they are IL4 producing subset similar to NKT2 or $\gamma\delta$ T cells. The calculated number of MAIT2 cells per thymus is less than 50 per mouse, and only about 10-20 cells are analyzable after enrichment. Even after we combine 5 mice, we can obtain 50-100 MAIT2 cells, and five to six dots correspond to 5-10%. In KN2 mice, the frequency of huCD2+ MAIT2 cells was 10% (Figure S1B) and probably, this is an experimental limit. To further convince our experimental results, however, we repeated the experiments using multiple mice (9 mice) with FMO control as the reviewer suggested and added more convincing data in the Figure S1C. Furthermore, as in **comment 2 of reviewer 3**, we checked if M2 cluster is like NKTp

cells rather than NKT2 cells. In Figure S13, however, we showed M2 cluster is like N3, which is one of subclusters of NKT2 cells. **We newly added 2nd paragraph in discussion (page 13, line 18-27), which explained the possibility that MAIT2 cells are not fully differentiated IL4 producing cells.**

Comment 6: The new data provided in Supp fig 9 showing CD24+CD25+ cells arising from CD24+CD25- cells is still confusing. Frequency of CD25+ cells is the same 22% regardless of which population was added to the culture. What was the initial sort purity like? The legend says the data is from 3 experiments. What does that mean?

Response: According to the reviewer's suggestion, we added the results of repeated experiments and sorting purity in Figure S14. The purities of sorted cells were more than 95% and we displayed pooled data of 3 independent experiments in the graph.

Comment 7: The response to my question 13 from original review is inadequate as it does not address my comment. Also the quadrant gates are not in the same position between populations in figure 7C. They are clearly in different positions!

Response: We apologized to the reviewer for our misunderstanding. Question 13 of the last round of revision was "The statement on p11 that "ROR γ t+ NKT, MAIT, and $\gamma\delta$ T cells produced both IFN- γ and IL-17A" is inaccurate. For gd and MAIT cells, it looks like just as many produce IL-17 but not IFN γ ". We understood this as you pointed out that there are as many IFN γ producing cells in ROR γ t+ cells as in TBET+ cells, but only a few cells produce IL17A. Our response was that because ROR γ t+ cells simultaneously express TBET, it is understandable that they also produce IFN γ and although IL17a production seems to be minimal, we obtained similar results from repetitive experiments.

As for the gating issue, we misunderstood your points, and we corrected again it and recalculated the frequencies of cells in each quadrant together with unstimulated control plots in Figure 7C.

Comment 8: I still do not understand the point being made in my original point 12 about TBET KO mice. Also, related to this point I did not notice last time but the figure legends do not describe how many experiments were performed for each figure. This is important because in some cases when the number of mice is very low and differences not very clear such as Figures 5 and 6 where n=3 mice are shown, this might be only one experiment, which is not sufficient. All experiments should be performed at least twice.

Response: We apologize for the very poor job we did in the previous version in interpreting the results of Tbet KO mice. In the previous version of our manuscript, we could not interpret the result of Tbet deficient mice properly, which had rather decreased number of MAIT17 and T γ δ 17 cells in the thymus compared to WT control. This result was inconsistent with our hypothesis that abortive differentiation into type1 lineage would facilitate the development of type2 and type17 cells. We thought there might be some cage specific effects or some minor genetic mutations accumulated in Tbet KO mice and **out-bred Tbet KO mice with WT B6 control** (we purchased them from Jax). We obtained Tbet het mice and crossed them again with each other to generate Tbet KO and WT control mice among the littermates. In this strictly controlled experiment, we found MAIT17 and T γ δ 17 cells were increased compared to their littermate WT controls. We changed Figure 6C and 6D and re-described results and discussion and highlighted them in yellow.

Regarding the number of experimental replicates, we always used representative facs plots from more than 3 independent experiments. We have clarified the number of experiments repeated and whether graphs contain one set or pooled data from multiple experiments in the figure legends.

Comment 9: I dont understand the explanation about Ki67 results in response to my original review as point 11. In the Winter et al paper, stage 1 MAIT cells are described as barely proliferating. This is inconsistent with the statement “we observed a uniformly high level of Ki-67 expression in MAIT cells”. Data is provided in Figure 5 for reviewers but it is very difficult to assess the data in Reviewer Figure 5 because no percentages or legend is provided and many stage 1 cells look like Ki67+ in contrasts with the Winter et al paper.

Response: We apologized to the reviewer for our misleading statements concerning Ki-67 expression in MAITs. During the last round of revision, we deleted the term ‘uniformly’ because Ki-67 expression levels were not uniform among subsets. According to Comment 10, we included Figure S16 for our Ki-67 expression analysis. Previously, Winter et al showed low Ki-67 expression levels in CD24+ MAIT0 cells, but they did not compare them using FMO or isotype control. We obtained identical Ki-67 expression patterns with those of Winter et al, but when we compared them with Ki-67 levels of S1 cells with isotype control, we found some of them were still positive. However, we do not know how much proliferative difference would be made with this low level of Ki-67 expressions. Therefore, to avoid any confusion with previous results and clarify our messages, we changed the description in the

main text as ‘However, this is unlikely as we and others observed a high level of Ki-67 expressions at least in stage 2 and 3 MAIT cells’. (page 8, line 29-30).

Comment 10: All data provided for reviewers such as IL4 cytokine staining and Ki67 staining, should really be included in the actual paper as it is likely to also be relevant to the general readership.

Response: Thank you for your suggestion. We have now moved these figures to Figure S1C, S4, S9E, S11, S13, S15 and S16.

Response to Reviewer #3

Comment 1: This resubmission of the manuscript entitled “transcriptional landscapes of innate T cells at the single cell level reveals a shared lineage differentiation process regulated by clonal selection and proliferation in the thymus” by Lee et al. examined by flow cytometry and sc-RNA-seq the developmental steps undertaken by NKT, MAIT and gd T cells in the mouse thymus. The message of the paper is still complicated and no effort has been made within the manuscript to try to clarify some of the statements. What does the sentence “their lineage fate is determined by stochastically by clonal selection” actually mean?

Response: We apologize to the reviewer for our complicated statements concerning lineage fate decision of innate T cells. We hope that in both the revised manuscript and the following answers to her/his questions we can convince the reviewer of the soundness of our core results.

As the reviewer #2 also pointed out, we found there were some misleading statements in our first revised manuscript and we did our best to clarify our messages and make our data more convincing in this second revised manuscript. **Please see our responses to Comment 1 and 2 of the reviewer #2 as we have revised our manuscript according to them.**

In this manuscript, we tried to show that after positive selection, innate T cells such as NKTs, MAITs and $\gamma\delta$ T cells also go through the process of clonal selection and proliferation. In $T\gamma\delta 17$ cells, TCR diversity decreases when they mature from immature progenitors, and non-canonical MAIT cells decrease after positive selection. Therefore, although we do not fully understand the mechanism, we concluded that some clonotypes are selected for the final maturation after positive selection. Unfortunately, we could not prove this in every subset (e.g. $T\gamma\delta 1$ or $T\gamma\delta 2$ cells due to their limited numbers in our single cell analysis) and could not generalize this in NKT cells because their CD24^{hi} immature

precursors were very few. However, we think our findings that innate T cells, such as T γ δ 17 and MAIT cells are clonally selected before their final maturation is worth of recognition in the field. We deleted the term ‘stochastically’ as we did not provide enough evidence whether these are really a randomly determined process.

Comment 2: Furthermore, the new results presented do not adequately address the previous comments. For example, whether MAIT2 cells secrete IL-4 or not remains dubious. The new results presented are hardly believable and at best would suggest that a tiny fraction of MAIT2 is capable of IL-4 secretion. Because capacity of IL-4 secretion is essentially used to discriminate a so-called NKTp from NKT2 cells, this is a rather important point. Along these lines, it would be very informative to display on the UMAP the actual expression of the genes that characterize the signature of the NKTp cells, as described in Lee et al. 2016, as well as their cycling status. As stated before these cells should be “less transcriptionally active” and express Hes1, Bex1, Taf9 etc... I realize that a signature score is currently displayed but this is not particularly robust (Fig S7) and is a new signature of NKTp (based on the N1 cluster) is required then it should be re-defined.

Response: This is a very insightful comment and we should have considered this point in our initial data interpretation. As the reviewer suggested, it is possible that MAIT2 cells correspond to the NKTp population, which does not produce IL4 but gives rise to others. We addressed this issue in newly added **Figure S16 and Figure 3 for reviewers**. In **Figure 3A for reviewers**, we showed the expression of 9 NKTp unique genes (out of 45 genes from 2016 Lee et al, JI paper) that highly expressed in N1 cluster in the UMAP of NKTs. Although not all nine genes are visually impressive, genes such as Ki67, Rad51, PcnA and Mcm3 were unique to N1 cluster. **Figure 3B for reviewers** shows the overall signature scores of NKTp (defined from 2016 Lee et al, JI) and N1 (defined in our single cell analysis) unique genes in each NKT cluster, which showed NKTp and N1 are indeed similar populations. Also, **Figure 3C for reviewers** shows N1 cluster has highest fraction of cells in S phase. In newly added **Figure S13**, we showed **N1 (NKTp) and N3 (NKT2) signature genes are highly expressed in M6 (MAIT17i) and M2 (MAIT2) respectively, indicating MAIT2 cells are more likely NKT2 cells**. We agree with the reviewer’s point that phenotype of MAIT2 cells are not fully compatible with NKT2 cells, and **we mentioned this in the result section (page 7, line 17-18) and added 2nd paragraph in discussion section (page 13, line 18-27) to discuss about the possibility that PLZF^{hi} MAIT cells are not terminally differentiated and in between**

NKTp and NKT2 cells.

In an effort to clarify whether MAIT2 (M2) cells produce IL4 upon activation, we repeated the experiments with FMO control as suggested by reviewer #2 and included the result in Figure S1C, which showed more convincing pattern of IL4 secretion from PLZF^{hi} MAIT2 cells. We added detailed explanations about our IL4 ICS experiments in our response to **comment 5 of reviewer #2**.

Comment 3: Finally, the UMAP defining NKT cell clusters is not particularly supportive of the author's message and previous conclusions. While NKT17 might appear to be developing from NKTp, the so-called NKT2 cells (what defines these 4 clusters?) appear to be on the path to NKT1 cells, not a final stage of development. Because NKT cells are used as a roadmap to explain MAIT cell and gd T cell development, this is concerning. It is also unclear why NKT and MAIT cells were sorted from the thymus of 6 week old mice but gd T cells from the thymus of 5 day old mice?

Response: We thank the reviewer for this point. Taking first the issue of the UMAP of NKTs, it should be noted that we did not draw any conclusions on the precursor-progeny relationships between NKT cell clusters from their UMAP in Figure 3. Even though UMAP is a popular dimensionality reduction method known to arrange cell clusters along differentiation trajectories, its accuracy is largely determined by sampling enough number of transient cells constructing a differentiation continuum of differentiation trajectories. In our previous experiments, we showed intrathymically transferred NKT2 cells do not give rise to NKT1 cells (Lee et al, 2013 Nat Immunol) and agree with the reviewer that the UMAP of NKTs is not consistent with this finding. The possible explanation of this discrepancy is that the developmental intermediates between NKTp and NKT1 cells were not sampled due to their low frequency. In addition, NKT1 cells are a long-lived population expressing CD122, and developmental intermediate of NKT1 cells at the given time point could not well be represented in our single cell analysis.

Regarding the number of cell clusters of NKT2 cells, the optimal number of clusters is a subjective concept that is largely dependent on the definition of cell types and algorithm-specific resolution parameter. In addition, the Louvain algorithm, which is our clustering method used in this analysis and one of the popular clustering methods in the single-cell field, gives different numbers of cell clusters according to the density of data points. The seven clusters of NKT cells were generated according to their transcriptomic heterogeneity and the reason why only NKT2 cells are composed of 4 clusters might be because they are more

abundant than others rather than they are more heterogeneous. In **Figure 4A for reviewers**, we down-sampled NKT2 cells and found, when there are less NKT2 cells, they are grouped as single population. In addition, we directly compared the heterogeneity index of 4 subsets of NKT cells (**Figure 4B for reviewers**), and found the heterogeneity index was highest in NKT1 cells followed by in an order of NKT17, NKT2 and NKTp cells with statistical significance. Also we listed unique pathways enriched in four clusters of NKT2 cells (**Figure 5 for reviewers**) and it seems like multiple factors influence the sub-clustering of NKT2 cells.

Finally, as for the age of mice analyzed, we used adult mice to get NKT and MAIT cells because they are very rare in neonates and expand until they become adult mice. However, $\gamma\delta$ T cells are enriched in fetal to neonatal period, especially CD24^{low} populations become rare in adult mice as shown in **Figure 4C for reviewers**. Also, V γ 6+ T $\gamma\delta$ 17 cells are only fetal derived and V γ 4+ T $\gamma\delta$ 17 cells are only post-natal derived and we determined that post-natal day 5 is best to obtain significant number of CD24^{low} $\gamma\delta$ T cells with all subsets. It would be best to match the ages of mice to exclude any biological effects due to aging, but that was not technically possible. Our result showed T $\gamma\delta$ 17 cells are very similar with MAIT17 and NKT17 cells and this finding suggests that ageing has minor effects in their transcriptional natures.

Comment 4: The idea proposed here that these three populations compete for a “developmental” niche needs further explanation. What are these cells competing for exactly? The proportion of MAIT cells within the mouse thymus was recently reported to be dependent upon the microbiota. Could it be that in absence of gd T cells, the microbiota is perturbed and leads to an increase of MAIT cells? What is the “developmental niche” then?

Response: We thank the reviewer for her/his comments on a thymic niche. We used both the “thymic niche” and “developmental niche” in the previous version of the manuscript. However, we found the former is more accurate and replaced the latter with the former in the revised manuscript. As for the thymic niche, we think it generally refers to anatomic space or room in the thymus and cells compete to occupy it for their development. We are sorry for the confusion.

Although the specific involvement of microbiota for the $\gamma\delta$ T cells to suppress the development of MAIT cells is a very interesting idea, it is beyond the scope of this manuscript and technically impossible to perform within three months of revision period. In the interim, during communication between the editor and ourselves, the editor also agreed

that answering the reviewer's question about the microbiota/niche competition is out of the scope of this manuscript and we would like to address these issues in our future studies.

Comment 5: The idea that the TCR specificity of any of these populations has no role in subset differentiation is based on the share of a few gd clonotypes amongst different subsets. This occurs albeit strong biases for each subset. Therefore, does it represent the rule or an exception? MAIT cells did not display such an overlap, while NKT cells did show some clonotype overlaps between subsets. Does this mean that the repertoire of NKT cells is actually oligoclonal with expansions of particular clonotypes? This is rather surprising in light of previous results with deeper sequencing showing that the NKT repertoire is extremely polyclonal.

Response: In this manuscript, we do not argue that the TCR specificity of $\gamma\delta$ T cells has no role in their lineage decision process. In V γ 4/6 deficient mice, they had normal number of T $\gamma\delta$ 17 cells, and ROR γ t deficient mice had normal number of V γ 4+ cells that developed as T $\gamma\delta$ 1 cells. We also showed the same clonotypes were present in both T $\gamma\delta$ 1 and T $\gamma\delta$ 17 lineages and phenotypic analysis of TBET (please see our response to Comment 8 of Reviewer #2) and Gata3 deficient mice supported our conclusion. All of these findings indicate that **$\gamma\delta$ TCRs are not a single factor that determine the lineage fate of $\gamma\delta$ T cells.** We speculate that $\gamma\delta$ TCRs recognize thymic self-antigen and certain threshold of TCR signaling provides advantages to the cells to become a specific lineage (e.g., V γ 4/6 to become T $\gamma\delta$ 17 cells). **Therefore, rather than arguing $\gamma\delta$ TCRs have no roles, we argue that they are one of factors that determine their lineage fates.** Because we have not tested V γ 1, 2, 5, or 7 deficient mice (C γ 3 is pseudogene), **we cannot generalize the limited roles of $\gamma\delta$ TCRs.** However, we think our findings provide important insights to understand the lineage plasticity of $\gamma\delta$ T cells, especially T $\gamma\delta$ 17 cells, which are important for many inflammatory disorders. We included this in discussion part and highlighted in green (page 13, line 28-32).

It is indeed surprising that the TCR repertoire of innate T cells are extremely diverse even though they have canonical TCRs (we added this in page 8 line 16-17). We added **Figure S15 (page 8, line 24)** to show the statistical analysis of clonal heterogeneity between NKT, MAIT and $\gamma\delta$ T cells, which showed iNKT cells are more oligoclonal than MAIT cells. However, it requires further investigations as for the physiological relevance of this finding.

Comment 6: The identical proportion of Ki-67+ in Vg1Vd6+ rorgt+ cells in wt and Vg4/6 Ko

mice does not exclude the possibility of niche filling by a pre-existing population of such cells, since its precursor might be proliferating, not the end-product.

Response: We thank the reviewer for emphasizing this important issue and agree with reviewer that Ki-67 level is not a direct evidence of excluding possibility of niche filling by a pre-existing population of cells. This was one of three points that we proposed to explain why we think the expansion of V γ 1+ cells is not simple niche filling. The disproportional conversion of pre-existing minor T γ δ 17 cells (e.g. V γ 1+V δ 6.3+ cells) in V γ 4/6 KO mice, which cannot be explained by simple niche filling by a pre-existing population of cells, provides more convincing evidence to support our conclusion (Figure 5C, pie chart).

Comment 7: The naïve MAIT cells identified in Legoux et al. are PLZFneg and therefore cannot be similar to the M2 cluster identified here as stated in the text.

Response: As we pointed out in **Figure 1 for reviewer and our response to Comment 1 of Reviewer #2**, Legoux et al mentioned that cluster 7a and 7b express PLZF. Although they did not confirm this result by flow cytometry we think our result is consistent with theirs.

REVIEWER COMMENTS

Reviewer #2 (Remarks to the Author):

The authors have gone to great lengths to respond to all of my comments and concerns. The manuscript is much improved and I have no additional questions.

Reviewer #3 (Remarks to the Author):

Sorry for the delay.

I have now re-reviewed the manuscript by Lee et al. Although the manuscript undeniably contains a lot of data it remains extremely confusing to the reader. What is the message of the manuscript? That gd T cells, MAIT cells and iNKT cells follow an identical developmental pathway? That MAIT cells are positively selected with a broad repertoire but that this repertoire is then weaned out? How? Is there a requirement for a second TCR engagement in the course of MAIT development or are cells bearing these unconventional TCRs contaminants with no MR1 reactivity? Although it might be demanding to test, if this is part of the manuscript, it should be tested properly. That the TCR specificity of each of these populations is not a driver of functionality? What of Pereira et al., Critical role of TCR specificity in the development of Vg1/Vd6.3 innate NKTgd cells, JI 2013 then? (not cited) That these populations occupy the same "niche" and compete with one another for such "niche"? How? Why only these populations? Why not ILCs or NK cells since it was previously reported that iNKT cells might be competing with NK cells for IL-15? How would any such competition apply to the other subsets that are CD122 negative?

Further points:

Because of prior breadth of knowledge regarding iNKT cell development, this is taken as a roadmap for projecting the developmental paths of MAIT cells and gd T cells. Yet by the author's own admission, the UMAP obtained out of the sc-RNA-seq data does not appear compatible with the current model of iNKT cell development, which is itself based on a previous experiment by the senior author but has yet to be independently confirmed (i.e. that 4 days after transfer of iNKT2 cells into the thymus, the cells conserve a stable phenotype). Although UMAP is thought to conserve the global structure of the data, the authors certainly stay away from providing any potential developmental trajectory for iNKT cells based on their data. Yet, they have no problem doing so with gd and MAIT cells. Why?

It's unclear whether CD44 levels in MAIT2 track with hCD2 levels (Figures S1A and S1B) - there are more CD44^{hi} cells than CD44^{lo} cells in MAIT2 but the proportions are reversed for hCD2 expression. Why might this be?

It's inappropriate to refer to these MAIT2 cells as naive-like (Page 5, Line 93) because they express CD44

Is the high ratio of MAIT2 unique to this mouse colony? Is that why thymi from other mouse facilities do not have detectable MAIT2 proportions? Along these lines, only 6% of NKT1 even in the thymus of a 7-8 week old BALB/c mouse is on the very low side.

IL-4 staining in MAIT2 is not believable for S1C. Can you confirm whether these cells are actively secreting IL-4 through the use of an IL-4-dependent cell line like HT-2 or CTLL-2 (PMID: 9617571)?

It is unclear what the Venn diagram in figure S5D is depicting. The text states that the 349 genes are genes uniquely overexpressed in Tgd17i cells but the Venn diagram depicts genes that are at the intersection of all 4 other gd T cell subsets

The authors observe greater numbers of MAIT cells in TCRd KO mouse. A recent publication has stated that due to the nature of TCRA recombination, productive TCRd rearrangements promote MAITa rearrangements (PMID: 28591585). However, because the TCRd KO mouse was generated by inserting a neomycin-resistance gene in the Cd gene (and the presence of neomycin has been previously shown to dramatically affect TCR rearrangement in its vicinity, PMID: 22814339), could rearrangement of proximal Va genes be reduced in TCRd KO mice, leading to an increase in distal Va rearrangement? Therefore, the overall increase in MAIT cell numbers in the TCRd KO mouse could be due to increased rearrangement and not due to niche filling by MAIT cells in the absence of gd T cells.

Figure S10A and S10B - clusters G2 and G3 do not seem to have high expression of CD25 compared to any other cell and while the authors claim in the text that S10A-C depicts the signature gene set of Tgd25+ cells, no such signature was plotted on any UMAP or dot plot

What do the authors conclude from Figure S14? There appears to be a lot of noise in the FTOC experiment despite the fact that the sort strategy describes pure populations, so it's hard to conclude whether the CD25- cells actually give rise to the CD25+ cells. It's unclear what the purpose of this experiment is within the context of this manuscript.

If the MAIT cells are undergoing clonal expansion after stage 2, is this TCR-dependent? Is there any evidence for a TCR signaling signature in post-stage 2 cells? Additionally, since the authors argue that the non-canonical TCR bearing cells are not contaminants, would this suggest that these TCRs recognize MR1+ligand on DPs but not on other cells post-stage 2, leading to them being culled from the mature MAIT repertoire?

It is unfair to say that the Vg1 Vd6.3 cells were 'redirected' into the Tgd17 lineage in a Vg4/6 KO mouse. The proportion of RORgt+ Vg1 Vd6.3 cells is unchanged in the KO mouse. The numbers are higher so a niche filling could account for the increased numbers. Based on the pie chart in Figure 5, it is possible that RORgt+ Vg1+ cells (irrespective of Vd6.3 expression) are better equipped to expand in the absence of Vg4. Have the authors conducted Ki67 staining to confirm that these cells are not cycling but have been redirected? Additionally, if the cells are indeed being redirected, is the Vg1 Vd6.3 population in Tgd2 affected since this is where the cells are normally observed?

The authors frequently switch between displaying numbers and proportions of their various cell populations (in Figures 5 and 6, for example). It would be best to include both numbers and proportions for all figures so it is clear what specifically is affected in any given situation. By this token, are Tgd1 and Tgd17 numbers changed in the GATA3 cKO mouse? In Figure 6A, it appears that the GATA3 cKO has fewer GL3- CD3+ cells, suggesting that mature ab SP thymocytes are reduced in this mouse. Does that somehow affect Tgd subset proportions similar to how DPs have been demonstrated to help with gd development in trans?

Are there actually 2x as many gd T cells in a T-bet-deficient thymus? Or was this merely due to differential enrichment?

Reviewer #2 (Remarks to the Author):

The authors have gone to great lengths to respond to all of my comments and concerns. The manuscript is much improved and I have no additional questions.

We thank you for the reviewer #2 for the final acceptance of our manuscript.

Reviewer #3 (Remarks to the Author):

We carefully read all reviewer's additional comments and edited our manuscript and highlighted them in yellow.

1. I have now re-reviewed the manuscript by Lee et al. Although the manuscript undeniably contains a lot of data it remains extremely confusing to the reader. What is the message of the manuscript? That gd T cells, MAIT cells and iNKT cells follow an identical developmental pathway? That MAIT cells are positively selected with a broad repertoire but that this repertoire is then weaned out? How? Is there a requirement for a second TCR engagement in the course of MAIT development or are cells bearing these unconventional TCRs contaminants with no MR1 reactivity? Although it might be demanding to test, if this is part of the manuscript, it should be tested properly.

Response) The key messages of our manuscript are that (1) there are analogous effector subsets in NKT, MAIT and $\gamma\delta$ T cells and (2) the intrathymic development process of MAIT and $\gamma\delta$ T cells is also explained by the lineage differentiation model similar to NKT cells. During their maturation process, we showed that non-conventional portion of MAIT cells and TCR diversity of T $\gamma\delta$ 17 cells decrease, indicating there is an additional repertoire selection process after their positive selection. Although it is required to further analyze detailed molecular mechanisms of the selection process of these cells, we think they are beyond the scope of current manuscript as it is not possible to perform within 3 months of revision period. As the reviewer also pointed out, we already have undeniably a lot of data and we think these findings have novel features that have not been published elsewhere. We also have addressed the contamination issues in our previous responses (please see our responses on Q1 of reviewer #2 at 1st revision and Q2 of reviewer #2 at 2nd revision).

2. That the TCR specificity of each of these populations is not a driver of functionality? What of Pereira et al., Critical role of TCR specificity in the development of Vg1/Vd6.3 innate NKTgd cells, JI 2013 then? (not cited)

Response) Pereira et al. (J Immunol August 15, 2013, 191 (4) 1716-1723) showed that transgenic over-expression of V γ 1 TCR increased the frequency of PLZF+ NKT $\gamma\delta$ T cells. However transgenic expression of both V γ 1 and V δ 6B, which are not normally used in T $\gamma\delta$ 2 cells, only the cells rearranged V δ 6.3/6.4 as second TCR became T $\gamma\delta$ 2 cells. Similarly,

Boehmer et al (PNAS, 2009 106 (30) 12453–12458) also showed V γ 1V δ 6.3 Tg mice had expanded PLZF+ $\gamma\delta$ T cells, which induced spontaneous dermatitis. These findings do indicate that V γ 1V δ 6.3 TCR is necessary for T $\gamma\delta$ 2 lineage fate, but it does not indicate they are sufficient to become T $\gamma\delta$ 2 cells as not all cells in these Tg mice were T $\gamma\delta$ 2 cells (they have not shown exact proportions). We have shown in Figure S1E and S1F that less than 50% of T $\gamma\delta$ 2 cells are V γ 1V δ 6.3 cells in fetal and neonatal periods and V γ 1V δ 6.3 T cells actually generated T $\gamma\delta$ 2 (60-70%), T $\gamma\delta$ 1 (20-40%) and T $\gamma\delta$ 17 (5-10%) cells. As we have already responded in Q5 of reviewer #3 at second revision, we think TCR is one of critical factors dictating lineage fate of $\gamma\delta$ T cells but this alone cannot explain the phenotypes of TBET, ROR γ t and GATA3 deficient mice.

3. That these populations occupy the same “niche” and compete with one another for such “niche”? How ? Why only these populations? Why not ILCs or NK cells since it was previously reported that iNKT cells might be competing with NK cells for IL-15? How would any such competition apply to the other subsets that are CD122 negative?

Response) This is an interesting point, but it seems to be beyond the scope of this manuscript to further analyze ILC and NK cells.

Further points:

4. Because of prior breadth of knowledge regarding iNKT cell development, this is taken as a roadmap for projecting the developmental paths of MAIT cells and gd T cells. Yet by the author’s own admission, the UMAP obtained out of the sc-RNA-seq data does not appear compatible with the current model of iNKT cell development, which is itself based on a previous experiment by the senior author but has yet to be independently confirmed (i.e. that 4 days after transfer of iNKT2 cells into the thymus, the cells conserve a stable phenotype). Although UMAP is thought to conserve the global structure of the data, the authors certainly stay away from providing any potential developmental trajectory for iNKT cells based on their data. Yet, they have no problem doing so with gd and MAIT cells. Why?

Response) Please see Q3 of reviewer #3 in which we argued that developmental intermediates to NKT1 cells were not present in our analysis, probably due to their low frequencies. Capturing enough number of developmental intermediates is a pre-requisite of revealing the precursor-progeny relationships from single-cell trajectory analysis. As for MAIT and $\gamma\delta$ T cells, we could obtain enough number of cells to construct cell trajectories from progenitors to the final effectors. Most of all, we have not made any conclusions for iNKT cells from our trajectory analysis and validated our results of MAIT and $\gamma\delta$ T cells using various KO mice. Further supporting that NKT2 cells do not give rise to NKT1 cells, we showed there is minimal overlap of TCR repertoire diversity between NKT1 and NKT2 cells

(Figure S8D). Previous publications showed that GATA3 KO (I-Cheng Ho et al, 2006, J Immunol) and IL17RB KO (Taniguche et al, 2012 Plos Biology) mice have little or no IL4 producing NKT cells, but they have normal development of NKT1 cells. These are strong genetic evidence convincing the lineage differentiation model of iNKT cells. We have not included this point in the current manuscript as we have addressed this issues in our previous paper (Lee et al, 2013 Nat Immunol).

5. It's unclear whether CD44 levels in MAIT2 track with hCD2 levels (Figures S1A and S1B) - there are more CD44^{hi} cells than CD44^{lo} cells in MAIT2 but the proportions are reversed for hCD2 expression. Why might this be?

Response) We re-analyzed our data and showed that huCD2⁺ cells are in both CD44^{hi} and CD44^{low} cells in NKT, MAIT and $\gamma\delta$ T cells (added in Figure S1D). Although CD44 expression is generally considered as a marker of memory T cells, its expression was not correlated with that of IL-4 production. This is consistent with previous findings that showed IL-4 is efficiently produced from CD44^{low} NKT cells (Benlagha, 2002, Science), we think this is similar feature in MAIT2 and T $\gamma\delta$ 2 cells. We added this in result section (page4).

6. It's inappropriate to refer to these MAIT2 cells as naive-like (Page 5, Line 93) because they express CD44

Response) We agree with the reviewer's point that MAIT2 cells are not naïve as some of them are CD44^{hi} and secrete IL4 upon activation. We deleted this expression and edited the manuscript.

7. Is the high ratio of MAIT2 unique to this mouse colony? Is that why thymi from other mouse facilities do not have detectable MAIT2 proportions? Along these lines, only 6% of NKT1 even in the thymus of a 7-8 week old BALB/c mouse is on the very low side.

IL-4 staining in MAIT2 is not believable for S1C. Can you confirm whether these cells are actively secreting IL-4 through the use of an IL-4-dependent cell line like HT-2 or CTLL-2 (PMID: 9617571)?

[Redacted]

Response) Left Figure (A) shows previous paper showing the frequency of 'naïve' MAIT cells in the thymus and spleen (Logoux et al, 2019 Nat Immunol). As we have explained in figure 1 for reviewer at 2nd revision, they

are CD44^{low} MAIT2 cells. In our mouse colony, MAIT2 cell frequencies are average 6.46% among total MAITs (Figure B). Therefore, we do not think there are significant discrepancies between our and other mouse facilities for their MAIT2 cell frequencies.

We agree that 6% of NKT1 cells in 7-8 weeks old BALB/c mice are low side, but they are within normal variations. In our previous literature (Figure S7B, Lee et al, 2013 Nat Immunol, also addressed in Q1 of reviewer #2 at 1st revision) we found 5-25% of NKT1 cells among total iNKT cells in 7-8 week old BLAB/c mice.

Calculated number of MAIT2 cells is about 50 per mouse, and we can only analyze 10-20 cells in flowcytometry after MACS enrichment (we usually combine 3-5 mice together to obtain 50-100 M2 cells for facs plots). For this reason, we previously failed to purify these cells for bulk-seq analysis and that is why we decided to used total MAIT cells as a single cell analysis. We thank for your suggestion, but we need to isolate at least several hundred cells for your proposed experiments, which is not technically feasible.

8. It is unclear what the Venn diagram in figure S5D is depicting. The text states that the 349 genes are genes uniquely overexpressed in Tgd17i cells but the Venn diagram depicts genes that are at the intersection of all 4 other gd T cell subsets

Response) As the legend indicates, Venn-diagrams show number of genes overexpressed in T $\gamma\delta$ 17i cells compared to other $\gamma\delta$ T subsets. Therefore, there are 349 genes that are overexpressed in T $\gamma\delta$ 17i cells compared to all the other subset.

9. The authors observe greater numbers of MAIT cells in TCRd KO mouse. A recent publication has stated that due to the nature of TCRa recombination, productive TCRd rearrangements promote MAITa rearrangements (PMID: 28591585). However, because the TCRd KO mouse was generated by inserting a neomycin-resistance gene in the Cd gene (and the presence of neomycin has been previously shown to dramatically affect TCR rearrangement in its vicinity, PMID: 22814339), could rearrangement of proximal Va genes be reduced in TCRd KO mice, leading to an increase in distal Va rearrangement? Therefore, the overall increase in MAIT cell numbers in the TCRd KO mouse could be due to increased rearrangement and not due to niche filling by MAIT cells in the absence of gd T cells.

Response) This is an interesting point, but we do not think expanded MAIT cells are due to an artificial effects. MAIT cells (V α 19) and NKT cells (V α 14) have distal TCR V α chains and they are all gone in ROR γ t deficient mice, which cannot rearrange distal V α chains. Because TCR δ KO mice do not have expanded NKT cells we think it is unlikely that the phenotype is due to the presence of neomycin resistant cassette.

10. Figure S10A and S10B - clusters G2 and G3 do not seem to have high expression of CD25 compared to any other cell and while the authors claim in the text that S10A-C depicts

the signature gene set of Tgd25+ cells, no such signature was plotted on any UMAP or dot plot

Response) According to the reviewer's suggestion, we added violin plots for signature scores in Figure S10.

11. What do the authors conclude from Figure S14? There appears to be a lot of noise in the FTOC experiment despite the fact that the sort strategy describes pure populations, so it's hard to conclude whether the CD25⁻ cells actually give rise to the CD25⁺ cells. It's unclear what the purpose of this experiment is within the context of this manuscript.

Response) As our analysis indicated G1 is more immature population than G2, which expressed higher level of CD25 (newly added violin plot in Figure S10H), we performed validation experiment to show that $\gamma\delta^{24+25+}$ cells are not the earliest precursors. Consistent with our single cell analysis, we showed the generation of $\gamma\delta^{24+25+}$ cells from $\gamma\delta^{24+25-}$ cells in FTOC experiments (Figure S14). Although there are some variability (or noise), as long as we have highly purified CD25⁺ (95%) and CD25⁻ $\gamma\delta^{24+}$ cells (99%) cells, we think these results indicate that CD25⁻ cells give rise to CD25⁺ cells at least in FTOC condition. We edited the manuscript to clarify this point in the manuscript (page 7).

12. If the MAIT cells are undergoing clonal expansion after stage 2, is this TCR-dependent? Is there any evidence for a TCR signaling signature in post-stage 2 cells? Additionally, since the authors argue that the non-canonical TCR bearing cells are not contaminants, would this suggest that these TCRs recognize MR1+ligand on DPs but not on other cells post-stage 2, leading to them being culled from the mature MAIT repertoire?

Response) We showed that there are little clonal expansion in MAIT cells (Figure 4B). In germ free mice, we and others (Koey et al, 2016 Nat Immunol) observed that CD24^{low} MAIT cells disappear more dramatically than CD24^{high} MAIT cells. Also recent report (Lehoux et al, 2019, Science) showed that microbial metabolite control the thymic development of MAIT cell by directly engaging their TCRs. Based on these results, it is possible that MAIT cells require extrathymic ligands for their maturation and canonical TCRs are more responsive to them. However, when we compared signature scores of TCR signaling signature genes between canonical and non-canonical MAIT cells for each cell cluster, there were no difference between them (below figure). Therefore, it is unlikely that TCR dependent signaling alone determines the selection of canonical MAIT cells and requires further investigation to understand the selection mechanism of canonical MAIT cells after positive selection.

Box plot showing expression of TCR signature genes obtained from a GO term (GO:0050862, positive regulation of T cell receptor signaling pathway) in MAITs with canonical and non-canonical TCRs in each cluster. * $P < 0.01$; NS, non-significant ($P > 0.01$, Wilcoxon rank sum test).

13. It is unfair to say that the Vg1 Vd6.3 cells were 'redirected' into the Tgd17 lineage in a Vg4/6 KO mouse. The proportion of RORgt+ Vg1 Vd6.3 cells is unchanged in the KO mouse. The numbers are higher so a niche filling could account for the increased numbers. Based on the pie chart in Figure 5, it is possible that RORgt+ Vg1+ cells (irrespective of Vd6.3 expression) are better equipped to expand in the absence of Vg4. Have the authors conducted Ki67 staining to confirm that these cells are not cycling but have been redirected? Additionally, if the cells are indeed being redirected, is the Vg1 Vd6.3 population in Tgd2 affected since this is where the cells are normally observed?

Response) Please see our prior responses to this comment (main question of reviewer #3 at 1st revision and Q6 of reviewer #3 of 2nd revision). We previously compared Ki67 expression levels of Tγδ17 cells between WT and Vγ4/6 KO mice and found there is no difference (Figure 6 for reviewers at 1st revision and newly added Figure S16B). Also, as the reviewer suggested, we compared the frequencies of Vγ1+Vδ6+ among total Tγδ2 cells from WT and Vγ4/6 KO mice and found Vγ4/6 KO mice had indeed lower frequencies of Vγ1+Vδ6+ cells (newly added figure S16C). As the reviewer suggested, these results suggest that the fate of Vγ1+Vδ6+ was redirected into Tγδf17 cells in Vγ4/6 KO mice. However, we agree with reviewer's point that we cannot make a solid conclusion from these experiments and **edited manuscript and toned down our conclusion (page 10-11)**.

14. The authors frequently switch between displaying numbers and proportions of their various cell populations (in Figures 5 and 6, for example). It would be best to include both numbers and proportions for all figures so it is clear what specifically is affected in any given situation. By this token, are Tgd1 and Tgd17 numbers changed in the GATA3 cKO mouse? In Figure 6A, it appears that the GATA3 cKO has fewer GL3- CD3+ cells, suggesting that mature ab SP thymocytes are reduced in this mouse. Does that somehow affect Tgd subset proportions similar to how DPs have been demonstrated to help with gd development in trans?

Response) We compared number of cells for all data (TBET, RORγt and Vγ4/6 KO) except GATA3 cKO mice because they have smaller thymi than WT mice due to the much reduced numbers of both DP and SP thymocytes (I-Cheng Ho, 2004 Immunity). In our colony,

numbers of total thymocytes of Gata3 cKO mice were less than 30% of that of WT. As DP thymocytes are required for trans-conditioning of $\gamma\delta$ T cells (Hayday, 2005, Science), we assumed that the reduction of $\gamma\delta$ T cells is proportional to that of DP thymocytes and compared relative frequencies of each subset of $\gamma\delta$ T cells among total $\gamma\delta$ T cells. We added this point to the manuscript (page 11) and added frequency data in Figures 5C 6D, 6C and 6D.

Hayday et al also showed that TCR α KO mice have no defect in $\gamma\delta$ T cell phenotype, suggesting SP thymocytes do not affect $\gamma\delta$ T cell development. However, we have not analyzed $\gamma\delta$ T cell development in terms of T $\gamma\delta$ 1, T $\gamma\delta$ 2 and T $\gamma\delta$ 17 developmental schematics in TCR α KO mice, and unfortunately TCR α KO mice is not available in our facility currently.

15. Are there actually 2x as many gd T cells in a T-bet-deficient thymus? Or was this merely due to differential enrichment?

Response) Figure 6C is one particular example, and there is no difference of total number of $\gamma\delta$ T cells between WT and Tbet KO mice. We included this in Figure 6C graph.

REVIEWERS' COMMENTS:

Reviewer #2 (Remarks to the Author):

I am focussing my response to the comments from reviewer 3 and the authors responses to these comments. I think some modifications to the text should adequately cover these last remaining concerns.

7. Is the high ratio of MAIT2 unique to this mouse colony? Is that why thymi from other mouse facilities do not have detectable MAIT2 proportions?

Response. As we have explained in figure 1 for reviewer at 2nd revision, they are CD44^{low} MAIT2 cells. In our mouse colony, MAIT2 cell frequencies are average 6.46% among total MAITs (Figure B). Therefore, we do not think there are significant discrepancies between our and other mouse facilities for their MAIT2 cell frequencies.

I am concerned about the authors response to reviewer 3 on this point. The authors response depends on MAIT 2 cells being CD44^{low}, CD62L⁺ and they highlight this in the figure they cite from Legoux et al 2019 Nature Immunol. But this clashes with response in the previous review after I asked about this – where they agree that most MAIT 2 cells are CD44⁺. The CD44^{lo} CD62L⁺ cells identified in the Legoux paper do not really support their argument that these are MAIT2 cells and that such cells are demonstrated in other studies. I think this MAIT 2 observation is the one part of the authors paper that needs careful attention. Do MAIT 2 cells really exist in the thymus and are they CD44^{lo} or CD44⁺. The authors seem to be switching between CD44^{lo} or high phenotype here. The authors should describe their MAIT 2 population more carefully in the text.

8. It is unclear what the Venn diagram in figure S5D is depicting. The text states that the 349 genes are genes uniquely overexpressed in Tgd17i cells but the Venn diagram depicts genes that are at the intersection of all 4 other gd T cell subsets

Response. As the legend indicates, Venn-diagrams show number of genes overexpressed in Tγδ17i cells compared to other γδ T subsets. Therefore, there are 349 genes that are overexpressed in Tγδ17i cells compared to all the other subset.

Rev 3 is correct - this is a confusing diagram that does not follow the normally layout for a venn diagram. An overlapping part of the diagram would normally mean that those 349 genes are coexpressed by the other gd T cell subsets. While the legend tries to explain why this is different from a normal venn diagram, this is not a very clear way to make this point.

9. The authors observe greater numbers of MAIT cells in TCRd KO mouse. A recent publication has stated that due to the nature of TCRA recombination, productive TCRd rearrangements promote MAITa rearrangements (PMID: 28591585). However, because the TCRd KO mouse was generated by inserting a neomycin-resistance gene in the Cd gene (and the presence of neomycin has been previously shown to dramatically affect TCR rearrangement in its vicinity, PMID: 22814339), could rearrangement of proximal Va genes be reduced in TCRd KO mice, leading to an increase in distal Va rearrangement? Therefore, the overall increase in MAIT cell numbers in the TCRd KO mouse could be due to increased rearrangement and not due to niche filling by MAIT cells in the absence of gd T cells.

Response. This is an interesting point, but we do not think expanded MAIT cells are due to an artificial effects. MAIT cells (Va19) and NKT cells (Va14) have distal TCR Va chains and they are all gone in RORγt deficient mice, which cannot rearrange distal Va chains. Because TCRδ KO mice do not have expanded NKT cells we think it is unlikely that the phenotype is due to the presence of neomycin resistant cassette.

Rev 3 raises a good point. That NKT cells are not also expanded is not really adequate proof that MAIT cells are not expanded because they may be differently regulated. The authors could examine pre-selected TCRd KO vs WT DP thymocytes for Va19 usage, but this is not a trivial undertaking at this late stage. They should at least acknowledge the possibility that MAIT cell expansion is due to altered TCRa gene rearrangement in TCRd KO mice.

Reviewer #2 (Remarks to the Author):

I am focussing my response to the comments from reviewer 3 and the authors responses to these comments. I think some modifications to the text should adequately cover these last remaining concerns.

Response: We highly appreciate reviewer#2 for his or her meditation of our responses and modified our manuscript according to the suggestions.

7. Is the high ratio of MAIT2 unique to this mouse colony? Is that why thymi from other mouse facilities do not have detectable MAIT2 proportions?

Response. As we have explained in figure 1 for reviewer at 2nd revision, they are CD44^{low} MAIT2 cells. In our mouse colony, MAIT2 cell frequencies are average 6.46% among total MAITs (Figure B). Therefore, we do not think there are significant discrepancies between our and other mouse facilities for their MAIT2 cell frequencies.

I am concerned about the authors response to reviewer 3 on this point. The authors response depends on MAIT 2 cells being CD44^{low}, CD62L⁺ and they highlight this in the figure they cite from Legoux et al 2019 Nature Immunol. But this clashes with response in the previous review after I asked about this – where they agree that most MAIT 2 cells are CD44⁺. The CD44^{lo} CD62L⁺ cells identified in the Legoux paper do not really support their argument that these are MAIT2 cells and that such cells are demonstrated in other studies. I think this MAIT 2 observation is the one part of the authors paper that needs careful attention. Do MAIT 2 cells really exist in the thymus and are they CD44^{lo} or CD44⁺. The authors seem to be switching between CD44^{lo} or high phenotype here. The authors should describe their MAIT 2 population more carefully in the text.

Response: We are sorry for the confusion. MAIT2 cells have wide range of CD44 expression (Figure S1A) and naïve MAIT cells expressing PLZF defined by Logoux et al (2019 Nat Immunol) belongs to CD44^{lo} MAIT2 cells. In our response, we meant that naïve MAIT cells are part of MAIT2 cells. At this time, it is hard to compare the exact frequencies of

MAIT2 cells between our colonies and other's because naïve MAIT cells were originally defined as a result of single cell analysis. For this reason, we used indirect method and compared the frequencies of CD44^{lo}CD62L⁺ MAIT cells and MAIT2 cells.

8. It is unclear what the Venn diagram in figure S5D is depicting. The text states that the 349 genes are genes uniquely overexpressed in Tgd17i cells but the Venn diagram depicts genes that are at the intersection of all 4 other gd T cell subsets Response. As the legend indicates, Venn-diagrams show number of genes overexpressed in Tγδ17i cells compared to other γδ T subsets. Therefore, there are 349 genes that are overexpressed in Tγδ17i cells compared to all the other subset.

Rev 3 is correct - this is a confusing diagram that does not follow the normally layout for a venn diagram. An overlapping part of the diagram would normally mean that those 349 genes are coexpressed by the other gd T cell subsets. While the legend tries to explain why this is different from a normal venn diagram, this is not a very clear way to make this point.

Response: As the reviewer commented, we removed the Venn diagram labeled by Tγδ subsets and added volcano plots to clarify the step of the pairwise comparison between Tγδ17i and other subsets. Now, an updated figure in S5D depicts that Tγδ17i-specific genes are the intersection of DEGs between Tγδ17i and other cell Tγδ subsets.

9. The authors observe greater numbers of MAIT cells in TCRd KO mouse. A recent publication has stated that due to the nature of TCRa recombination, productive TCRd rearrangements promote MAITa rearrangements (PMID: 28591585). However, because the TCRd KO mouse was generated by inserting a neomycin-resistance gene in the Cd gene (and the presence of neomycin has been previously shown to dramatically affect TCR rearrangement in its vicinity, PMID: 22814339), could rearrangement of proximal Va genes be reduced in TCRd KO mice, leading to an increase in distal Va rearrangement? Therefore, the overall increase in MAIT cell numbers in the TCRd KO mouse could be due to increased rearrangement and not

due to niche filling by MAIT cells in the absence of gd T cells.

Response. This is an interesting point, but we do not think expanded MAIT cells are due to an artificial effects. MAIT cells (V α 19) and NKT cells (V α 14) have distal TCR V α chains and they are all gone in ROR γ t deficient mice, which cannot rearrange distal V α chains. Because TCR δ KO mice do not have expanded NKT cells we think it is unlikely that the phenotype is due to the presence of neomycin resistant cassette.

Rev 3 raises a good point. That NKT cells are not also expanded is not really adequate proof that MAIT cells are not expanded because they may be differently regulated. The authors could examine pre-selected TCR δ KO vs WT DP thymocytes for Va19 usage, but this is not a trivial undertaking at this late stage. They should at least acknowledge the possibility that MAIT cell expansion is due to altered TCRA gene rearrangement in TCR δ KO mice.

Response: Thank you for the suggestion. We included this possibility at the 2nd paragraph of discussion.